JCB Journal of Cell Biology

# Centrosome age breaks spindle size symmetry even in cells thought to divide symmetrically

Alexandre Thomas[1,2] and Patrick Meraldi[1,2]

**Centrosomes are the main microtubule-organizing centers in animal cells. Due to the semiconservative nature of centrosome duplication, the two centrosomes differ in age. In asymmetric stem cell divisions, centrosome age can induce an asymmetry in half-spindle lengths. However, whether centrosome age affects the symmetry of the two half-spindles in tissue culture cells thought to divide symmetrically is unknown. Here, we show that in human epithelial and fibroblastic cell lines centrosome age imposes a mild spindle asymmetry that leads to asymmetric cell daughter sizes. At the mechanistic level, we show that this asymmetry depends on a cenexin-bound pool of the mitotic kinase Plk1, which favors the preferential accumulation on old centrosomes of the microtubule nucleation–organizing proteins pericentrin, γ-tubulin, and Cdk5Rap2, and microtubule regulators TPX2 and ch-TOG. Consistently, we find that old centrosomes have a higher microtubule nucleation capacity. We postulate that centrosome age breaks spindle size symmetry via microtubule nucleation even in cells thought to divide symmetrically.**

## Introduction

During mitosis, cells assemble a bipolar spindle to ensure faithful chromosome segregation between the two daughter cells (Prosser and Pelletier, 2017). This assembly relies on the dynamicity of spindle microtubules and the cooperative action of microtubule-associated proteins (MAPs) (Petry, 2016). In most animal cells, microtubules are first nucleated from the centrosome, the main microtubule organizing center during mitosis (Sanchez and Feldman, 2017; Meraldi, 2016). Microtubules subsequently also emerge from chromosomes and kinetochores, the microtubule-binding sites on chromosomes, and by branching off existing microtubules via the augmin complex (Heald et al., 1996; Gruss et al., 2002; Goshima et al., 2008; Uehara et al., 2009; Petry et al., 2013; David et al., 2019; Wu et al., 2023). Although centrosomes are dispensable for spindle assembly in various cell types (Heald et al., 1996; Basto et al., 2006; Bobinnec et al., 1998; Khodjakov et al., 2000), and absent in plants, planarians, and oocytes (Yi and Goshima, 2018; Mogessie et al., 2018; Azimzadeh et al., 2012), they regulate mitotic spindle assembly and are crucial for spindle orientation and faithful chromosome segregation (Khodjakov and Rieder, 2001; Buffin et al., 2007; Basto et al., 2006; Hayward et al., 2014; Sir et al., 2013; Dudka et al., 2019).

Centrosomes are composed of two centrioles surrounded by a pericentriolar material (PCM) required for microtubule nucleation (Conduit et al., 2015; Vasquez-Limeta and Loncarek, 2021).

They duplicate once per cell cycle in a semiconservative manner, as each centriole seeds a new daughter centriole resulting in centrosomes of different ages (Nigg and Stearns, 2011; Nigg and Holland, 2018; Fırat-Karalar and Stearns, 2014). The duplicated old centrosome contains the oldest centriole called the grandmother centriole and its daughter centriole, while the young centrosome contains a mother centriole and its daughter centriole (Sullenberger et al., 2020). The grandmother centriole is longer than the mother centriole and possesses distal (DA) and sub-distal (SDA) appendages, while mother centrioles only possess DA (Kong et al., 2020; Sullenberger et al., 2020). At mitotic entry, both centrosomes mature, massively expanding the PCM that drives microtubule nucleation (Conduit et al., 2015; Palazzo et al., 2000). PCM expansion depends on the mitotic kinase Plk1, which phosphorylates the pericentriolar proteins pericentrin and Cdk5Rap2/Cep215 (Lee and Rhee, 2011; Conduit et al., 2014; Ohta et al., 2021). This promotes the recruitment of the γ-tubulin ring complex, which nucleates microtubules (Ohta et al., 2021; Fong et al., 2008; Choi et al., 2010; Dictenberg et al., 1998; Moritz et al., 1995). As the mitotic spindle forms, the centrioles and the PCM become embedded in the spindle poles, where minus-end binding proteins accumulate (Akhmanova and Steinmetz, 2019).

Centrosomes play an important role in the regulation of the mitotic spindle size (Dudka and Meraldi, 2017). In early

[1]Department of Cell Physiology and Metabolism, Faculty of Medicine, University of Geneva, Geneva, Switzerland;   [2]Translational Research Centre in Onco-hematology, Faculty of Medicine, University of Geneva, Geneva, Switzerland.

Correspondence to Patrick Meraldi: patrick.meraldi@unige.ch.



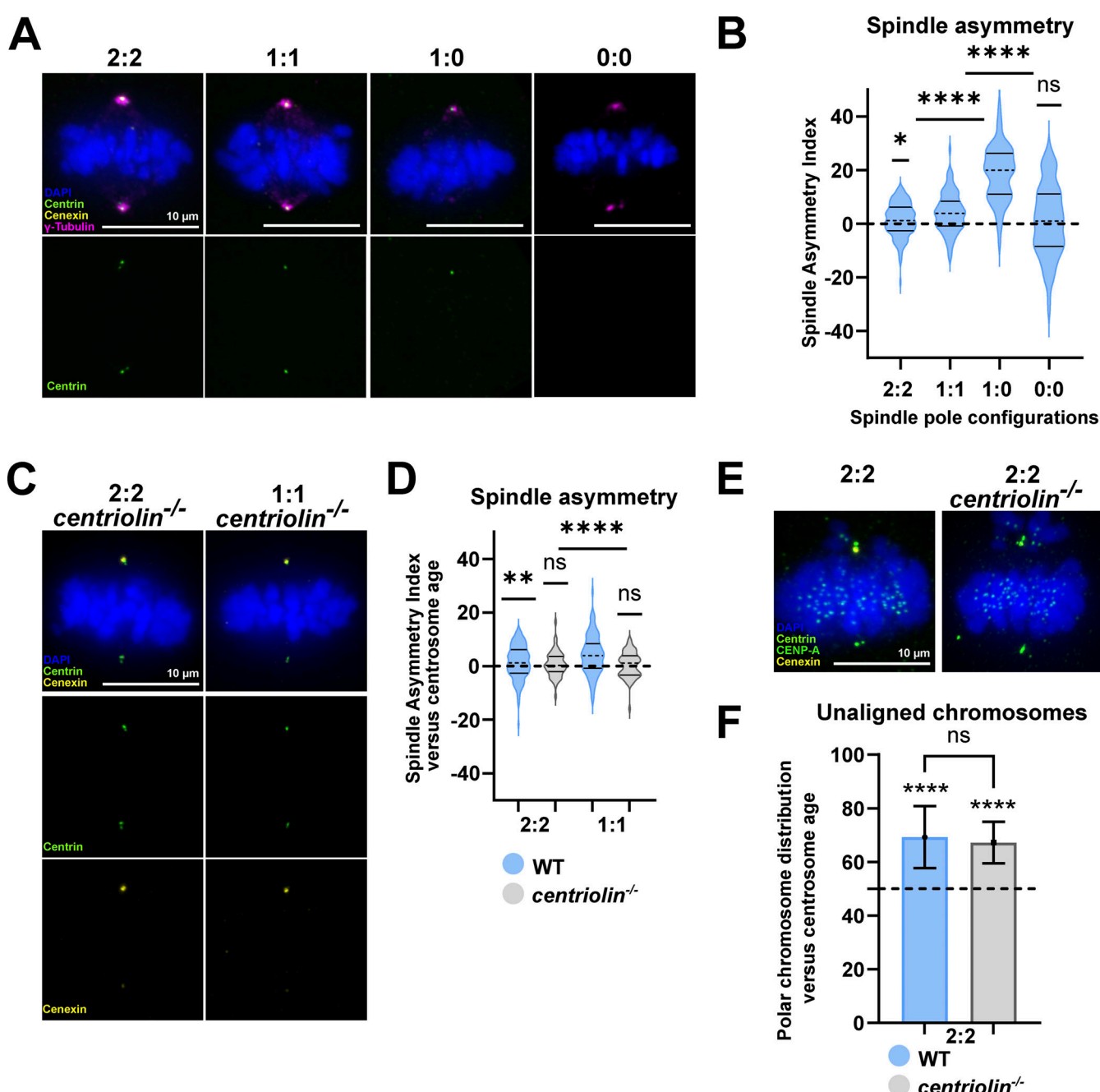

*Ceanorahbditis elegans* and zebrafish embryos, spindle size scales with centrosome size (Greenan et al., 2010; Rathbun et al., 2020). Moreover, *Chlamydomonas reinhardtii*, *C. elegans*, and human cells with an unequal number of centrioles at each pole generate half-spindles of unequal size (Keller et al., 2010; Greenan et al., 2010; Dudka et al., 2019; Tan et al., 2015). In the extreme case, human cells with one centriole in one spindle pole and no centriole in the other spindle pole form spindles that differ up to 20% in their half-spindle length (Dudka et al., 2019). Spindle size asymmetry affects the position of the cytokinetic furrow leading to asymmetric cell daughter size (Dudka et al., 2019; Tan et al., 2015). At the molecular level, spindle size is controlled by several MAPs located at spindle poles (Goshima and Scholey, 2010; Dumont and Mitchison, 2009; Lacroix and Dumont, 2022). These include ch-TOG, CLASP, or EB1, which regulate microtubule polymerization, but also microtubule severing enzymes, such as katanin, and microtubule depolymerases of the 8- and 13-kinesin families (Kif18A, Kif2A, and MCAK) (Reber et al., 2013; Barr and Gergely, 2008; Goshima et al., 2005; Maiato et al., 2005; McNally et al., 2006; Mayr et al., 2007; Gaetz and Kapoor, 2004). These latter enzymes also control the size of meiotic spindles in different *Xenopus* species, where elevated levels of katanin or Kif2A reduce spindle length in a species-dependent manner (Loughlin et al., 2011; Miller et al., 2019). In human cells and *Xenopus laevis* extracts, the mitotic spindle size also depends on the ability of the Aurora-A/TPX2 complex to regulate microtubule nucleation (Bird and Hyman, 2008; Fu et al., 2015; Helmke and Heald, 2014), while in *C. elegans* embryos, spindle size correlates with the ability to recruit microtubule nucleating material at centrosomes (Greenan et al., 2010). Finally, the asymmetric spindle of the one-cell-stage leech *Hellobdella robusta* relies on the downregulation of the microtubule nucleator γ-tubulin on one spindle pole (Ren and Weisblat, 2006).

A final important aspect of centrosome behavior is centrosome age. Old and young centrosomes are segregated in a stereotypical manner in many stem cell divisions, such as the budding yeast *Saccharomyces cerevisiae*, murine neuroprogenitors, and *Drosophila melanogaster* neuroblasts and male germline stem cell divisions (Chen and Yamashita, 2021; Yamashita et al., 2007; Wang et al., 2009; Pereira et al., 2001; Januschke et al., 2011; Conduit and Raff, 2010). Centrosome-age-dependent distribution has been linked to the asymmetric assembly of the primary cilium in daughter cells in mice (Anderson and Stearns, 2009) and to cellular senescence in yeast (Shcheprova et al., 2008). Centrosome age can also affect the relative half-spindle length in asymmetric stem cell divisions. In *D. melanogaster* neuroblasts, the half-spindle associated to the young centrosome, which is inherited by the stem cell, is longer (Fuse et al., 2003; Roubinet et al., 2017). At the molecular level, the spindle size asymmetry is controlled by the kinase Plk1$^{POLO}$, which primes centrosome asymmetry (Januschke et al., 2013; Gallaud et al., 2020). In somatic human tissue culture cells, centrosome age affects via differential kinetochore–microtubule stability the resolution of polar chromosomes in prometaphase and promotes the segregation of non-disjoint chromosomes towards the daughter cell inheriting the old centrosome in anaphase (Gasic et al., 2015; Colicino et al., 2019). However, whether centrosome age also affects spindle size in somatic tissue culture cells, and thus the relative spindle size symmetry, is unknown.

Here, we use human epithelial and fibroblastic cell lines to show that centrosome age imposes a subtle but persistent mitotic spindle size asymmetry, and that spindle size asymmetry translates into an equivalent asymmetry of the nascent daughter cell size. We demonstrate that the polar chromosome asymmetry and the spindle size asymmetry can be experimentally uncoupled, pointing to the existence of distinct spindle asymmetries. At the molecular level, spindle size asymmetry depends on proteins required for microtubule nucleation that are themselves asymmetrically distributed on old and young centrosomes. This asymmetric recruitment is regulated by a pool of Plk1 bound to cenexin on old centrosomes, which drives an asymmetric PCM expansion in early mitosis to break spindle size symmetry. Our study therefore highlights how even in somatic cells thought so far to divide symmetrically, centrosome age breaks spindle and cell division symmetries in multiple ways.

## Results

### Centrosome age breaks spindle size symmetry in cells thought to divide symmetrically

To investigate the contribution of centrosome age to spindle size symmetry, we used non-cancerous human retina pigment epithelial cells immortalized with telomerase (hTert-RPE1, called RPE1 hereafter) expressing the centriolar marker GFP-centrin1. In a previous study that did not take centrosome age into account, we had reported that the bulk population of RPE1 cells has on average symmetric spindles (Dudka et al., 2019). Here, we analyzed spindle size (a)symmetry versus centrosome age in cells with different centriole configurations: non-treated WT cells containing two centrioles at each pole (hereafter called 2:2 cells); cells with one centriole at one spindle pole and no centriole on the other pole as positive control for spindle asymmetry (hereafter caller 1:0 cells); cells without any centrioles as negative control (hereafter called 0:0 cells), and since daughter centrioles have also been implicated in spindle length control, cells with one centriole at each spindle pole, (hereafter called 1:1 cells; Fig. S1 A) (Tan et al., 2015; Dudka et al., 2019). To obtain such centriole configurations, we inhibited centriole duplication with the Plk4 inhibitor centrinone for 24 (1:1), 48 (1:0), or 72 h (0:0; Fig. S1 A) (Wong et al., 2015).

Cells were fixed, stained for γ-tubulin to visualize spindle poles, and for the SDA marker cenexin to differentiate between old and young centrosomes (Fig. 1 A) (Ishikawa et al., 2005). We quantified in single cells the two half-spindle lengths in 3D by measuring the distance between the center of the centrin signal in 2:2 or 1:1 cells, or the γ-tubulin signal in centriole-free poles versus the center of gravity of the DNA mass (Fig. 1, B and C). By subtracting the half-spindle length associated with the young centrosome from the half-spindle length of the old centrosome and by dividing this difference by the sum of both values, we obtained for each cell a spindle (a)symmetry index (SAI hereafter; Fig. S1 C; in 0:0 cells the two half-spindles were randomly assigned). An SAI of +5%, for example, indicated that this cell possessed an “old” half-spindle that was 5% longer than the

Figure 1. **Centrosome age breaks spindle size symmetry in RPE1 cells. (A)** Immunofluorescence images of 2:2, 1:1, 1:0, and 0:0 RPE1 GFP-centrin1 metaphase cells. Cells are stained with cenexin (yellow) and γ-tubulin (magenta) antibodies, DAPI (blue), and GFP-centrin1 (green). Note that these images are also shown in Fig. S2 B. **(B)** Quantification of the SAI of 2:2, 1:1, 1:0, and 1:0 cells. The dashed line and full lines of each condition represent the mean and SD, respectively. Each condition was compared with the null hypothesis, a symmetric spindle centered at 0 (dashed line). For 2:2 cells, mean SAI = 1.4 ± 6.2%, n = 133 cells, P = 0.0104; for 1:1 cells = 4.1 ± 7.3%, n = 96 cells, P < 0.0001; for 1:0 cells = 19.3 ± 11.2%, n = 111 cells, P < 0.0001; and for 0:0 cells = 0.9 ± 13.3%, n = 77 cells, P = 0.54 in one-sample t tests. **(C)** Immunofluorescence images of 2:2 and 1:1 centriolin$^{-/-}$ RPE1 metaphase cells, stained with DAPI (blue), GFP-centrin1 (green), and cenexin antibody (yellow). **(D)** SAI quantification of 2:2 and 1:1 WT RPE1 cells (light blue) versus centriolin$^{-/-}$ RPE1 cells (light grey). For 2:2 centriolin$^{-/-}$ RPE1 cells, mean SAI = 0.7 ± 4.7%, n = 56 cells, P = 0.3573; and for 1:1 centriolin$^{-/-}$ RPE1 cells, mean SAI = 0.4 ± 4.7%, n = 65 cells, P = 0.4229 in one-sample t tests. **(E)** Immunofluorescence images of 2:2 WT GFP-centrin1 and 2:2 centriolin$^{-/-}$ GFP-centrin1 RPE1 metaphase cells treated with nocodazole and stained with cenexin (yellow) and CENP-A (green) antibodies, DAPI (blue), and GFP-centrin1 (green). **(F)** Quantification of the percentage of polar chromosome distribution versus centrosome age. The dashed line represents a symmetric distribution (50%) of the polar chromosomes between the two poles. Means of 69.2 ± 11.6% (from 223 polar chromosomes in 86 cells) in WT 2:2 cells, and 67.2 ± 7.8% (from 203 polar chromosomes in 169 cells) in centriolin$^{-/-}$ 2:2 cells. P values of < 0.0001 for both conditions using binomial tests to compare individual conditions to a random distribution of 50%. P value of 0.8902 using a Fisher's exact test to compare the two conditions. All scale bars = 10 μm. ns = not significant; P ≤ 0.05 = *; P ≤ 0.01 = **; P ≤ 0.0001 = ****.

average or 10% longer than the "young" half-spindle. The distribution of the SAIs in each condition (2:2, 1:1, 1:0, and 0:0 cells) was statistically compared to the null hypothesis, a symmetric spindle with an SAI of 0. Consistent with previous studies, 1:0 cells assembled highly asymmetric spindles (mean SAI of 19.3 ± 11.2%, $n$ = 111 cells, P < 0.0001), while in contrast 0:0 cells displayed a broad range of SAI centered around 0 (mean SAI of 0.9 ± 13.3%, $n$ = 77 cells, P = 0.5414; Fig. 1, A and B) (Dudka et al., 2019). The SAI distribution in 2:2 and 1:1 cells showed a mild asymmetry with means of 1.4 ± 6.2%, $n$ = 133 cells (2:2) and 4.1 ± 7.3%, $n$ = 96 cells (1:1), indicating that half-spindles associated with the old centrosomes were longer in 2:2 and 1:1 cells (P = 0.0104 and P = < 0.0001; Fig. 1, A and B). If centrosome age dictates spindle size asymmetry, we reasoned that it should depend on the SDA, the main structural difference between the grandmother at old and the mother centriole at young centrosomes. We therefore tested whether abrogation of SDA imposed symmetric spindles in 2:2 and 1:1 cells. Since we had to use cenexin as a centrosome age marker, we used RPE1 cells in which centriolin, an SDA protein downstream of cenexin, was deleted by CRISPR (Mazo et al., 2016). This enabled us to differentiate between old and young centrosomes while removing most subdistal appendage proteins. When we measured the SAI, we found symmetric spindles in both in 2:2 (0.7 ± 4.7%, $n$ = 56 cells, P = 0.3573) and 1:1 centriolin$^{-/-}$ cells (0.4 ± 4.7%, $n$ = 65 cells, P = 0.4229; Fig. 1, C and D). We conclude that even in cells thought to divide symmetrically, centrosome age breaks the symmetry of the mitotic spindle size. Moreover, the fact that 1:1 cells showed a persistently higher spindle size asymmetry raised the possibility that the presence of daughter centrioles dampens centrosome-age-dependent size asymmetry.

Since polar chromosomes (unaligned chromosomes located behind spindle poles) preferentially accumulate at the old pole in a cenexin-dependent manner (Gasic et al., 2015; Colicino et al., 2019), we next tested whether the loss of centriolin would also impose a symmetric polar chromosome distribution. Surprisingly, this was not the case as both in WT (69.2 ± 11.6%) and centriolin$^{-/-}$ cells (67.2 ± 7.8%) most polar chromosomes were associated with the old centrosome ($N$ = 203 and 223 polar chromosomes, $n$ = 86 and 169 cells, respectively, P = 0.8902; Fig. 1, E and F). This indicated that the centrosome-age-dependent asymmetries in terms of polar chromosomes and half-spindle sizes can be uncoupled and that the molecular mechanisms governing these processes must differ.

**Centrosome age breaks the symmetry of daughter cell size**

In human cells, the position of the metaphase plate at anaphase onset influences the position of the actomyosin contractile ring during cytokinesis (Dudka et al., 2019; Tan et al., 2015). Therefore, an off-center positioning of the metaphase plate or the spindle itself can lead to asymmetric daughter cell sizes (Tan et al., 2015). Spindle asymmetries can, however, diminish as cells approach anaphase (Dudka et al., 2019; Tan et al., 2015). We therefore monitored by live cell imaging metaphase 2:2 GFP-centrin1/mScarlet-cenexin RPE1 cells in the presence of SiR-DNA to plot the position of the old and young centrosomes versus the chromosome mass and tested if spindle size asymmetry persisted

until anaphase onset (Fig. S1 D and Video 1). Our live cell data confirmed that metaphase half-spindles associated with old centrosomes were on average longer, with a mean SAI of 2.7 ± 5.5% ($n$ = 28 cells, P = 0.0145) and that at the last time point before anaphase onset ($t$ = −1 min), spindle size asymmetry persisted (mean SAI of 3.8 ± 5.6%, $n$ = 28 cells, P = 0.0012; Fig. 2 A). We therefore tested whether this asymmetry translated into a daughter cell asymmetry by probing daughter cell size (a) symmetry versus centrosome age in fixed late telophase 2:2, 1:1, and 0:0 cells (Fig. 2 B and Fig. S1 E, note that in telophase both centrosomes contain cenexin, but the cenexin signal is higher on the old centrosome; Kong et al., 2014). Since at this stage, cells have not yet re-adhered, we used the maximal sphere area of each daughter cell in 3D as a proxy for cell size, as previously described (Kiyomitsu and Cheeseman, 2013). In both 2:2 and 1:1 cells, the values were larger in the cell inheriting the old centrosome (2 ± 5.2%, $n$ = 65 cells, P = 0.0025; and 4.7 ± 7.5%, $n$ = 58 cells, P < 0.0001; Fig. 2 C). In contrast, in 0:0 cells daughter cell sizes were symmetric (area asymmetry of 0.3 ± 10.5%, $n$ = 70 cells, P = 0.8370; Fig. 2 D). To exclude that the observed size asymmetry originates from an asymmetric position of the spindle itself, we also measured the average distance between old or young centrosome and the cell cortex in metaphase and found no significant difference in either fixed (6.8 ± 1.2 μm for the old and 7.1 ± 1.5 μm for the young centrosomes, $n$ = 44 cells, P = 0.2135) or live cells just before the onset of anaphase (6.7 ± 1.7 μm for the old and 7.3 ± 2 μm for the centrosomes, $n$ = 28 cells, P = 0.4491; Fig. 2, E and F; Fig. S1 F; and Video 1). To confirm that the asymmetry in daughter cell size depends on centrosome age, we quantified daughter cell sizes in 1:1 centriolin$^{-/-}$ RPE1 cells and found no asymmetry (−0.4 ± 5.8%, $n$ = 54 cells, P = 0.6246; Fig. 2, G and H). We concluded that by breaking spindle size symmetry, centrosome age also leads to asymmetric daughter cell sizes.

**Centrosome age breaks spindle size symmetry via microtubule nucleation**

We next aimed to identify the molecular mechanisms by which centrosome age breaks spindle size symmetry and generally test how centrosomes affect spindle size. We quantified the relative abundance of a series of centrosome and spindle pole proteins that have been implicated in spindle size control. For each cell we plotted, based on 3D immunofluorescence images, the relative protein distribution asymmetry between the two poles versus the SAI (Fig. 3, A and B; and Fig. S2). We carried out this analysis on 2:2 and 1:1 cells, which show a centrosome-age-dependent spindle asymmetry, but also in 1:0 and 0:0 cells. This allowed us to investigate whether the protein abundance distribution correlates specifically with centrosome-age-dependent spindle asymmetry, or more generally to spindles with an asymmetric distribution of centrioles, or no centrosomes at all. From these plots, we extracted two key values: the Pearson's R correlation coefficient, which reflects how protein abundance distribution correlates with spindle (a)symmetry, and the slope of the linear regression, which indicates the potential degree by which a protein might contribute to spindle (a)symmetry (Fig. 3 B, Fig. S2, and Table 1). We considered proteins as potential drivers of spindle (a)symmetry when the correlation coefficient

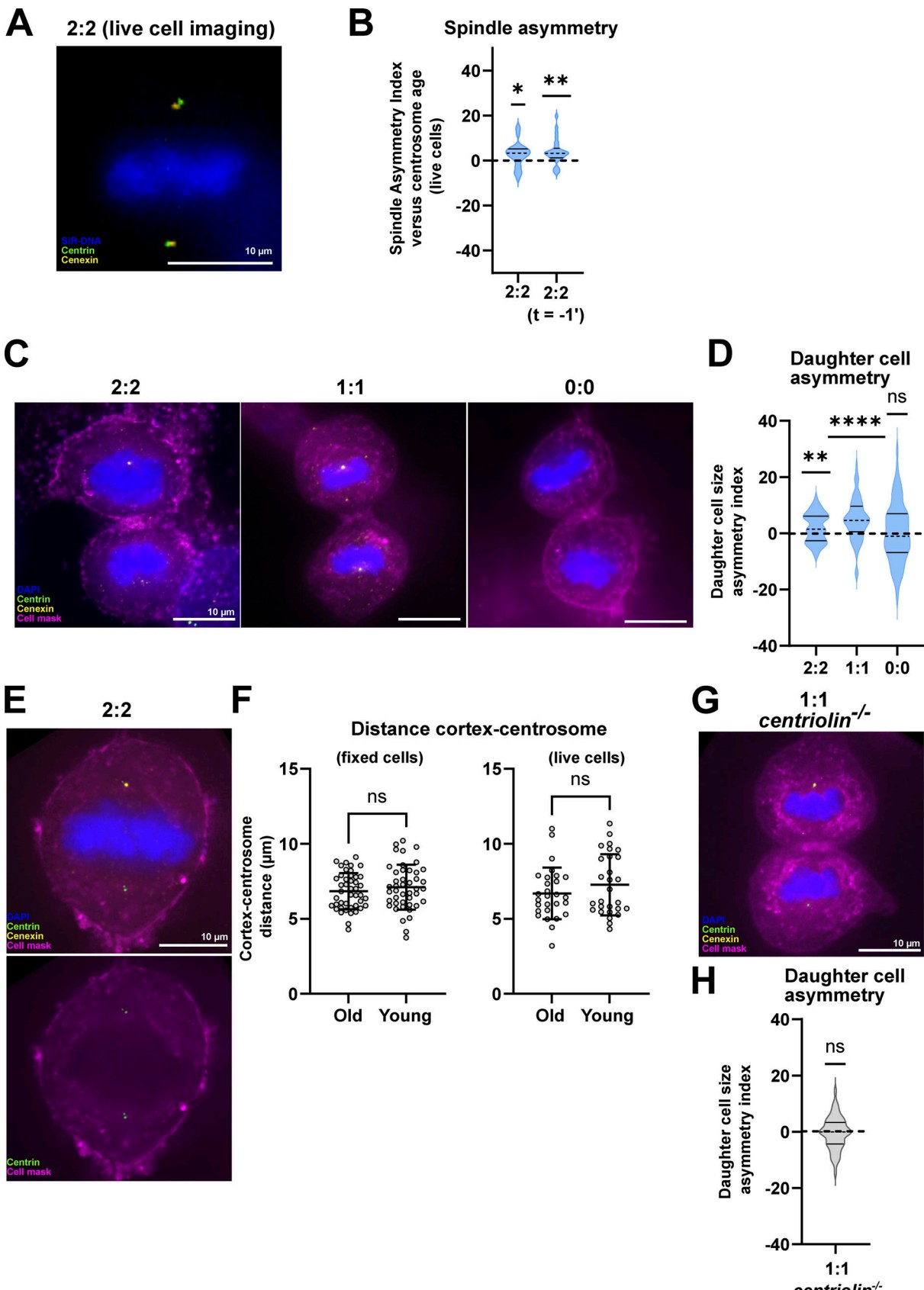

Figure 2.   **Centrosome-age-dependent asymmetric spindles' asymmetric daughter cell size. (A)** Image of a live 2:2 RPE1 GFP-centrin1 mScarlet-cenexin metaphase cell. **(B)** Quantification of the SAI versus centrosome age in live cells. On average, the mean SAI is 2.7 ± 5.5%, *n* = 28 cells, P = 0.0145. At *t* = −1 min before anaphase onset, the mean SAI is 3.8 ± 5.6%, *n* = 28 cells, P = 0.0012 in one-sample Wilcoxon tests. **(C)** Immunofluorescence images of 2:2, 1:1, and 0:0

RPE1 GFP-centrin1 late telophase cells. Cells are stained with DAPI (blue), GFP-centrin1 (green), cenexin (yellow) antibody and the cell mask (magenta). **(D)** Quantification of the daughter cell area asymmetry index. For 2:2 cells = 2 ± 5.3%, $n$ = 65 cells, P = 0.0025; for 1:1 cells = 4.7 ± 7.5%, $n$ = 58 cells, P <0.0001, and for 0:0 cells = 0.25 ± 10.5%, $n$ = 70 cells, P = 0.8370 in one-sample $t$ tests. **(E)** Immunofluorescence images of 2:2 and 1:1 RPE1 GFP-Centrin1 metaphase cells stained with DAPI (blue), GFP-centrin1 (green), cenexin antibody (yellow) and the cell mask (magenta). **(F)** Quantification of the cortex-centrosome distances in fixed and live cells. Mean distances of 6.8 ± 1.2 μm and 7.1 ± 1.5 μm for the old and young centrosomes, respectively (44 cells, P = 0.2135, ns in paired $t$ test) in fixed cells and means of 6.7 ± 1.7 μm and 7.3 ± 2 μm for the old and young centrosomes, respectively (28 cells, P = 0.4491, ns in paired $t$ test) in live cells 1 min before anaphase onset. **(G)** Immunofluorescence image of a 1:1 $centriolin^{-/-}$ GFP-centrin1 RPE1 late telophase cell stained with DAPI (blue), GFP-centrin1 (green), cenexin antibody (yellow) and the cell mask (magenta). **(H)** Quantification of the daughter cell area asymmetry index of 1:1 $centriolin^{-/-}$ GFP-centrin1 RPE1 cells = −0.4 ± 5.8%, $n$ = 54 cells, P = 0.6246, in one-sample $t$ test. All scale bars = 10 μm. ns = not significant; P ≤ 0.05 = *; P ≤ 0.01 = **; P ≤ 0.0001 = ****.

was statistically significant (P < 0.01) and the slope of the linear regression exceeded 0.15.

The list of investigated proteins included centrobin, a daughter-centriole-specific protein (Zou et al., 2005; Jeffery et al., 2010); the mitotic kinase Plk1 and Aurora-A (specifically the activated phosphorylated form of Aurora-A) and their centrosomal activator CEP192, which have been implicated in the regulation of spindle microtubule dynamics (Joukov et al., 2014;

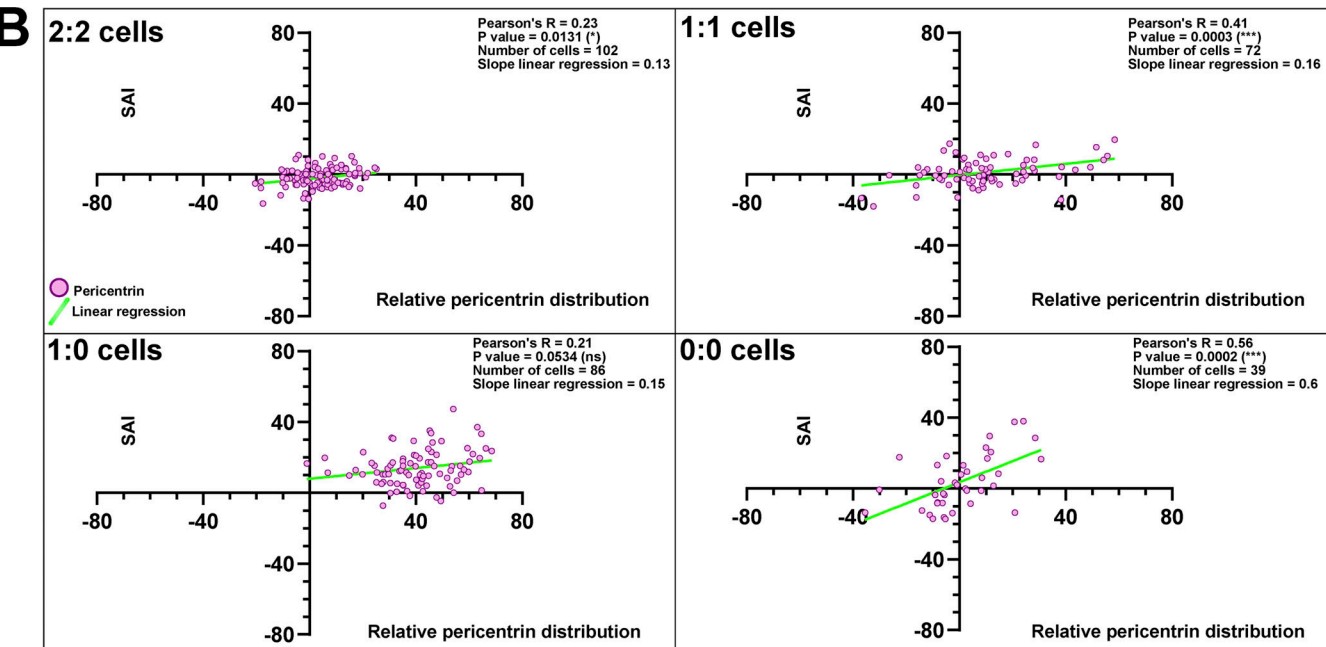

**Figure 3.** **The spindle pole abundance of proteins regulating microtubule nucleation scales with half-spindle size. (A)** Immunofluorescence images of 2:2, 1:1, 1:0, and 0:0 RPE1 GFP-centrin1 metaphase cells stained with DAPI (blue), GFP-centrin1 (green), and pericentrin antibody (magenta). Scale bars = 10 μm. **(B)** Correlation plots between the relative pericentrin distribution (x axis) and the SAI (y axis) for 2:2, 1:1, 1:0, and 0:0 cells. Dots represent single-cell values. The light green line indicates the slope of the linear regression. For each plot, the Pearson correlation coefficient, its associated P value, the number of cells analyzed, and the slope of the linear regression are indicated. The data for all the proteins can be found in Table 1.

Table 1. **Identification of potential drivers of spindle (a)symmetry in RPE1 cells**

| RPE1 cells[a] | | | | |
|---|---|---|---|---|
| Proteins/Conditions | 2:2 cells | 1:1 cells | 1:0 cells | 0:0 cells |
| Centrobin | −0.45[b] | −0.03 | | |
| | ****[c] | ns | | |
| | 73[d] | 99 | | |
| | −0.17[e] | −0.003 | | |
| γ-tubulin | 0.17 | 0.4 | 0.48 | 0.54 |
| | ns | *** | **** | **** |
| | 89 | 72 | 92 | 59 |
| | 0.08 | 0.21 | 0.68 | 0.33 |
| CEP192 | 0.22 | 0.27 | 0.23 | |
| | ns | ns | ns | |
| | 122 | 63 | 49 | |
| | 0.09 | 0.19 | 0.12 | |
| Cdk5Rap2 | 0.22 | 0.48 | 0.33 | 0.6 |
| | * | **** | ** | *** |
| | 102 | 72 | 86 | 34 |
| | 0.1 | 0.19 | 0.17 | 0.43 |
| Pericentrin | 0.25 | 0.41 | 0.21 | 0.56 |
| | * | *** | ns | *** |
| | 102 | 72 | 86 | 39 |
| | 0.14 | 0.16 | 0.15 | 0.6 |
| TACC3 | −0.12 | 0.25 | 0.5 | 0.8 |
| | ns | ns | ** | **** |
| | 47 | 38 | 40 | 18 |
| | −0.04 | 0.14 | 0.3 | 0.73 |
| ch-TOG | 0.22 | 0.36 | 0.49 | 0.24 |
| | * | *** | *** | ns |
| | 90 | 88 | 46 | 18 |
| | 0.20 | 0.35 | 0.46 | 0.7 |
| Kif2A | 0.1 | 0.31 | 0.33 | 0.62 |
| | ns | ** | ns | ** |
| | 56 | 82 | 52 | 19 |
| | 0.06 | 0.13 | 0.13 | 0.37 |
| TPX2 | 0.17 | 0.58 | 0.34 | 0.68 |
| | ns | **** | ns | ** |
| | 42 | 79 | 52 | 15 |
| | 0.12 | 0.41 | 0.22 | 0.82 |
| Katanin | 0.29 | 0.52 | 0.27 | 0.66 |
| | ns | **** | ns | ** |
| | 58 | 57 | 68 | 16 |
| | 0.15 | 0.19 | 0.13 | 0.27 |
| Plk1 | 0.08 | 0.03 | | |
| | ns | ns | | |
| | 81 | 48 | | |
| | 0.04 | 0.01 | | |

Table 1.  **Identification of potential drivers of spindle (a)symmetry in RPE1 cells** (*Continued*)

| Phospho-Aurora-A[f] | 0.01 | 0.14 | 0.29 | 0.46 |
|---|---|---|---|---|
| | ns | ns | ns | ns |
| | 73 | 96 | 71 | 21 |
| | 0.01 | 0.04 | 0.25 | 0.25 |
| MCAK | 0.41 | 0.23 | 0.29 | 0.32 |
| | ** | ns | ns | ns |
| | 57 | 59 | 55 | 10 |
| | 0.12 | 0.04 | 0.09 | 0.05 |
| EB1 | 0.22 | 0.47 | | |
| | ns | **** | | |
| | 76 | 64 | | |
| | 0.12 | 0.29 | | |

[a]Correlation between the relative abundance of different centriolar, centrosomal, and spindle pole proteins at the two poles of the mitotic spindle and the SAI, in metaphase RPE1 cells with different centriole numbers.
[b]Pearson's correlation coefficient between the relative protein distribution and the SAI.
[c]P value of the correlation. ns = not significant; $P \leq 0.05 = *$; $P \leq 0.01 = **$; $P \leq 0.001 = ***$; $P \leq 0.0001 = ****$.
[d]Number of cells.
[e]Slope of the linear regression.
[f]The active form of Aurora-A (anti phospho-threonine 288) was quantified.

Barr and Gergely, 2007; Asteriti et al., 2015); pericentrosomal proteins required for microtubule nucleation such as pericentrin, Cdk5Rap2, and γ-tubulin that have been implicated in spindle size regulation (Fong et al., 2008; Choi et al., 2010; Greenan et al., 2010; Ren and Weisblat, 2006; Watanabe et al., 2020); and microtubule dynamics regulators at spindle poles such as the microtubule depolymerases Kif2A and MCAK (Ganem and Compton, 2004; Jang et al., 2009; Domnitz et al., 2012), the microtubule-severing enzyme katanin (Loughlin et al., 2011; Huang et al., 2021; Guerreiro et al., 2021), as well as the MAPs TPX2 (Bird and Hyman, 2008; Sobajima et al., 2023), TACC3 (Cassimeris and Morabito, 2004; Gergely et al., 2003), ch-TOG (Barr and Gergely, 2008; Brouhard et al., 2008), and EB1 (Dema et al., 2022).

A first group of hits we identified in this analysis were pericentrin, Cdk5Rap2, and γ-tubulin, whose protein abundance distribution significantly correlated with the SAI in 1:1, 1:0, and 0:0 cells (Fig. 3 B, Fig. S2, and Table 1), meaning that at the single-cell level, the longer half-spindle also tended to display more of those proteins at the spindle pole. This suggested that pericentriolar proteins implicated in microtubule nucleation might control spindle (a)symmetry both in the presence or absence of centrosomes. Even in 2:2 cells, there was a visible correlation between the relative pericentrin and Cdk5Rap2 distribution and the SAI (Table 1). We next tested whether their distribution follows centrosome age and whether these proteins contribute to spindle (a)symmetry using 1:1 cells, which display a sharper spindle asymmetry than 2:2 cells (Fig. 1 B). We found that all three proteins were enriched on the spindle pole associated with the old centrosome (mean of 14 ± 13.8% for pericentrin, 13.3 ± 15.7% for Cdk5Rap2, and 4.4 ± 6.2% for γ-tubulin; Fig. 4, A and B). Loss of centriolin (*centriolin*$^{-/-}$ cells) diminished the asymmetric distribution of pericentrin (mean of 6.1 ± 13.1%)

and abolished the γ-tubulin asymmetry (mean of 0.7 ± 9.4%; Fig. 4 D; Cdk5Rap2 levels could not be tested due to antibody incompatibilities). At the functional level, depletion of either protein abolished the asymmetry of 1:1 cells (SAI means of 0.3 ± 6.1% n = 54 cells for *sipericentrin*, −0.3 ± 7% n = 51 cells for *siCdk5Rap2*, and 0.9 ± 9.5% n = 60 cells for *siγ-tubulin*; Fig. 4, A–C; validation of all siRNA treatments in Fig. S3, A–C). In contrast after depletion of centrobin, a protein not known to control spindle size, spindles were still asymmetric (SAI means of 3.4 ± 7.7%, n = 23 cells in *siCtrl* versus 4.5 ± 6.1%, n = 33 cells in *sicentrobin*; Fig. 4, E and F; and Fig. S3 D). Moreover, we noted that depletion of pericentrin or Cdk5Rap2 slightly increased spindle asymmetry in 1:0 cells (γ-tubulin could not be tested, as its depletion disrupted spindle formation in 1:0 cells; Fig. 4, G and I; note that in the case of pericentrin and Cdk5Rap2 depletion, there might be some remaining protein despite an effective depletion, since their knockout prevents bipolar spindle assembly in 1:0 cells [Watanabe et al., 2020]). This implied that proteins implicated in microtubule nucleation at centrosomes are specifically required for centrosome-age-dependent spindle asymmetry. Consistent with this hypothesis, the capacity to nucleate microtubules at centrosomes correlated with their age. Indeed, when microtubules were depolymerized by a 1-h cold treatment, both centrosomes contained the same quantity of residual microtubules; in contrast, when microtubules were allowed re-nucleate for 15 s in warm medium, more tubulin could be found at the old centrosome (Fig. 4, J and K). In contrast, when we depleted Cdk5Rap2, we found that rapid centrosome-dependent microtubule nucleation was abolished on both spindle poles, as the first microtubules now emerged from kinetochores after 1 min (Fig. 4 L). This suggested that centrosome-age-dependent spindle asymmetry depends on the rapid centrosome-driven microtubule nucleation and that spindles are symmetric

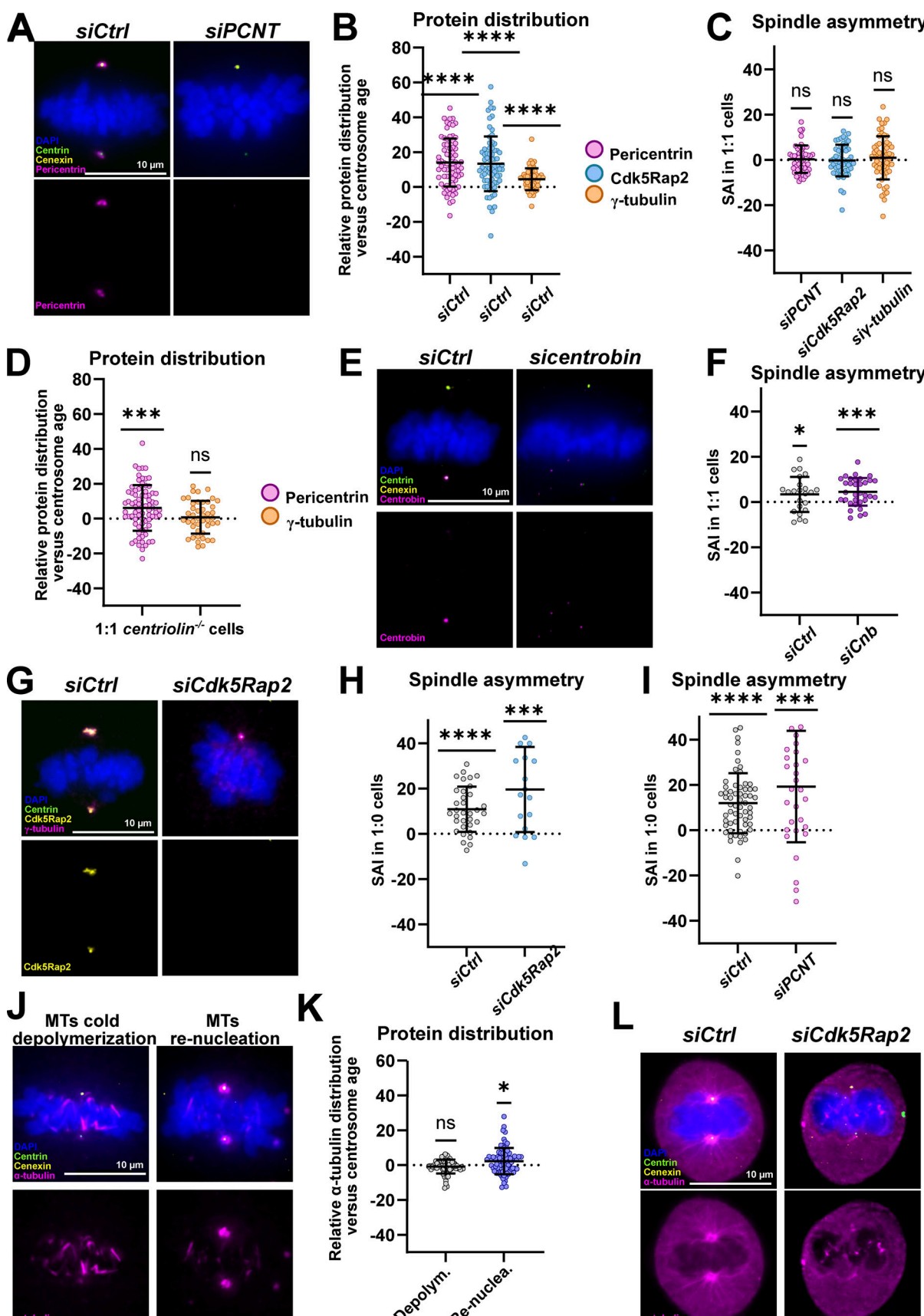

Figure 4. **Proteins regulating microtubule nucleation break spindle symmetry. (A)** Immunofluorescence images of *siCtrl* and *siPCNT*-treated RPE1 GFP-centrin1 1:1 cells stained with DAPI (blue), centrin1-GFP (green), cenexin (yellow), and pericentrin (magenta) antibodies. **(B)** Quantification of the relative

pericentrin, Cdk5Rap2, and γ-tubulin asymmetry index versus centrosome age (using cenexin as old centrosome marker) in control-depleted 1:1 RPE1 GFP-centrin1 cells. Means of 14 ± 13.8% $n$ = 73 cells, 13.3 ± 15.7% $n$ = 72 cells, and 4.4 ± 6.2% $n$ = 54 cells for pericentrin, Cdk5Rap2, and γ-tubulin, respectively. P < 0.0001 in one-sample $t$ tests for pericentrin and Cdk5Rap2, and one-sample Wilcoxon test for γ-tubulin. **(C)** Quantification of the SAI after *siPCNT*, *siCdk5Rap2*, and *siγ-tubulin* treatment in 1:1 RPE1 GFP-centrin1 cells. Means of respectively 0.27 ± 6.06% $n$ = 54 cells, −0.29 ± 6.96% $n$ = 50 cells, and 0.92 ± 9.52% $n$ = 60 cells for pericentrin, Cdk5Rap2, and γ-tubulin depletion, respectively. P values of respectively 0.8246, 0.8902, and 0.3277 for *siPCNT*, *siCdk5Rap2*, and *siγ-tubulin* in one-sample $t$ tests. **(D)** Quantification of the relative protein distribution of pericentrin and γ-tubulin versus centrosome age in 1:1 *centriolin⁻/⁻* RPE1 cells. Means of 6.1 ± 13.05% $n$ = 77 cells and 0.73 ± 9.42% $n$ = 44 cells for pericentrin, and γ-tubulin, respectively. P < 0.0001 and 0.6059 for pericentrin and γ-tubulin, respectively, in one-sample $t$ tests. **(E)** Immunofluorescence images of *siCtrl* and *sicentrobin*-treated 1:1 RPE1 GFP-centrin1 cells stained with DAPI (blue), GFP-centrin1 (green), cenexin (yellow), and centrobin (magenta) antibodies. Note that these images are also shown in Fig. S3 D. **(F)** Quantification of the SAI in *siCtrl and sicentrobin*-treated in RPE1 GFP-centrin1 cells. SAI means 3.4 ± 7.74%, $n$ = 23 cells, P = 0.0493, in *siCtrl* cells and 4.5 ± 6.1%, $n$ = 33 cells, P = 0.0002, in *sicentrobin* cells. One-sample $t$ tests were used for statistical analyses. **(G)** Immunofluorescence images of *siCtrl* and *siCdk5Rap2*-treated 1:0 RPE1 GFP-centrin1 cells, stained with DAPI (blue), GFP-centrin1 (green), Cdk5Rap2 (yellow) and γ-tubulin (magenta) antibodies. **(H and I)** Quantification of the SAI of (H) *siCtrl* and *siCdk5Rap2*- or (I) *siCtrl* and *siPCNT*-treated 1:0 RPE1 GFP-centrin1 cells. SAI means of (H) 10.8 ± 10%, $n$ = 24 cells, P < 0.0001 in *siCtrl* cells and 19.6 ± 18.8%, $n$ = 18 cells, P = 0.0007, in *siCdk5Rap2* cells. One-sample Wilcoxon tests were used for statistical analyses. SAI means of (I) 11.9 ± 13.2%, $n$ = 61 cells, P < 0.0001, in *siCtrl* cells and 19.2 ± 24.6%, $n$ = 32 cells, P < 0.0001 in *siCdk5Rap2* cells. One-sample $t$ tests were used for statistical analyses. **(J)** Immunofluorescence images of 1:1 RPE1 GFP-centrin1 cells after 1 h of cold depolymerization (left) and 15 s after re-nucleation (right). Cells are stained with DAPI (blue), GFP-centrin1 (green), cenexin (yellow) and α-tubulin (magenta) antibodies. **(K)** Quantification of the relative α-tubulin distribution at centrosomes versus centrosome age. After the cold depolymerization, mean of −0.8 ± 4%, $n$ = 75 cells, P = 0.2044, and after the re-nucleation, mean of 2.3 ± 7.5%, $n$ = 76 cells, P = 0.0256. One-sample Wilcoxon tests were used for statistical analyses. **(L)** Immunofluorescence images of 1:1 RPE1 GFP-centrin1 cells after *siCtrl* and *siCdk5Rap2* treatments and after 1 h of cold depolymerization (left) and 1 m after re-nucleation (right). Cells are stained with DAPI (blue), GFP-centrin1 (green), cenexin (yellow), and α-tubulin (magenta) antibodies. Note the microtubule nucleation in the chromatin region in siCdk5Rap2-treated cells. All scale bars = 10 μm. ns = not significant; P ≤ 0.05 = *; P ≤ 0.001 = ***; P ≤ 0.0001 = ****.

when alternative nucleation sources dominate, for example, kinetochores (Maiato et al., 2004; Wu et al., 2023).

### Centrosome age breaks spindle size symmetry via a pericentrin–TPX2/ch-TOG axis

Our analysis revealed a second group of potential hits, which included the microtubule depolymerase Kif2A, the microtubule severing enzyme katanin, and the regulators of microtubule dynamics TPX2, ch-TOG, and TACC3. Kif2A, katanin, and TPX2 were enriched on the spindle pole of the longer half-spindle in 1:1 and 0:0 cells, ch-TOG was enriched on the pole of the longer spindle in all conditions, while TACC3 only showed such a correlation in 1:0 and 0:0 cells (Table 1). This suggested that TACC3 may only regulate spindle symmetry in the context of centrosome-free poles and may not depend on centrosome age. This observation is consistent with its localization at spindle poles in human oocytes, as a component of the centrosome-free oocyte microtubule organizing centers (Wu et al., 2022).

The higher abundance of Kif2A and katanin on the longer half-spindle appeared at first counter-intuitive, as one would expect such enzymes to reduce half-spindle length. Nevertheless, this profile could be explained by the known ability of TPX2 to form complexes with Kif2A/CLASP1 and with katanin/WDR62 on the mitotic spindle (Fu et al., 2015; Huang et al., 2021). We reasoned that TPX2, which regulates spindle size on its own (Bird and Hyman, 2008; Fu et al., 2015; Helmke and Heald, 2014; Sobajima et al., 2023), could be a critical factor regulating spindle asymmetry. We therefore quantified the SAI in both 1:1 and 1:0 cells after depleting TPX2 by siRNA (validation, see Fig. S4 A; to visualize the spindle poles in *siTPX2* 1:0 cells we used antibodies against the minus-end binding protein NuMa). Our measurements indicated a rescue of spindle symmetry in 1:1 cells compared with a control depletion (2.5 ± 5.9%, $n$ = 74 cells in *siControl* cells and 1.1 ± 9.2%, $n$ = 102 cells in *siTPX2* cells) and a significant decrease of the spindle asymmetry in 1:0 cells (Fig. 5, A and B; and Fig. S4, B and C). These results imply that TPX2 in

contrast to the pericentrin/Cdk5Rap2/γ-tubulin module contributes to the spindle asymmetry in both 1:1 and 1:0 cells. Consistent with a role in centrosome-age-dependent spindle asymmetry, TPX2 itself localized in an asymmetric manner in WT RPE1 2:2 cells, but not in *centriolin⁻/⁻* RPE1 2:2 cells, indicating that its localization is influenced by centrosome age (means of 3 ± 6.5%, $n$ = 36 cells in WT cells versus 0.7 ± 5%, $n$ = 55 cells in *centriolin⁻/⁻* cells; Fig. 5 C).

Given that TPX2 and the pericentriolar proteins pericentrin/Cdk5Rap2/γ-tubulin are both required for centrosome-age-dependent spindle asymmetry, we used siRNA depletions to test whether they affect each other's recruitment at spindle poles (Fig. 5 D), as to our knowledge no interaction between TPX2 and pericentrin or Cdk5Rap2 has been reported so far. Quantitative immunofluorescence indicated that pericentrin depletion reduced TPX2 intensity at spindle poles by 50%, while TPX2 depletion did not significantly change pericentrin abundance (Fig. 5, D and E; and Fig. S4 D). We conclude that pericentrin contributes to TPX2 recruitment at spindle poles and that it acts upstream of TPX2 in the centrosome-age-dependent breaking of spindle symmetry.

The other potential driver of spindle asymmetry was ch-TOG (Table 1), a conserved protein that promotes microtubule polymerization and which regulates spindle size in various species (Gergely et al., 2003; Reber et al., 2013; Barr and Bakal, 2015). Like TPX2, ch-TOG was also more abundant on old centrosomes (mean of 2.9 ± 6.5%, $n$ = 34 cells; Fig. 5, E and F; and Fig. S4 F), and its depletion led to the formation of symmetric spindles in 1:1 RPE1 cells (means of 2.8 ± 7.4% in *siControl* cells versus 1.5 ± 8.5% in *sich-TOG* cells; Fig. 5 G and Fig. S4, E and F). This similarity persisted in terms of recruitment dependencies, as pericentrin depletion also reduced the recruitment of ch-TOG at spindle poles (Fig. 5 H; the reciprocal ch-TOG depletion had a minor effect on pericentrin levels, Fig. S4 G). This implied that centrosome age promotes an asymmetric microtubule nucleation not only via the PCM but also more downstream via the

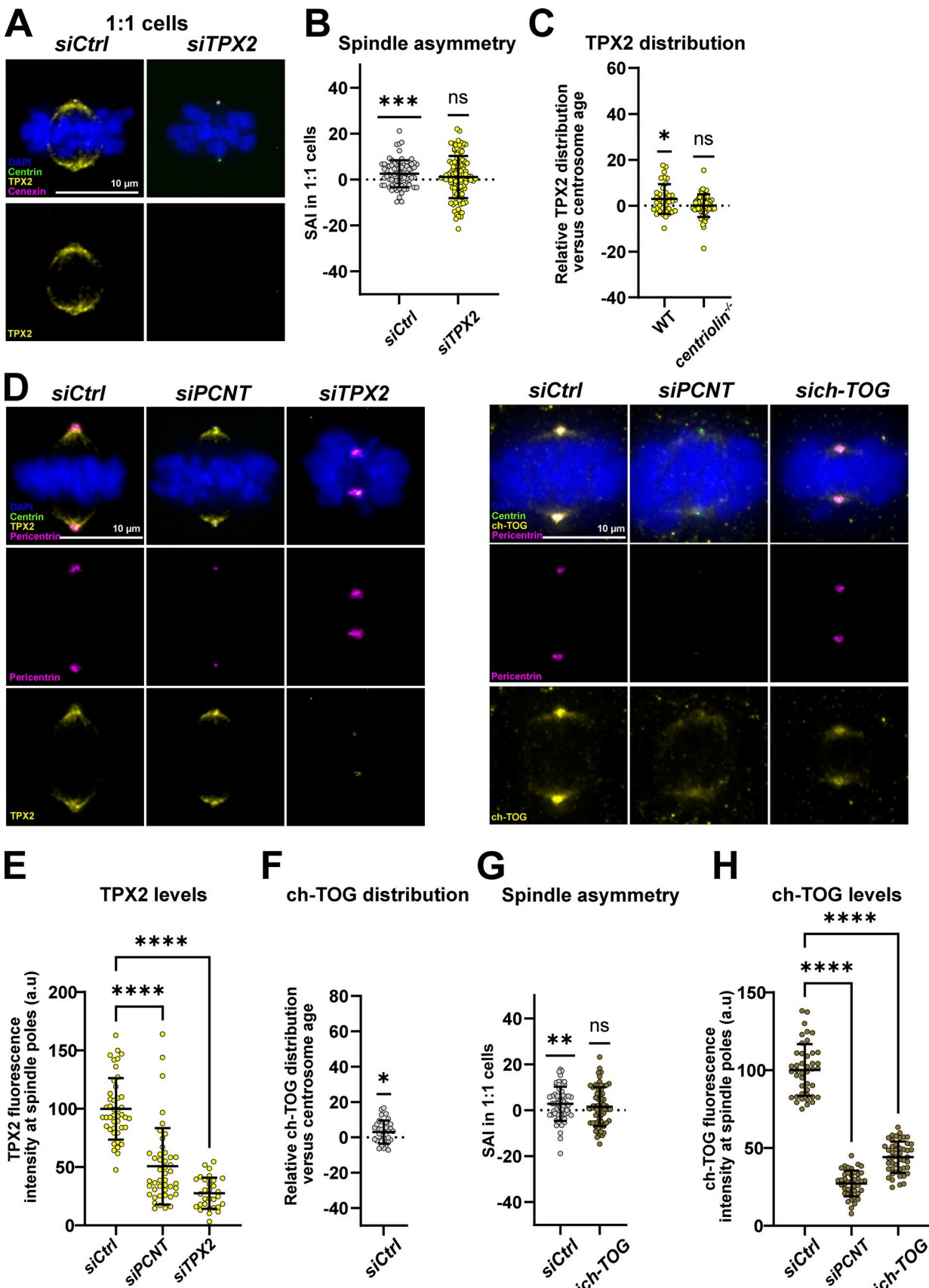

Figure 5. **A pericentrin–TPX2 axis drives the formation of asymmetric spindles. (A)** Immunofluorescence images of *siCtrl* and *siTPX2*-treated 1:1 RPE1 GFP-centrin1 cells stained with DAPI (blue), GFP-centrin1 (green), cenexin (magenta), and TPX2 (yellow) antibodies. **(B)** Quantification of the SAI of *siCtrl*

*and siTPX2*-treated 1:1 RPE1 GFP-centrin1 cells. SAI means of 2.5 ± 5.9%, *n* = 74 cells, P = 0.0005, in *siCtrl* cells and 1.1 ± 9.2%, *n* = 102 cells, P = 0.2454, in *siTPX2* cells. One-sample *t* tests were used for statistical analyses. **(C)** Quantification of the TPX2 asymmetry versus centrosome age in 2:2 RPE1 GFP-centrin1 WT and *centriolin⁻/⁻* cells. Relative TPX2 distribution means of 2.9 ± 6.5%, *n* = 36 cells, P = 0.0210, in WT cells and 0.1 ± 5%, *n* = 55 cells, P = 0.5917, in *centriolin⁻/⁻* cells. One-sample *t* tests were used for statistical analyses. **(D)** Left panels: immunofluorescence images of *siCtrl, siPCNT*, and *siTPX2*-treated 2:2 RPE1 GFP-centrin1 cells stained with DAPI (blue), GFP-centrin1 (green), TPX2 (yellow), and pericentrin (magenta) antibodies; right panels: immunofluorescence images of *siCtrl, siPCNT*, and *sich-TOG*–treated 2:2 RPE1 GFP-centrin1 cells stained with DAPI (blue), GFP-centrin1 (green), ch-TOG (yellow), and pericentrin (magenta) antibodies. Scale bars = 10 µm. **(E)** Quantification of TPX2 fluorescence intensity at spindle poles. Means of 100 ± 26.3%, *n* = 44 cells in *siCtrl* cells, 50.1 ± 32.7%, *n* = 45 cells, P < 0.0001 in *siPCNT* cells, and 27.6 ± 13.4%, *n* = 30 cells, P < 0.0001 in *siTPX2*-treated cells. One-way Kruskal–Wallis with Dunnett's multiple comparisons tests. **(F)** Quantification of the relative protein distribution of ch-TOG in *siCtrl*-treated 1:1 RPE1 GFP-centrin1 mScarlet-cenexin cells. Mean of 2.9 ± 6.6%, *n* = 34 cells, P = 0.0249, one-sample Wilcoxon-test. **(G)** Quantification of the SAI of *siCtrl* and *sich-TOG*–treated 1:1 RPE1 Centrin1-GFP mScarlet-cenexin cells. SAI means of 2.8 ± 7.4%, *n* = 59 cells, P = 0.0057, in *siCtrl* cells and 1.5 ± 8.4%, *n* = 63 cells, P = 0.1587, in *sich-TOG* cells. One-sample *t* tests were used for statistical analyses. **(H)** Quantification of ch-TOG fluorescence intensity at spindle poles. Means of 100 ± 16.6%, *n* = 43 cells in *siCtrl* cells, 27.2 ± 8.3%, *n* = 41 cells, P < 0.0001 in *siPCNT* cells, and 44.2 ± 10%, *n* = 47 cells, P < 0.0001 in *sich-TOG*–treated cells. One-way Kruskal–Wallis with Dunnett's multiple comparisons tests. ns = not significant; P ≤ 0.05 = *; P ≤ 0.01 = **; P ≤ 0.001 = ***; P ≤ 0.0001 = ****.

recruitment of key regulators of microtubule nucleation and polymerization, such as TPX2 and ch-TOG.

### The asymmetric localization of microtubule nucleators depends on Plk1

The PCM, responsible for microtubule nucleation, expands strongly at mitotic entry, leading to centrosome maturation (Palazzo et al., 2000). Hence, we investigated whether the centrosome-age-dependent distribution of pericentrin and γ-tubulin already appears in G2 and prophase (Fig. 6, A and C). We found that pericentrin is already more abundant at old centrosomes in G2 (means of 9.2 ± 12.9%, *n* = 36 cells in G2 and 9.5 ± 9.3%, *n* = 29 cells in prophase), whereas γ-tubulin distribution becomes asymmetric only upon mitotic entry (means of 3.6 ± 16.8%, *n* = 50 cells in G2 and 7.3 ± 10.8%, *n* = 53 cells in prophase; Fig. 6, B and D). This suggested that this asymmetry is initiated with pericentrin in G2 and that it is expanded to γ-tubulin as cells enter mitosis.

How about upstream regulators of the PCM? The abundance of the main PCM regulator, the mitotic kinase Plk1 (Lee and Rhee, 2011; Lane and Nigg, 1996; Conduit et al., 2014), did not correlate with spindle size asymmetry in our initial screen (Table 1). We nevertheless tested whether Plk1 was enriched on old centrosomes and found that in prophase (10.9 ± 20.8%; the earliest time point Plk1 levels could be reliably quantified) and metaphase (6.6 ± 10.8 %; Fig. 6, E and F) Plk1 was enriched on the old centrosome. To test whether this Plk1 asymmetry controls the asymmetric distribution of microtubule nucleators and spindle size asymmetry, we partially inhibited Plk1 (overnight 50 nM treatment) with the chemical inhibitor BI2536 (Lénárt et al., 2007). This led to monopolar spindles in 90% of cells and reduced the γ-tubulin signal, reflecting a strong Plk1 inhibition. Nevertheless, it allowed for the formation of short bipolar spindles in 10% of the cases (means of 7.9 ± 1.3 µm in BI2536- versus 11.2 ± 1.3 µm in DSMO-treated cells; Fig. 6, G and H). In those spindles, γ-tubulin and pericentrin distribution were symmetric (γ-tubulin = –0.6 ± 4.8% in BI2536-treated cells versus 4.0 ± 5.5% in DMSO-treated cells; pericentrin = –3.3 ± 12.4% in BI2536-treated cells versus 4.5 ± 9.6% in DMSO-treated cells; Fig. 6, I and J). It also resulted in 1:1 cells with symmetric spindle sizes (SAI = 0.4 ± 10.0%, in BI2536- versus 2.4 ± 4.6% in DMSO-treated cells; Fig. 6 K). We conclude that the asymmetric localization of microtubule nucleators and spindle (a)symmetry depend on Plk1 activity.

### Centrosome age controls spindle size symmetry and polar chromosome asymmetry via cenexin-bound Plk1

What is the origin of the Plk1 asymmetry? Previous studies indicated that a subpool of Plk1 is recruited to cenexin on S796 that has been phosphorylated by CDK1 (Soung et al., 2009). To test whether this is the symmetry-breaking event, we introduced in RPE1 *cenexin⁻/⁻* cells either WT mScarlet-cenexin or S796A mScarlet-cenexin. Both exogenous cenexin versions localized predominantly to one centrosome (presumably the older centrosome), and none of the two mutants affected mitotic timing (defined as the time between nuclear envelope breakdown and anaphase onset, Fig. 7 A and Fig. S5). Their effect on spindle size symmetry, however, was different. While 1:1 WT mScarlet-cenexin had asymmetric spindles (SAI mean of 2 ± 6.5%, *n* = 73 cells, P = 0.0044; Fig. 7, A and B), 1:1 RPE1 mScarlet-cenexin S796A cells harbored symmetric spindles (SAI mean of 0.4 ± 7.5%, *n* = 75 cells, P = 0.7096; Fig. 7, A and B). To confirm that the symmetric spindles in RPE1 mScarlet-cenexin S796A cells resulted from a non-biased distribution of Plk1, we quantified the relative Plk1 distribution with respect to centrosome age. We found that RPE1 WT mScarlet-cenexin cells displayed an asymmetric Plk1 distribution (means of 9.1 ± 11.8% [prophase] and 8.4 ± 11% [metaphase]; Fig. 7 D), while mScarlet-cenexin S796A imposed a symmetric distribution of Plk1 (means of 0.9 ± 13.9% in prophase and 1 ± 14.9% in metaphase; Fig. 7, C and E). Consistently, γ-tubulin distribution was asymmetric in 1:1 WT mScarlet-cenexin cells (2.4 ± 7.6%), but symmetric in 1:1 mScarlet-cenexin S796A cells (1.2 ± 6.1%; Fig. 7, F and G). Interestingly, pericentrin was still more enriched at old centrosomes in 1:1 mScarlet-cenexin S796A cells (4.6 ± 10.6%; Fig. 7 H), but to a lower extent than 1:1 WT RPE1 cells (Fig. 4 B).

Finally, we asked whether the absence of the Plk1 binding site on cenexin also affects polar chromosome asymmetry, as one would predict based on previous studies showing that this asymmetry depends both on cenexin and Plk1 (Gasic et al., 2015; Colicino et al., 2019). We found that 1:1 WT mScarlet-cenexin cells but not 1:1 mScarlet-cenexin S796A cells displayed an asymmetric distribution of polar chromosomes (69.7 ± 15.2% [WT] versus 50.2 ± 8.4% [S796A] of polar chromosomes associated to the old centrosome; N = 267 and 235 polar chromosomes, *n* = 96 and 75 cells; Fig. 7, I and J). We conclude that the cenexin-bound pool of Plk1 breaks spindle symmetry both in

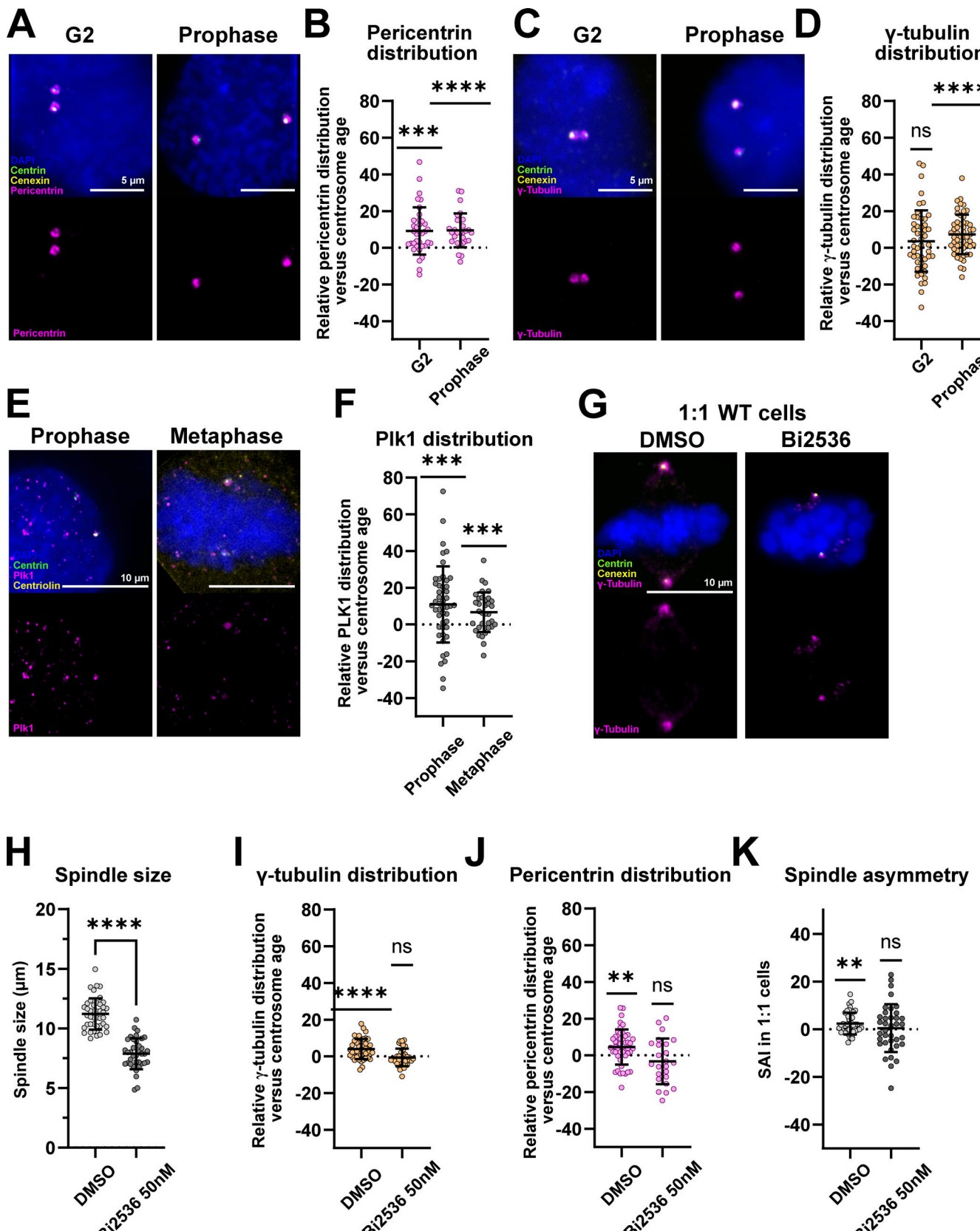

Figure 6. **The mitotic kinase Plk1 drives the enrichment of PCM proteins at old centrosomes. (A)** Immunofluorescence images of WT RPE1 GFP-centrin1 G2 and prophase cells stained with DAPI (blue), centrin1-GFP (green), cenexin (yellow), and pericentrin (magenta) antibodies. Scale bars = 5 µm. **(B)** Quantification of the relative pericentrin distribution at centrosomes versus centrosome age in G2 and prophase WT RPE1 GFP-centrin1 cells. Means of 9.2 ± 12.9%,

*n* = 36 cells, P = 0.0001 in G2 and 9.5 ± 9.3%, *n* = 29 cells, P < 0.0001 in prophase. One-sample *t* tests were used for statistical analyses. **(C)** Immunofluo-rescence images of WT RPE1 GFP-centrin1 G2 and prophase cells stained with DAPI (blue), centrin1-GFP (green), cenexin (yellow), and γ-tubulin (magenta) antibodies. Scale bars = 5 μm. **(D)** Quantification of the relative γ-tubulin distribution at centrosomes versus centrosome age in G2 and prophase WT RPE1 GFP-centrin1 cells. Means of 3.6 ± 16.8%, *n* = 50 cells, P = 0.1375 in G2 and 7.3 ± 10.8%, *n* = 53 cells, P < 0.0001 in prophase. One-sample *t* tests were used for statistical analyses. **(E)** Immunofluorescence images of WT RPE1 GFP-centrin1 prophase and metaphase cells stained with DAPI (blue), GFP-centrin1 (green), centriolin (yellow), and Plk1 (magenta) antibodies. Scale bars = 10 μm. **(F)** Quantification of the relative Plk1 distribution at centrosomes versus centrosome age in prophase and metaphase cells. Means of 10.9 ± 20.8%, *n* = 47 cells, P = 0.0008 for G2 cells and 6.6 ± 10.8%, *n* = 35 cells, P = 0.0009 for prophase cells. One-sample *t* tests were used for statistical analyses. **(G)** Immunofluorescence images of 1:1 RPE1 Centrin1-GFP cells treated with DMSO (control condition) or 50 nM Bi2536 (Plk1 inhibitor). Cells are stained with DAPI (blue), centrin1-GFP (green), cenexin (yellow), and γ-tubulin (magenta) antibodies. Scale bar = 10 μm. **(H)** Quantification of spindle size measurements in DMSO- and BI2536-treated cells. Means of 11.2 ± 1.3 μm, *n* = 43 cells, and 7.9 ± 1.3 μm, *n* = 37 cells in DMSO- and BI2536-treated cells, respectively, P value < 0.0001 in unpaired *t* test. **(I)** Quantification of the relative γ-tubulin distribution at centrosomes versus centrosome age in DMSO- and BI2536-treated RPE1 cells. Means of 4.0 ± 5.5%, *n* = 43 cells, P <0.0001 for and −0.6 ± 4.8%, *n* = 37 cells, P = 0.45 in DMSO- and BI2536-treated cells, respectively. One-sample *t* tests were used for statistical analyses. **(J)** Quantification of the relative pericentrin distribution at centrosomes versus centrosome age in DMSO- and BI2536-treated RPE1 cells. Means of 4.5 ± 9.6%, *n* = 46 cells, P = 0.0025 and −3.3 ± 12.4%, *n* = 24 cells, P = 0.2102 in DMSO- and BI2536-treated cells, respectively. One-sample *t* tests were used for statistical analyses. **(K)** SAI quantification in DMSO- and BI2536treated 1:1 RPE1 metaphase cells. SAI means of 2.4 ± 4.6%, *n* = 43 cells, P = 0.0012 and 0.4 ± 10.0%, *n* = 37 cells, P = 0.79 in DMSO- and BI2536-treated cells, respectively. One-sample *t* tests were used for statistical analyses. ns = not significant; P ≤ 0.01 = **; P ≤ 0.001 = ***; P ≤ 0.0001 = ****.

terms of half-spindle sizes and in terms of polar chromosome distribution.

### Daughter centrioles bind microtubule nucleators and dampen spindle symmetry

One of our surprising results was the fact that 1:1 cells lacking daughter centrioles displayed a stronger spindle asymmetry than WT 2:2 RPE1 cells. This implied that the presence of daughter centrioles dampens the symmetry breaking imposed by centrosome age. Daughter centrioles have so far considered to be immature; however, our and others work have shown that in conditions where daughter centrioles prematurely disengage during mitosis, such daughter centrioles can organize spindle poles and recruit γ-tubulin (Wilhelm et al., 2019; Logarinho et al., 2012). To test whether daughter centrioles contribute to the recruitment of microtubule nucleators at centrosomes, we compared by quantitative immunofluorescence pericentrin and γ-tubulin levels in WT (2:2) and 1:1 cells (Fig. 8 A). We found that in 1:1 metaphase cells, pericentrin (−32.6%) and γ-tubulin levels (−22.6%) decreased and that the pericentrin-positive PCM area at spindle poles decreased by a similar amount (−32.4%; Fig. 8, B–D). High-resolution Airyscan microscopy on WT metaphase RPE1 cells indicated that pericentrin formed not only rings around the grandmother and mother centrioles, as previously published (Mennella et al., 2012; Lawo et al., 2012) but that it also formed incomplete pericentrin rings around both daughter centrioles (Fig. 8 E). We conclude that daughter centrioles recruit PCM and may thus dampen the spindle asymmetry imposed by the grandmother centriole.

### Centrosome age also breaks spindle size symmetry in human fibroblast cells

To investigate if spindle size asymmetry is RPE1 cell dependent or a more general feature of human tissue culture cells, we carried out equivalent experiments with the fibroblastic, non-transformed human cell line BJ-hTERT (called BJ cells hereafter). We first quantified the SAI of WT metaphase BJ cells (2:2 cells), using cenexin as a marker for centrosome age, and found that BJ cells also had longer half-spindles associated with the old centrosome (3.1 ± 7.6%, *n* = 106 cells, P < 0.0001; Fig. 9, A and B).

This spindle size asymmetry correlated with an asymmetric daughter cell size in 1:1 cells, as daughter cells inheriting the grandmother centriole were statistically larger than the one inheriting the mother centriole (4.2 ± 7.8%, *n* = 47 cells, P = 0.0007; Fig. 9, C and D).

We next investigated the distribution of two main regulators identified in RPE1 cells, pericentrin, and TPX2. Quantitative immunofluorescence using cenexin as a marker for centrosome age indicated that pericentrin was asymmetrically distributed (8.7 ± 12.1%, *n* = 56 cells), while TPX2 displayed a symmetric distribution (1.7 ± 9.9%, *n* = 51 cells; Fig. 9 E). At the single-cell level, the abundance of pericentrin (Pearson correlation coefficient of 0.60, slope value of 0.38) but not the abundance of TPX2 was highly correlated with the SAI (Fig. 9 F). Consistent with our data showing an important role of PCM, pericentrin depletion led to symmetric spindles in 2:2 BJ cells (2.4 ± 6.8%, *n* = 43 cells in *siCtrl* cells, and 0.1 ± 5.6%, *n* = 31 cells in *siPCNT* cells; Fig. 7, G–I). We conclude that a centrosome-age-linked spindle symmetry is a general feature of human cell lines, present in epithelial and fibroblastic cells, and that spindle size asymmetry in both cases depends on the PCM.

## Discussion

It has long been thought that human generic tissue culture cells divide symmetrically with half-spindles of equal length. Here, we demonstrate that centrosome age breaks spindle size symmetry even in such cells resulting in the formation of daughter cells of unequal sizes (see model Fig. 9 J). This symmetry breaking is related but different to the previously identified polar chromosome asymmetry. At the molecular level, we show that in RPE1 cells centrosome-age-dependent spindle asymmetry depends on unequal microtubule nucleation, via an axis involving cenexin-bound Plk1 and the SDA, the recruitment of PCM, and finally the microtubule regulators TPX2 and ch-TOG (Fig. 9 J). This asymmetry is already present in G2 cells and is amplified at the PCM as centrosomes mature at mitotic entry. Finally, we find an unexpected role of daughter centrioles in dampening spindle asymmetry.

It is well known that old and young centrosomes display different behaviors in stem cell divisions (Chen and Yamashita,

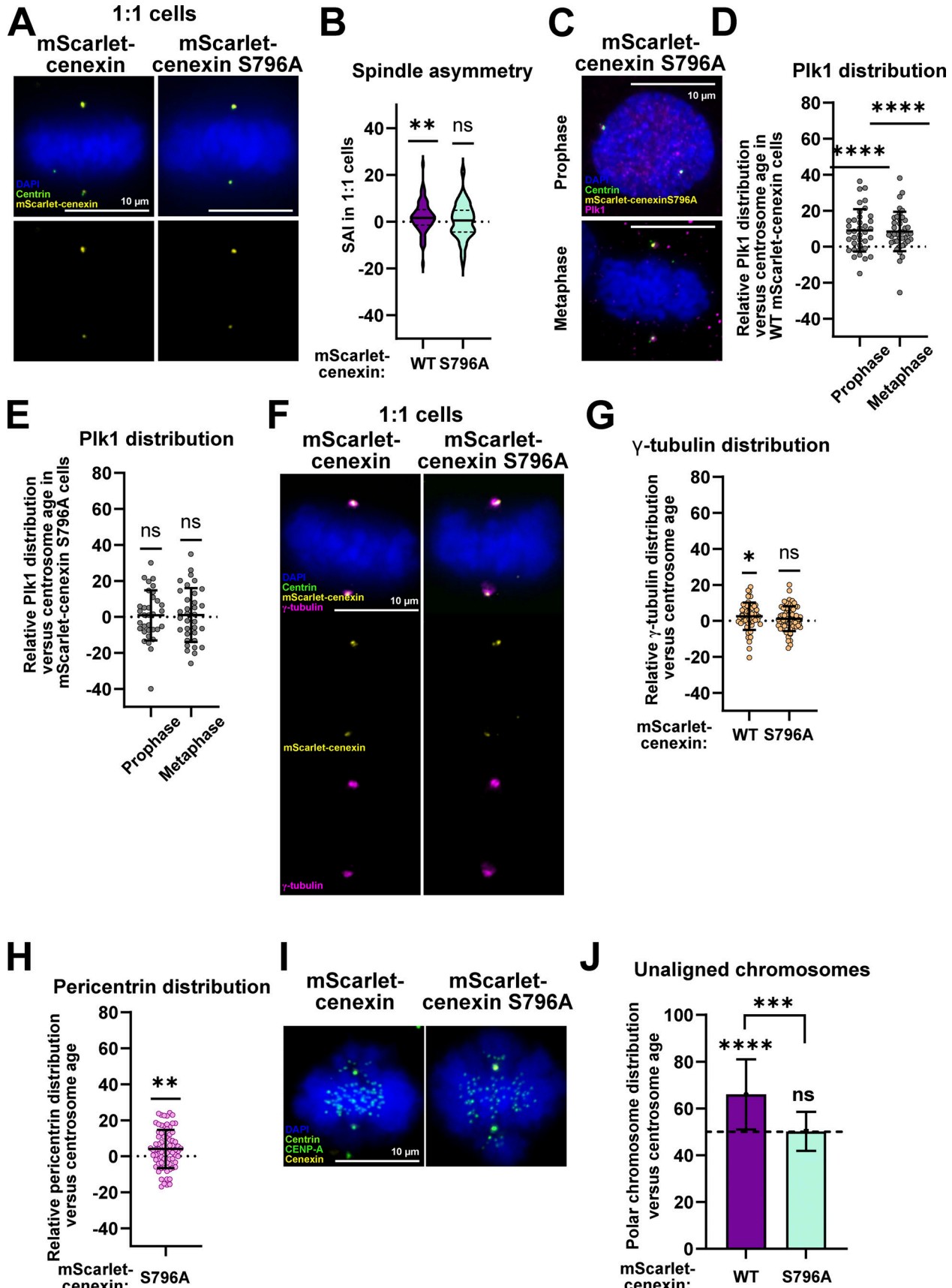

Figure 7. **The phosphorylation of cenexin S796 drives the formation of asymmetric spindles. (A)** Immunofluorescence images of 1:1 metaphase RPE1 GFP-centrin1 cells expressing either WT or S796A mScarlet-cenexin, stained with DAPI (blue), GFP-centrin1 (green), and mScarlet-cenexin (yellow). Scale

bars = 10 µm. **(B)** Quantification of the SAI in 1:1 RPE1 GFP-centrin1 cells expressing either WT or S796A mScarlet-cenexin. WT-cenexin SAI mean: 2.0 ± 6.5%, $n$ = 73 cells, P = 0.0044 in one-sample Wilcoxon test compared to a symmetric null hypothesis. S796A cenexin SAI mean: 0.35 ± 7.5%, $n$ = 75 cells, P = 0.7096 in one-sample $t$ test. **(C)** Immunofluorescence images of RPE1 GFP-centrin1 S796A mScarlet-cenexin prophase and metaphase cells stained with DAPI (blue), GFP-centrin1 (green), mScarlet-cenexin (yellow), and Plk1 (magenta) antibody. Scale bar = 10 µm. **(D)** Quantification of the relative Plk1 distribution at centrosomes versus centrosome age in prophase and metaphase in WT mScarlet-cenexin cells. Means of 9.1 ± 11.8%, $n$ = 37 cells, P < 0.0001 and 8.4 ± 11%, $n$ = 40 cells, P < 0.0001 in one-sample $t$ tests. **(E)** Quantification of the relative Plk1 distribution at centrosomes versus centrosome age in prophase and metaphase in S796A mScarlet-cenexin cells. Means of 0.9 ± 13.9%, $n$ = 33 cells, P = 0.7242 and 1 ± 14.9%, $n$ = 34 cells, P = 0.6970 in prophase and metaphase, respectively, in one-sample $t$ tests. **(F)** Immunofluorescence images of 1:1 metaphase RPE1 GFP-centrin1 cells expressing either WT mScarlet-cenexin or S796A mScarlet-cenexin, stained with DAPI (blue), GFP-centrin1 (green), mScarlet-cenexin (yellow), and γ-tubulin (magenta) antibody. Scale bar = 10 µm. **(G)** Quantification of the relative γ-tubulin distribution versus centrosome age in 1:1 RPE GFP-centrin1cells expressing either WT or S796A mScarlet-cenexin. In WT-cenexin cells relative distribution mean of 2.6 ± 7.6%, $n$ = 40 cells, P = 0.020, and in S796A cenexin cells relative distribution mean of 1.2 ± 6.1%, $n$ = 55 cells, P = 0.149 in one-sample Wilcoxon test. **(H)** Quantification of the relative pericentrin distribution versus centrosome age in 1:1 RPE GFP-centrin1 mScarlet-cenexin S796A cells. Relative distribution mean of 4.1 ± 10.6%, $n$ = 75 cells, P = 0.0014 in one-sample $t$ test. **(I)** Immunofluorescence images of 1:1 RPE GFP-centrin1 cells expressing either WT or S796A mScarlet-cenexin metaphase cells treated with nocodazole and stained with cenexin (yellow) and CENP-A (green) antibody, DAPI (blue), and GFP-centrin1 (green). **(J)** Quantification of the percentage of unaligned chromosomes distributed versus centrosome age. The dashed line represents a symmetric distribution (50%) of the polar chromosomes between the two poles. Means of 69.7 ± 15.2% (from 267 polar chromosomes in 96 cells) for mScarlet-cenexin cells, and 50.2 ± 8.4% (from 235 polar chromosomes in 75 cells) for mScarlet-cenexin S796A cells. P values of < 0.0001 and 1 for mScarlet-cenexin and mScarlet-cenexin S796A cells, respectively, in a binomial test against a random distribution. P value of 0.0003 (***) using a Fisher's exact test to compare the two conditions. ns = not significant; P ≤ 0.05 = *; P ≤ 0.01 = **; P ≤ 0.001 = ***; P ≤ 0.0001 = ****.

2021; Yamashita et al., 2007; Wang et al., 2009; Pereira et al., 2001; Januschke et al., 2011). In *D. melanogaster* neuroblasts, PCM components are enriched on the apical centrosome (Conduit and Raff, 2010), where they affect the relative half-spindle lengths (Fuse et al., 2003; Roubinet et al., 2017). Centrosome-age-dependent PCM-size asymmetry is also found in *D. melanogaster* syncytial embryos (Conduit et al., 2010). Here, we show that centrosome age also breaks the symmetry of half-spindle lengths in human cells thought to divide symmetrically, creating a bias toward longer half-spindles associated with the old centrosome. These results differ from our previous measurements in HeLa (Tan et al., 2015) and RPE1 cells (Dudka et al., 2019), which pointed to symmetric spindles. These earlier studies relied on the brightest GFP-centrin1 signal as a marker for the old centrosome, which we found to only partially coincide with the cenexin-positive old centrosome (see Fig. S5). We thus hypothesize that this inconsistency masked the subtle half-spindle length asymmetry. Microtubule nucleation-dependent spindle size asymmetry also biases the placement of the cell division plane, resulting in asymmetric daughter cell sizes. It did, however, not bias the placement of the whole spindle, which relies on a dynamic negative feedback loop, in which spindle pole-bound Plk1 displaces cortical dynein in a proximity-dependent manner (Kiyomitsu and Cheeseman, 2012). Since old centrosomes contain more Plk1, this most likely counterbalances any force imbalance due to additional microtubules at old poles reaching the cell cortex. Even though the difference in daughter size is subtle, previous studies showed that such differences can impact the length of the ensuing G1 phase and the probability of cell death (Kiyomitsu and Cheeseman, 2013). Altering the symmetry of the cell division could also bias the inheritance of organelles between the two daughter cells, as it has been seen in *D. melanogaster* male germline stem cell divisions (Chen et al., 2016).

Our data also show that the half-spindle size asymmetry is related, yet different to the previously identified polar chromosome asymmetry (Gasic et al., 2015; Colicino et al., 2019). Both depend on the cenexin-bound pool of Plk1, but only the spindle size asymmetry depends on centriolin (or its downstream factors) at SDA. This indicates that both asymmetries are independent of each other, as they can be uncoupled, and point to, at least in part, differing molecular mechanisms imposing these asymmetries. Whether the centrosome-age-dependent spindle size asymmetry has a direct function in generic cell divisions, or whether it is a passive consequence of a generally higher microtubule nucleation capacity at old centrosomes, remains unclear. The study of "symmetric" cells expressing only S796A cenexin and comparing them to isogenic "asymmetric" cells expressing WT cenexin will be key in future experiments aiming to study the potential long-term consequences of centrosome-age-dependent asymmetries.

Our correlation analysis of protein abundances and SAI at the single-cell level allowed us to identify potential drivers of spindle asymmetry that could then be (in)validated by siRNA treatments. These included proteins that had been previously found to display asymmetries, such as pericentrin, Cdk5Rap2, or centrobin (Tan et al., 2015; Gasic et al., 2015; Colicino et al., 2019; Le Roux-Bourdieu et al., 2022), but also other proteins such as γ-tubulin, TPX2, ch-TOG, or TACC3. Our functional data indicate that proteins implicated in microtubule nucleation and polymerization (TPX2/pericentrin/Cdk5Rap2/γ-tubulin/ch-TOG) are key for the centrosome-age-dependent spindle size asymmetry resulting in an asymmetric microtubule nucleation capacity at centrosomes. These data are consistent with previous studies in *C. elegans*, showing that spindle size scales with the size of the microtubule-nucleating PCM (Greenan et al., 2010), the fact that the leech *H. robusta* downregulates γ-tubulin on one spindle pole to create an asymmetric cell division (Ren and Weisblat, 2006), and the fact that asymmetric stem cell divisions in *D. melanogaster* are characterized by a strong difference in microtubule nucleation capacity between the old and young centrosome (Yamashita et al., 2007; Januschke et al., 2013). Our comparison of two different cell lines suggests that the principle of an asymmetric spindle size is conserved in human cells, but that the molecular regulation of this process can vary depending on the cellular background, as we find that TPX2 does not correlate with spindle asymmetry in BJ cells. Since spindle size asymmetry responds to the abundance of multiple centrosomal proteins, this might give cells the ability to modulate this

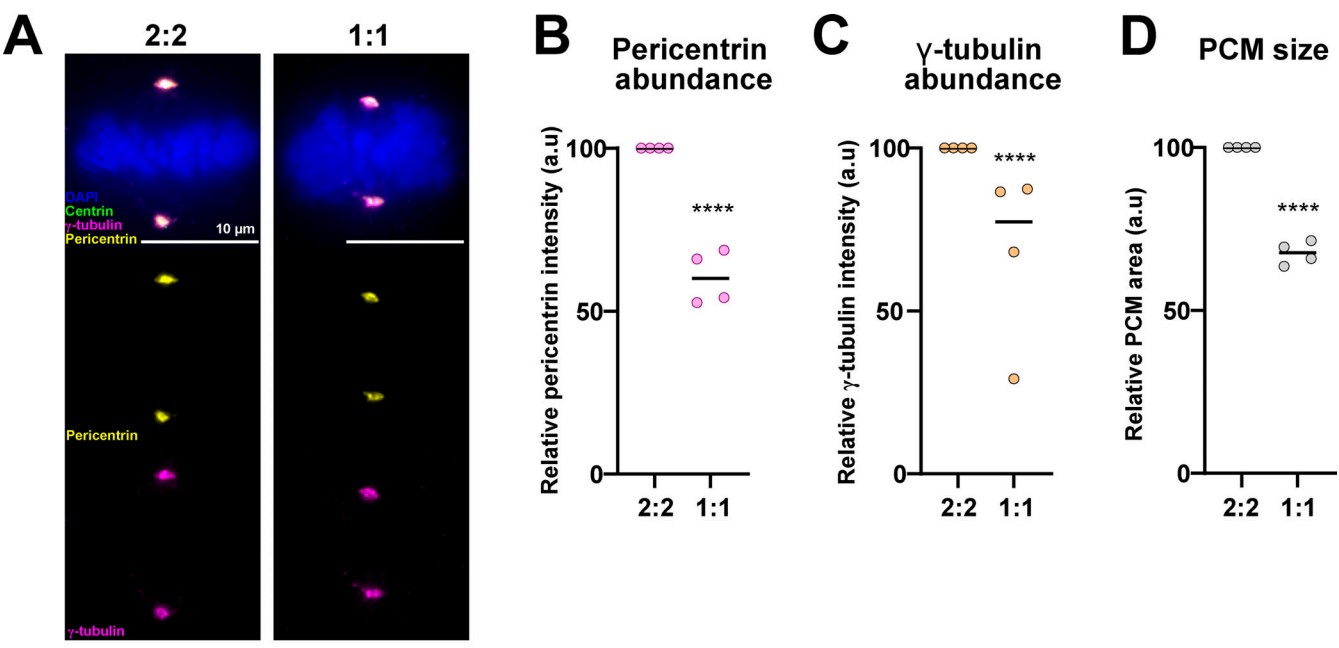

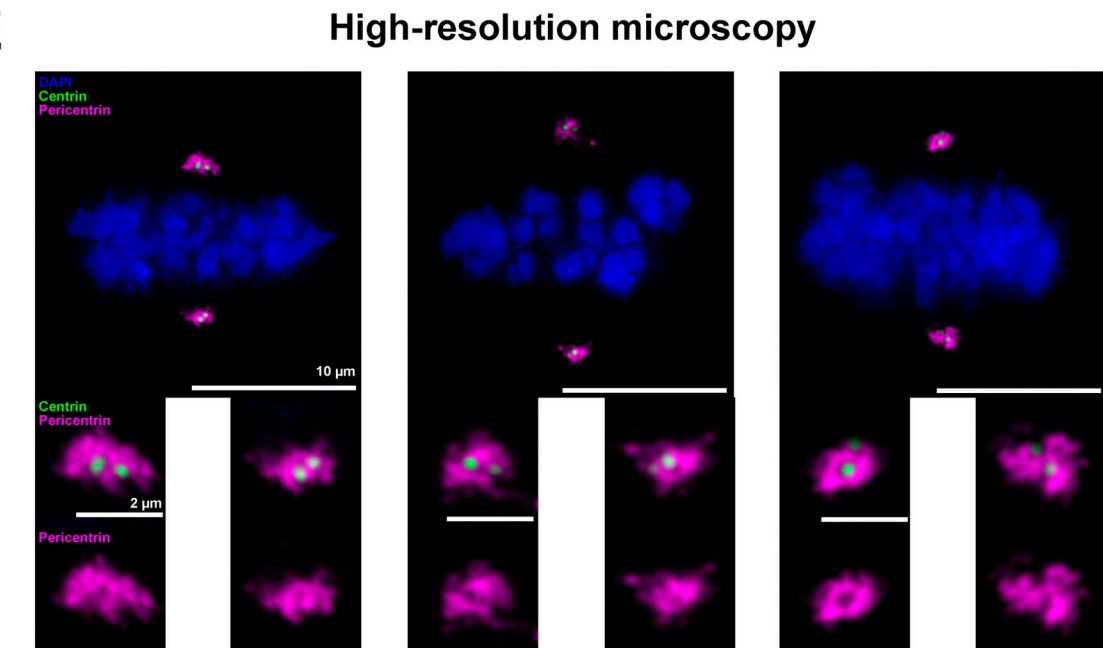

**Figure 8. Daughter centrioles contribute to the recruitment of PCM proteins to centrosomes. (A)** Immunofluorescence images of 2:2 and 1:1 RPE1 GFP-Centrin1 cells in metaphase and stained for GFP-centrin1 (green), pericentrin (yellow), γ-tubulin (magenta) antibodies and chromosomes (DAPI). Scale bars = 10 μm. **(B)** Quantification of the relative pericentrin distribution in 2:2 versus 1:1 cells. Medians of 100%, n = 73 cells in 2:2 cells and 67.4% SEM ± 8.2%, n = 60 cells in 1:1 cells. P < 0.0001 in Mann–Whitney test. **(C)** Quantification of the relative γ-tubulin distribution in 2:2 versus 1:1 cells. Medians of 100%, n = 73 cells in 2:2 cells and 77.4% SEM ± 27.3%, n = 60 cells in 1:1 cells. P < 0.0001 in Mann–Whitney test. **(D)** Quantification of the relative PCM volume based on pericentrin intensity in 2:2 versus 1:1 cells. Median of 100%, n = 73 cells in 2:2 cells and 67.6% SEM ± 3.5%, n = 60 cells in 1:1 cells. P < 0.0001 in unpaired t test with Welch's correction. **(E)** High-resolution immunofluorescence images of metaphase RPE1 GFP-centrin1 cells, stained with DAPI (blue), GFP-centrin1 (green), and pericentrin (magenta) antibody. Scale bars = 10 μm. Insets show a maximum intensity projection of pericentrin at centrosomes. Scale bars = 2 μm.

asymmetry, particularly in the context of asymmetric stem cell divisions. Our results also point to the existence of other, centrosome-independent mechanisms that may control spindle size symmetry: while TPX2 depletion reduces spindle asymmetry in 1:0 cells, depletion of pericentrin or Cdk5Rap2 increases it; moreover, we find that the abundance of TACC3 only correlates with spindle size asymmetry in spindles with centrosome-free spindle poles. Consistently, the HURP MAP is specifically

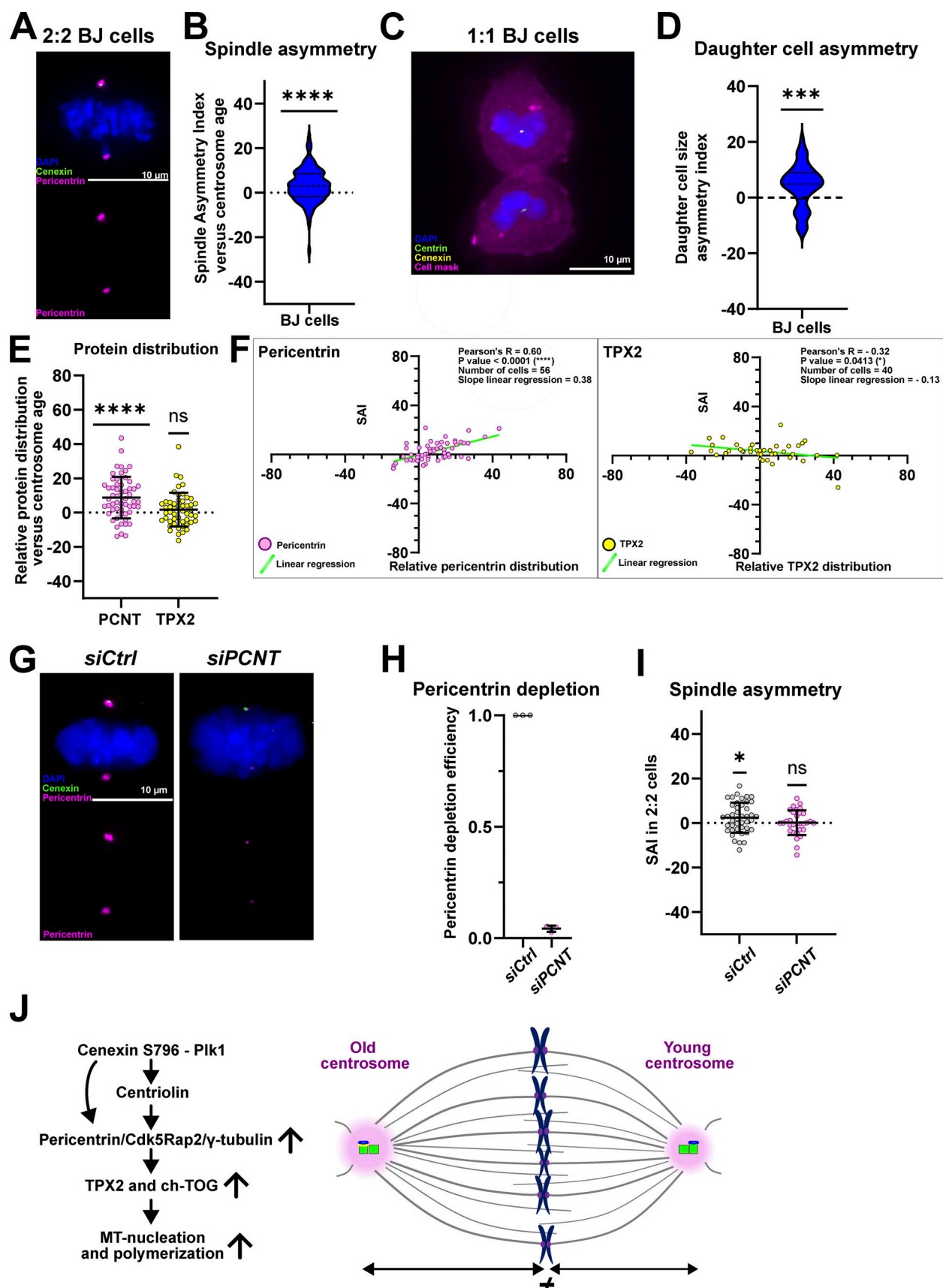

**Figure 9.** **Centrosome age also breaks spindle symmetry in human fibroblast cells. (A)** Immunofluorescence image of 2:2 BJ metaphase cell stained with DAPI (blue), cenexin (green), and pericentrin (magenta) antibodies. **(B)** Quantification of the SAI in 2:2 BJ cells. SAI mean of 3.1 ± 7.6%, *n* = 106 cells, P < 0.0001

in one-sample *t* test. **(C)** Immunofluorescence images of 1:1 BJ late telophase cell stained with DAPI (blue), centrin (green), cenexin (yellow) antibodies and the cell mask (magenta). **(D)** Quantification of the daughter cell area asymmetry of 1:1 BJ cells. Daughter cell size asymmetry of 4.2 ± 7.8%, *n* = 47 cells, P = 0.0007 in one-sample *t* test. **(E)** Quantification of the relative protein distribution of pericentrin and TPX2 versus centrosome age in 2:2 BJ cells. Relative pericentrin distribution of 8.7 ± 12.1%, *n* = 56 cells, P < 0.0001. Relative TPX2 distribution of 1.7 ± 9.9%, *n* = 51 cells, P = 0.4329. A one-sample *t* test was used for pericentrin statistical analysis and a one-sample Wilcoxon-test was used for TPX2 statistical analysis. **(F)** Correlation between the relative pericentrin/TPX2 distribution (x axis) and the SAI (y axis) in 2:2 BJ cells. Each dot represents a cell for which the relative pericentrin/TPX2 distribution and the SAI were measured. The Pearson correlation coefficient and its associated P value, the number of cells analyzed as well as the slope of the linear regression (light green line) are indicated for each condition. **(G)** Immunofluorescence images of *siCtrl*- and *siPCNT*-treated 2:2 BJ cells, stained with DAPI (blue), cenexin (green), and pericentrin (magenta) antibodies. **(H)** Quantification of pericentrin depletion efficiency of *siCtrl*- and *siPCNT*-treated 2:2 BJ cells. Pericentrin fluorescence intensity means of 100%, *n* = 33 cells in *siCtrl* and 4.2 ± 1.4%, *n* = 30 cells in *siPCNT* cells. **(I)** Quantification of the SAI of *siCtrl*- and *siPCNT*-treated 2:2 BJ cells. SAI means of 2.4 ± 6.8%, *n* = 43 cells, P = 0.0250, in *siCtrl* cells and 0.1 ± 5.6%, *n* = 43 cells, P = 0.8988 in *siPCNT*-treated cells. One-sample *t* tests were used for statistical analyses. All scale bars = 10 μm. ns = not significant; P ≤ 0.05 = *; P ≤ 0.001 = ***; P ≤ 0.0001 = ****. **(J)** Schematic model for the control of spindle asymmetry in RPE1 cells. We postulate that spindle size asymmetry is controlled in a centrosome-age-dependent manner via the cenexin-bound pool of Plk1 and centriolin, which both regulate the abundance of pericentriolar material required for microtubule (MT) nucleation, which itself regulates the abundance of TPX2 and ch-TOG.

---

required for spindle asymmetry in 1:0 cells by controlling kinetochore–microtubule stability (Dudka et al., 2019). Such mechanisms might be particularly important in centrosome-free systems, such as meiotic oocytes. In line with this hypothesis, we note that TACC3 and TPX2, whose abundance strongly correlated with spindle size asymmetry in 1:0 cells, but also HURP, are particularly important for spindle assembly in murine and human oocytes (Brunet et al., 2008; Wu et al., 2022; Breuer et al., 2010).

At the detailed molecular and temporal level, we find that centrosome age affects microtubule nucleation via three levels. First, a cenexin-bound subpool of Plk1 on the old centrosomes promotes, via Plk1 activity, asymmetric recruitment of pericentrin, Cdk5Rap2, and γ-tubulin. These data are consistent with data showing that the S796 phosphorylation site on cenexin contributes to the efficient recruitment of pericentrin and γ-tubulin (Soung et al., 2009). It should be, however, noted that in the specific case of pericentrin, S796A-cenexin cells still display a partial enrichment on the old centrosomes. This could be due to the ability of pericentrin to bind to the longer older centriole via its PACT domain (Kong et al., 2020; Gillingham and Munro, 2000). At the temporal level, we show that pericentrin and Plk1 are already asymmetrically distributed before mitotic entry and that this asymmetry expands to the newly recruited γ-tubulin during centrosome maturation. Using *centriolin*$^{-/-}$ cells that still recruit cenexin (Mazo et al., 2016), we show that subdistal appendage proteins promote the asymmetry of the PCM beyond the mere recruitment of Plk1. Whether these asymmetries depend on the SDA themselves, which are present until late G2, or the individual proteins that are reorganized during mitosis (Sullenberger et al., 2020; Bowler et al., 2019), remains to be seen, since we observe these asymmetries both in G2 and mitosis. Downstream of the pericentriolar proteins, we show that pericentrin depletion diminishes the abundance of TPX2 or ch-TOG at spindle poles. TPX2 forms a complex with Aurora-A and the minus-end binding protein NuMA at spindle poles (Kufer et al., 2002; Polverino et al., 2021), but another subpool of TPX2 might interact with the PCM and might thus be recruited in a differential manner to both spindle poles. In the case of ch-TOG, we note that it is recruited to the distal part of the oldest centriole in interphase via its interaction with the subdistal appendage protein CEP128, which could explain its asymmetric distribution in mitosis (Ali et al., 2023). Future

investigations will be necessary to decipher the direct molecular interactions between all these players and potential missing intermediate proteins. Moreover, the fact that the depletion of any of those proteins imposes a symmetric spindle size suggests that their combined contribution to microtubule nucleation is important to establish spindle size asymmetry. The contribution of ch-TOG to spindle size symmetry (Fig. 5) and the potential contribution of EB1 (Table 1) suggests that it will be also important to test in the future whether the microtubule polymerization rates at both poles might differ.

Finally, we find that 1:1 cells display a stronger asymmetry than WT 2:2 cells, indicating that daughter centrioles are not just immature passengers of the spindle poles during mitosis, but that they have an active role. This is consistent with our observation that daughter centrioles can recruit PCM in mitosis, and the fact that under conditions of premature centriole disengagement, daughter centrioles can nucleate and organize spindle microtubules (Wilhelm et al., 2019; Logarinho et al., 2012). Although speculative at this stage, the fact that we identify an activity that in WT cells suppresses centrosome-age spindle asymmetry suggests that cells have selected mechanisms that can actively counteract this asymmetry and thus modulate it according to different types of cell division.

## Materials and methods

### Cell culture, cell lines, and drug treatments

hTert-RPE1, hTert-RPE1 GFP-centrin1 (kind gift of A. Khodjakov, New York State Department of Health, Wadsworth Center), hTert-RPE1 *cenexin*$^{-/-}$, hTert-RPE1 *centriolin*$^{-/-}$ (both king gift from M.F. Tsou, National Yang-Ming University), and hTert-RPE1 *cenexin*$^{-/-}$ GFP-centrin1 mScarlet-cenexin (this study) and hTert-RPE1 *cenexin*$^{-/-}$ GFP-centrin1 mScarlet-Cenexin S796A (this study) cells were cultured in high glucose DMEM (41965; Thermo Fisher Scientific), supplemented with 10% FCS (S1810; LabForce), and 1% penicillin/streptomycin (15140; Thermo Fisher Scientific). hTert-BJ cells (kind gift from R. Medema, Netherlands Cancer Institute) were cultured in DMEM:F12 (11320033; Thermo Fisher Scientific) and supplemented with 10% FCS and 1% penicillin/streptomycin. Centrinone (Wang et al., 2009) was added to the culture medium at 300 nM during either 24, 48, or 72 h to obtain 1:1, 1:0, and 0:0 cells, respectively. Centrinone-containing medium was exchanged twice

a day with fresh centrinone aliquots to prevent Plk4 reactivation. To increase the number of polar chromosomes, cells were treated for 2 h with a low dose of nocodazole (10 ng/ml) (Vasquez-Limeta and Loncarek, 2021; Gasic et al., 2015). Plk1 was inhibited for 16 h using 50 nM of the BI2536 inhibitor (Lénárt et al., 2007).

### Live cell imaging

Cells were cultured in 8-well µ-Slide Ibidi chambers (80806; Ibidi) in Leibovitz's L-15 medium without Phenol Red (21083; Thermo Fisher Scientific) supplemented with 10% FCS. To visualize DNA, 1 nM of SiR-DNA (Spirochrome) was added in the L-15 medium 4 h prior to imaging and recorded on a Nikon Eclipse Ti-E wide-field microscope (Nikon) equipped with a GFP/mCherry/Cy5 filter set (Chroma Technology Corp.), an Orca Flash 4.0 complementary metal-oxide-semiconductor camera (Hamamatsu). To measure spindle size asymmetry and cortex-centrosome distances, cells were imaged using a 60× oil objective at 1-min intervals (19 steps of 0.5-µm Z-stacks). For mitotic timing measurements, cells were imaged using a 40– oil objective at 2–3-min intervals (15 steps of 1-µm Z-stacks).

### Lentivirus production and cell transduction

To generate the hTert-RPE1 cenexin$^{-/-}$ expressing GFP-centrin1 and Scarlet-cenexin, hTert-RPE1 cenexin$^{-/-}$ GFP-centrin1 cells were transduced with lentivirus containing a plasmid coding mScarlet-cenexin. $4.5 \times 10^6$ 293T cells (CRL-11268; ATCC) were seeded in a 10-cm dish and transiently transfected for 24 h with 10 µg psPax2 packaging plasmid (plasmid #12260; Addgene), 5 µg pMD2.G envelope plasmid (plasmid #12259; Addgene), and with 15 µg mScarlet-cenexin plasmid (Twist BioScience) in a CaCl$_2$ 250 mM and Hebs1X (Hepes 6 g/L, NaCl 8 g/L, Na2HPO4 0.1 g/L, dextrose 1 g/L) medium. After 72 h, viruses were harvested from the supernatant and filtered with a 0.4-µm filter. $1 \times 10^6$ hTert-RPE1 cenexin$^{-/-}$ GFP-centrin-1 cells were transduced with 50 µl virus harvest in a 10-cm dish. After 24 h, the virus-containing medium was exchanged with fresh DMEM. Individual positive clones expressing mScarlet-cenexin were sorted by FACS after 72 h.

### Site-directed mutagenesis of the cenexin cDNA

To obtain the mScarlet-cenexin S796A construct, we replaced by PCR the UCU codon coding for Serine 796 with an Alanine GCU codon using the mScarlet-cenexin plasmid as template. Site-directed mutagenesis was performed according to manufacturer's instructions (QuickChange II, 200523-12; Agilent). Following the PCR reaction, the template plasmid was digested with Dpn I, and we used 10 µl of the reaction mix to transform 50 µl of E. coli Mach1 (C8620-03; Invitrogen) by heat shock. The sequence was verified by sequencing. Virus production and cell transduction were performed as described for the mScarlet-cenexin plasmid.

### siRNA transfections

h-Tert-RPE1 and h-Tert-BJ cells were transfected for 24 h (siTPX2 and siCdk5Rap2), 48 h (sipericentrin, sicentrobin, and sich-TOG), and 72 h (siKif2a) with 20 nM siRNAs using Opti-MEM and Lipofectamine RNAiMAX (51985091 and 13778030; Thermo Fisher Scientific) in MEM medium (41090; Thermo Fisher Scientific) supplemented with 10% FCS and according to the manufacturer's instructions. To deplete γ-tubulin, two consecutive transfections for 72 and 48 h with 40 nM siRNA were performed (Gomez-Ferreria et al., 2007). The medium was replaced 24 h after transfection with DMEM supplemented with FCS 10 and 1% penicillin/streptomycin. The following sense strands of validated siRNA duplexes were used: control (5′-GGACCUGGAGGUCUGCUGUUTT-3′; Qiagen); Kif2A (5′-GUUGUUUACUUUCCACGAA-3′; Qiagen, Wilhelm et al., 2019); TPX2 (5′-GAAUGGAACUGGAGGGCUUTT-3′; Qiagen); pericentrin (5′-UGGACGUCAUCCAAUGAGATT-3′; Dharmacon); Cdk5Rap2 (5′-UGGAAGAUCUCCUAACUAATT-3′; Dharmacon); ch-TOG (5′-GAGCCCAGAGUGGUCCAAATT-3′; Dharmacon); γ-tubulin (5′-GGACAUGUUCAAGGACAACUUUGAUTT-3′; Dharmacon; identical sequence used in Gomez-Ferreria et al. (2007); and centrobin (mix of 5′-UGGAAAUGGCAGAACGAGA-3′, GCAUGAGGCUGAGCGGACA-3′, 5′-GCCCAAGAAUUGAGUCGAA-3′, and 5′-CUCCAAACCUCACGUGAUA-3′; Dharmacon).

### Immunofluorescence

Cells were grown on glass coverslips and either fixed for 6 min at –20°C with ice-cold methanol or with 10% formaldehyde, Pipes 20 mM pH 6.8, EGTA 10 mM, triton 0.2% for 10 min at room temperature. Cells were washed twice with phosphate-buffer saline solution (PBS) and blocked overnight at 4°C in blocking buffer (PBS, BSA 3.75%, sodium azide 0.025%). Primary antibodies were diluted in blocking buffer and incubated for 1 h at room temperature. The following primary antibodies were used: cenexin (1:500, 43840; Abcam), centrin (1:2,000; 04–1624; clone 20H5; Merck Millipore), centriolin (1:250, 365521; Santa Cruz biotechnology), centrobin (1:000; 70448; Abcam), CEP192 (1:1,000 PA5-59199; Thermo Fisher Scientific), Aurora-A T288 (1:1,000, 100-2371; Novus biologicals), α-tubulin (1:500; Guerreiro and Meraldi, 2019), γ-tubulin 1/1,000 (Wilhelm et al., 2019), γ-tubulin (1:1,000; T6557; Sigma-Aldrich), pericentrin (1:1,000, 28144; Abcam), Cdk5Rap2 (1:500; 86340; Abcam), TPX2 (1:500, 32795; Abcam), NuMa (1:1,000, 97585; Abcam), Kif2A (1:500, PA3 16833; Invitrogen), katanin (1:500, 14969-1-AP; Lubio Science), TACC3 (1:500, 134151; Abcam), ch-TOG (1:500, 236981; Abcam), MCAK (1:1,000; Cytoskeleton), EB1 (1:500, 610535; BD biosciences), mScarlet (1:500, MBS448290; Mybiosource), and Plk1 (1:250, 189139; Abcam). After two washes with PBS, cells were incubated with the secondary antibodies tagged with Alexa Fluor fluorophores diluted in blocking buffer (1:500) for 1 h at room temperature, cells were washed twice with PBS, and coverslips were mounted on a slide with Vectashield medium containing DAPI (Vector laboratories). To stain cell membranes, cells were incubated (after the staining) for 15 min with Cell Mask (1:2,000, c10046; Thermo Fisher Scientific) in PBS, rinsed with PBS, and mounted with Vectashield medium containing DAPI. To stain polar chromosomes, cells were fixed in cold methanol for 6 min, washed, and blocked in the blocking buffer. Cells expressing either GFP-centrin1 or GFP-centrin1 and mScarlet-cenexin were incubated with CENP-A antibody (1:1,000, MA1-20832; Invitrogen) for 1 h, washed with PBS, and incubated for 1 h with

Alexa Fluor 488 diluted in blocking buffer (1:500). Cells were next washed twice with PBS and coverslips were mounted with Vectashield medium containing DAPI. Microscopy images were acquired using 60× and 100× (NA 1.4) oil objectives on an Olympus DeltaVision wide-field microscope (GE Healthcare) equipped with a DAPI/FITC/TRITC/Cy5 filter set (Chroma Technology Corp.) and Coolsnap HQ2 CCD camera (Roper Scientific) running Softworx (GE Healthcare). Z-stacks of 12.80-µm thickness were imaged with z-slices separated by 0.2 µm. 3D image stacks were deconvolved using Softworx (GE Healthcare) in conservative mode. Pericentrin ring images on metaphase cells were acquired using a 63× (NA 1.4) oil objective on Axio Imager LSM800 microscope (Leica) equipped with an Airyscan module. Z-slices of 0.2-µm thickness were taken to obtain a final volume containing the two centrosomes.

### Microtubule renucleation assay
h-Tert-RPE1 GFP-centrin-1 cells were treated with ice-cold medium and put on ice for 1 h. Cells were either fixed 10 min with cold fixation buffer (formaldehyde 4%, Pipes 20 mM pH 6.8, EGTA 10 mM, triton 0.2%) or incubated 15 s (WT) or 1 min (siCdk5Rap2) with warm DMEM before fixation with warm fixation buffer for 10 min. Cells were rinsed with PBS, blocked overnight at 4°C in blocking buffer (PBS, BSA 3.75%, sodium azide 0.025%), stained with anti-cenexin (1:500, 43840; Abcam) and anti-α-tubulin (1:500, Guerreiro and Meraldi, 2019) primary antibodies for 1 h, rinsed with PBS, and incubated with secondary antibodies (1:500; Alexa Fluor) for 1 h. Coverslips were mounted with Vectashield medium containing DAPI.

### Image processing and analysis
All images were processed and analyzed in 3D using Imaris (Bitplane). To measure the half-spindle length, the distance from the pole to the center of the metaphase plate was measured for the two half-spindles using a previously established method (Dudka et al., 2019). For each half-spindle, we measured in 3D the distance between the centrin or γ-tubulin center and the center of mass of the DAPI-stained DNA mass, as automatically determined by IMARIS (Bitplane). For 2:2 and 1:1 cells, the SAI percentage was calculated as the difference between the half-spindle length associated to the old centrosome and the half-spindle length associated to the young centrosome over the sum of both distances: $(L1–L2)/(L1+L2) *100$ (Fig. S1 C). For 1:0 cells, the half-spindle connected to the centriole-containing pole was used as a reference to measure the "old" L1 distance. For 0:0 cells, the L1 distance was attributed to the pole facing the upper side of the image. The cortex-to-centrosome distance was measured in fixed samples between the border of the cell determined by the membrane staining (cell mask) and the centrosome location based on centrin staining in Imaris (Bitplane). The D1 distance refers to the distance from the cortex to the old centrosome and the D2 distance from the cortex to the young centrosome. In live cell, the edge of the cell was determined using the cytoplasmic fluorescence of GFP-centrin1 protein and the location of the centrosome based on GFP-centrin1. The distance was measured by drawing a line profile in ImageJ for the last time point before anaphase onset ($t = –1$ min). For the

daughter cell area measurements, the maximal area of the two daughter cells was measured in 3D sections by drawing the real area using the polygon freehand line tool in ImageJ software, as previously described (Kiyomitsu and Cheeseman, 2013). The following formula was used to calculate the asymmetry of the daughter cell sizes: $(A1–A2)/(A1+A2) *100$. The daughter cell area ratio was obtained by dividing the area of the daughter cell inheriting the brightest cenexin dot with the area of its sibling cell.

For protein fluorescence intensity measurements at spindle poles, a sphere was drawn around the protein of interest (or based on centrin1 signal), and the mean intensity at both poles was extracted. The following sphere diameters were chosen for the quantifications (Table 1): 0.5 µm for Aurora-A and Plk1, 1 µm for centrobin, pericentrin, Cdk5Rap2, CEP192, ch-TOG, and a 1.5-µm-diameter sphere was used to quantify TACC3, EB1, and γ-tubulin fluorescence intensity at the spindle poles. The background signal was subtracted from the fluorescence intensity. The following formula was used to calculate the relative protein abundance distribution between the two poles: $(pole1–pole2)/(pole1+pole2)$. As the localization of TPX2, Kif2A, MCAK, and katanin was not restricted to the spindle pole, the surface volume was chosen to evaluate protein concentration at both poles. For each cell, the surface volume ratio was calculated as follows: $(surface1–surface2)/(surface1+surface2)$. A Pearson's correlation coefficient (or Spearman's correlation coefficient when appropriate) was used to assess the relationship between protein distribution and associated SAI for each cell in each condition. A linear regression was performed and the value of the slope was extracted.

The relative protein distribution with respect to centrosome age was measured by taking the mean intensity within a sphere of 0.5 µm for Plk1; 1 µm for pericentrin, Cdk5Rap2, ch-TOG, and γ-tubulin; and a sphere of 2 µm for TPX2.

To evaluate the protein depletion efficiency, the sum intensity was extracted from a 1-µm-diameter sphere and compared with the sum intensity of the respective control condition. The relative protein distribution with respect to the centrosome was measured by taking the pole containing the brightest cenexin or centriolin dot as a reference (pole1). The cortex-to-centrosome distances were measured on cells in metaphase and stained with the cell mask. The distances were measured on a single plane using the slice tool on Imaris. For the microtubule re-nucleation assay, the α-tubulin mean intensity inside a sphere of 2 µm was extracted. The pole containing the brightest cenexin was used as pole1. For the pericentrin ring images around the centrioles, a maximum intensity projection of five to eight Z-slices was performed for each centrosome. To correlate the presence of cenexin with GFP-centrin1 brightness, the mean fluorescence intensity of both cenexin and GFP-centrin1 was extracted from a volume of 300 nm and 200 × 500 nm, respectively. All images were prepared using ImageJ/FIJI (version 2.1.0) and mounted in Inkscape (version 1.2.1). G2 cells were selected based on the number of centrioles, with four centrioles clearly visible and having a PCM structure surrounding the centrioles. Prophase cells were selected based on the partial separation of centrosomes and a beginning DNA condensation of the chromosomes.

## Statistical analysis and reproducibility

Statistical analyses were performed using GraphPad Prism9 (GraphPad). A minimum of three independent biological replicates were performed in all experiments. For parametric tests, a Shapiro–Wilk test was used to test the normality of the data distribution using GraphPad Prism9. When two populations were compared and after testing for normality (Shapiro-Wilk test), an F-test was used to test whether the variances of these two populations were equal using GraphPad Prism9, before comparing the two conditions using a parametric test and applying the appropriate correction when necessary.

A detailed list of the number of cells, the P value, and the statistical test used for each experiment:

Fig. 1 B: WT 2:2 cells, $n$ = 133 cells, P = 0.0104; WT 1:1 cells, $n$ = 96 cells, P < 0.0001; WT 1:0 cells, $n$ = 111 cells, P < 0.0001; WT 0:0 cells, $n$ = 77 cells, P = 0.54. One-sample $t$ tests.

Fig. 1 D: WT 2:2 cells, $n$ = 133 cells, P = 0.0104; *centriolin*$^{-/-}$ 2:2 cells, $n$ = 56 cells, P = 0.3573; WT 1:1 cells, $n$ = 96 cells, P < 0.0001; *centriolin*$^{-/-}$ 1:1 cells, $n$ = 65 cells, P = 0.4229. One-sample $t$ tests.

Fig. 1 F: WT 2:2 cells, 86 cells, P < 0.0001; *centriolin*$^{-/-}$ 2:2 cells, 169 cells, P < 0.0001. Binomial tests. P value = 0.8901 in Fisher's exact test.

Fig. 2 B: WT 2:2 cells, $n$ = 28 cells, P = 0.0145; WT 2:2 (T-1') cells $n$ = 28 cells, P = 0.0012. One sample Wilcoxon tests.

Fig. 2 D: WT 2:2 cells, $n$ = 65 cells, P = 0.0025; WT 1:1 cells, $n$ = 58 cells, P < 0.0001; WT 0:0 cells, $n$ = 70 cells, P = 0.8370. One-sample $t$ tests.

Fig. 2 F: WT 2:2 fixed cells, $n$ = 44 cells, P = 0.2135. Paired $t$ test. WT 2:2 live cells, $n$ = 28 cells, P = 0.4491. Paired $t$ test.

Fig. 2 H: *centriolin*$^{-/-}$ 1:1 cells, $n$ = 54 cells, P = 0.6246. One-sample $t$ test.

Fig. 3: A Pearson correlation was used to analyze the coefficient of correlation between the two variables.

Fig. 4 B: Pericentrin in *siCtrl*, $n$ = 73 cells, P < 0.0001; Cdk5Rap2 in *siCtrl*, $n$ = 72 cells, P < 0.0001; γ-tubulin in *siCtrl*, $n$ = 54 cells, P < 0.0001. One-sample $t$ tests.

Fig. 4 C: *siPCNT*, $n$ = 54 cells, P = 0.8246; *siCdk5Rap2*, $n$ = 50 cells, P = 0.8902; *siγ-tubulin*, $n$ = 60 cells, P = 0.3277. One-sample $t$ tests.

Fig. 4 D: Pericentrin in *centriolin*$^{-/-}$ 1:1 cells, $n$ = 77 cells, P < 0.0001; γ-tubulin in *centriolin*$^{-/-}$ 1:1 cells, $n$ = 44 cells, P = 0.6059. One-sample $t$ tests.

Fig. 4 F: *siCtrl*, $n$ = 23 cells, P = 0.0493; *sicentrobin*, $n$ = 33 cells, P = 0.0002. One-sample $t$ tests.

Fig. 4 H: *siCtrl*, $n$ = 24 cells, P < 0.0001; *siPCNT*, $n$ = 18 cells, P = 0.0007. One-sample Wilcoxon tests.

Fig. 4 I: *siCtrl*, $n$ = 61 cells, P < 0.0001; *siCdk5Rap2*, $n$ = 32 cells, P < 0.0001. One-sample Wilcoxon tests.

Fig. 4 K: Cold depolymerization condition, $n$ = 75 cells, P = 0.2044; re-nucleation condition, $n$ = 76 cells, P = 0.0256. One-sample Wilcoxon tests.

Fig. 5 B: *siCtrl*, $n$ = 74 cells, P = 0.0005; *siTPX2*, $n$ = 102 cells, P = 0.2454. One-sample $t$ tests.

Fig. 5 C: WT 2:2 cells, $n$ = 36 cells, P = 0.0210; *centriolin*$^{-/-}$ 2:2 cells, $n$ = 55 cells, P = 0.5917. One-sample $t$ tests.

Fig. 5 E: *siCtrl*, $n$ = 44 cells; *siPCNT*, $n$ = 45 cells, P < 0.0001; *siTPX2*, $n$ = 30 cells, P < 0.0001. One-way Kruskal–Wallis with Dunnett's multiple comparisons tests.

Fig. 5 F: *siCtrl*, $n$ = 34 cells, P = 0.0249. One-sample Wilcoxon-test.

Fig. 5 G: *siCtrl*, $n$ = 59 cells, P = 0.0057; *sich-TOG*, $n$ = 63 cells, P = 0.1587. One-sample $t$ tests.

Fig. 5 H: *siCtrl*, $n$ = 43 cells, P < 0.0001; *siPCNT*, $n$ = 41 cells, P < 0.0001; *sich-TOG*, $n$ = 47 cells, P <0.0001. One-way Kruskal–Wallis with Dunnett's multiple comparisons tests.

Fig. 6 B: G2 cells, $n$ = 36 cells, P = 0.0001; prophase cells, $n$ = 29 cells, P < 0.0001. One-sample $t$ tests.

Fig. 6 D: G2 cells, $n$ = 50 cells, P = 0.1375; prophase cells, $n$ = 53 cells, P < 0.0001. One-sample $t$ tests.

Fig. 6 F: G2 cells, $n$ = 47 cells, P = 0.0008; prophase cells, $n$ = 35 cells, P = 0.0009. One-sample $t$ tests.

Fig. 6 H: DMSO, $n$ = 43 cells; BI2536, $n$ = 37 cells. P value < 0.0001 in unpaired $t$ test.

Fig. 6 I: DMSO, $n$ = 43 cells, P <0.0001; BI2536, $n$ = 37 cells, P = 0.45. One-sample $t$ tests.

Fig. 6 J: DMSO, $n$ = 46 cells, P = 0.0025; BI2536, $n$ = 24 cells, P = 0.2102. One-sample $t$ tests.

Fig. 6 K: DMSO, $n$ = 43 cells, P = 0.0012; BI2536, $n$ = 37 cells, P = 0.79. One-sample $t$ tests.

Fig. 7 B: WT mScarlet-cenexin 1:1 cells, $n$ = 73 cells, P = 0.0044. One-sample Wilcoxon test. mScarlet-cenexin S796A 1:1 cells, $n$ = 75 cells, P = 0.7096. One-sample $t$ test.

Fig. 7 D: Prophase cells, $n$ = 37 cells, P < 0.0001; metaphase cells, $n$ = 40 cells, P < 0.0001. One-sample $t$ tests.

Fig. 7 E: Prophase cells, $n$ = 33 cells, P = 0.7242; metaphase cells, $n$ = 34 cells, P = 0.6970. One-sample $t$ tests.

Fig. 7 G: WT mScarlet-cenexin cells, $n$ = 40 cells, P = 0.020; mScarlet-cenexin S796A cells, $n$ = 55 cells, P = 0.149. One-sample $t$ test.

Fig. 7 H: mScarlet-cenexin S796A cells, $n$ = 75 cells, P = 0.0014. One-sample $t$ test.

Fig. 7 J: WT mScarlet-cenexin cells, $n$ = 96 cells, P < 0.0001; mScarlet-cenexin S796A cells, $n$ = 75 cells, P = 1. Binomial tests. P value of 0.0003 (***) in Fisher's exact test.

Fig. 8 B: WT 2:2 cells, $n$ = 73 cells, WT 1:1 cells, $n$ = 60 cells, P < 0.0001. Mann–Whitney test.

Fig. 8 C: WT 2:2 cells, $n$ = 73 cells, WT 1:1 cells, $n$ = 60 cells, P < 0.0001. Mann–Whitney test.

Fig. 8 D: WT 2:2 cells, $n$ = 73 cells, WT 1:1 cells, $n$ = 60 cells, P < 0.0001. Mann–Whitney test.

Fig. 9 B: WT 2:2 cells, $n$ = 106 cells, P < 0.0001. One-sample $t$ test.

Fig. 9 D: WT 1:1 cells, $n$ = 47 cells, P = 0.0007. One-sample $t$ test.

Fig. 9 E: Pericentrin, $n$ = 56 cells, P < 0.0001. One-sample $t$ test. TPX2, $n$ = 51 cells, P = 0.4329. One-sample Wilcoxon-test.

Fig. 9 H: *siCtrl*, $n$ = 33 cells. *siPCNT*, $n$ = 30 cells.

Fig. 9 I: *siCtrl*, $n$ = 43 cells, P = 0.0250. *siPCNT*, $n$ = 43 cells, P = 0.8988. One-sample $t$ tests.

Fig. S3 A: *siCtrl*, $n$ = 72 cells; *siPCNT*, $n$ = 54 cells.

Fig. S3 B: *siCtrl*, $n$ = 67 cells; *siCdk5Rap2*, $n$ = 57 cells.

Fig. S3 C: *siCtrl*, $n$ = 54 cells; *siγ-tubulin*, $n$ = 59 cells.

Fig. S3 D: *siCtrl*, $n$ = 23 cells; *sicentrobin*, $n$ = 33 cells.

Fig. S4 A: *siCtrl*, $n$ = 55 cells; *siTPX2*, $n$ = 80 cells.

Submitted: 27 November 2023

Fig. S4 C: *siCtrl*, *n* = 41 cells, P <0.0001; *siTPX2*, *n* = 25 cells, P = 0.0009. One-sample *t* tests.

Fig. S4 D: *siCtrl*, *n* = 44 cells; *siPCNT*, *n* = 45 cells, P < 0.0001; *siTPX2*, *n* = 30 cells, P = 0.137. One-way Kruskal–Wallis with Dunnett's multiple comparisons tests.

Fig. S4 E: *siCtrl*, *n* = 55 cells; *sich-TOG*, *n* = 63 cells.

Fig. S4 G: *siCtrl*, *n* = 43 cells; *siPCNT*, *n* = 41 cells, P < 0.0001; *sich-TOG*, *n* = 47 cells, P = 0.0003. One-way Kruskal–Wallis with Dunnett's multiple comparisons tests.

Fig. S5 B: WT 2:2 cells, *n* = 33 cells; mScarlet-cenexin 2:2 cells, *n* = 76 cells, P = 0.3; mScarlet-cenexin S796A 2:2 cells, *n* = 28 cells, P = 0.45. A Kruskal–Wallis test with Dunn's multiple comparisons.

Fig. S5 D: WT 2:2 cells, *n* = 132 cells; 1:1 WT cells, *n* = 129 cells.

## Online supplemental material

Fig. S1 shows the different centriole configurations examined in this study and the methods used to measure the SAI, the daughter cell size asymmetry, and the cortex to centrosome distance. Fig. S2 depicts the relative abundance of Cdk5Rap2 and γ-tubulin versus the SAI. Fig. S3 quantifies the protein depletion efficiency. Fig. S4 shows additional analysis of the effect of TPX2 downregulation on SAI in 1:0 cells and additional quantification of the interaction between pericentrin and TPX2/ch-TOG. Fig. S5 shows the mitotic timing measurements of the different cell lines and the distribution of centrin with respect to centrosome age in 2:2 and 1:1 RPE1 cells. Video 1 shows our measurements of spindle asymmetry and centrosome to cell cortex distances in live cells approaching anaphase, supporting Fig. 2 F and Fig. S1 D.

## Data availability

All the raw data generated in this paper are available at: https://doi.org/10.26037/yareta:agnc5p5ce5gsfarku72jp3s4ma.

## Acknowledgments

We thank A. Khodjakov (New York State Department of Health, Albany, NY, USA), M.F. Tsou (National Yang-Ming University, Taipei, Taiwan), and R. Medema (Netherlands Cancer Institute, Amsterdam, Netherlands) for reagents. We thank the Flow Cytometry and the Bioimaging Facilities at the Medical Faculty of the University of Geneva for technical support, in particular N. Liaudet for his help in image data analysis. We thank I. Gasic, P. Guichard, M. Gotta, and the members of her laboratory (all at University of Geneva), as well as the members of the Meraldi laboratory for helpful suggestions and critical discussions.

This work was supported by the Swiss National Science Foundation project grant (No. 310030_208052), the Ernest Boninchi Foundation, and the University of Geneva. Open Access funding provided by Université de Genève.

Author contributions: A. Thomas: Conceptualization, Formal analysis, Investigation, Methodology, Validation, Visualization, Writing—original draft, Writing—review & editing, P. Meraldi: Conceptualization, Formal analysis, Funding acquisition, Project administration, Resources, Supervision, Validation, Writing—review & editing.

Disclosures: The authors declare no competing interests exist.

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

# Supplemental material

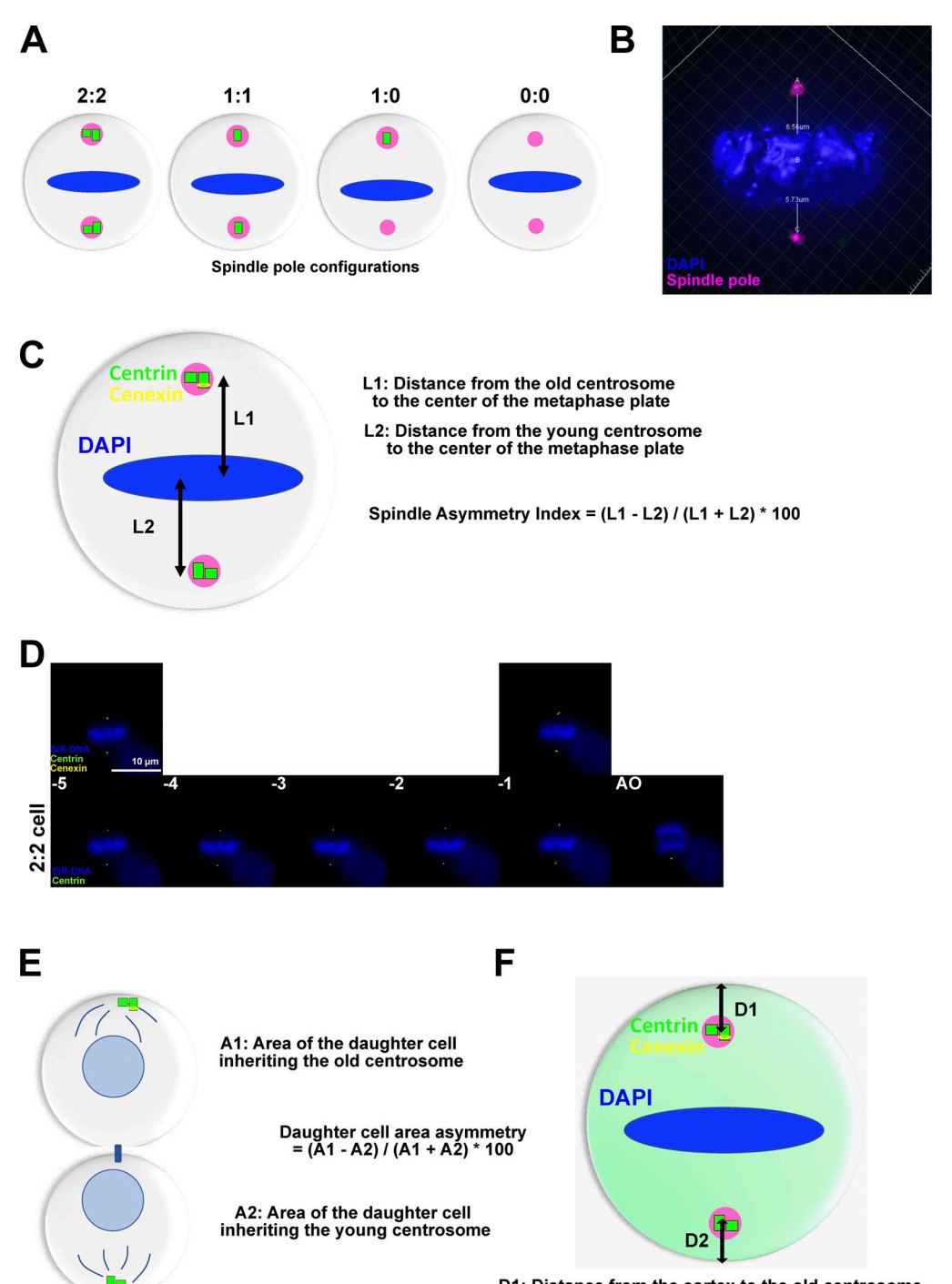

Figure S1. **Methodology to measure spindle size and cell size asymmetry. (A)** Schematic representation of RPE1 metaphase cells displaying different centriole numbers (green) at spindle poles (magenta). Control non-treated cells contain two centrioles at each pole (2:2), cells treated with centrinone during 24, 48, and 72 h display, respectively, one centriole at each pole (1:1), one centriole at one pole (1:0), or no centriole (0:0). **(B)** Immunofluorescence image of a cell in metaphase and stained for a spindle pole protein (magenta) and chromosomes (DAPI). The distances from the spindle poles to the center of the metaphase plate were measured in 3D automatically. **(C)** Scheme to explain the calculation of the SAI. The L1 distance refers to the distance from the spindle pole with the cenexin-enriched (yellow) old centrosome to the center of the DAPI-stained (blue) metaphase plate (measured in 3D). L2 refers to the distance between the other spindle poles to the metaphase plate. In 0:0 cells, the two spindle poles were assigned randomly. **(D)** Time-lapse images of a 2:2 RPE1 cell expressing GFP-centrin1 mScarlet-cenexin and stained for DAPI dividing from the time 5 min before anaphase onset till the anaphase onset (AO). The times –5 and –1 min before anaphase onset are shown with and without the mScarlet-cenexin channel. **(E)** Schematic representation of an RPE1 cell in late telophase for which the area of the two sibling cells was measured. Area 1 refers to the area of the daughter cell inheriting the old centrosome while area 2 refers to the area of its sibling. The formula = (A1–A2)/(A1+A2) *100 was used to calculate the daughter cell area asymmetry. **(F)** Schematic representation of an RPE1 cell in metaphase expressing GFP-centrin protein. The distance 1 (D1) refers to the distance from the cortex to the old centrosome and D2 the distance from the cortex to the young centrosome.

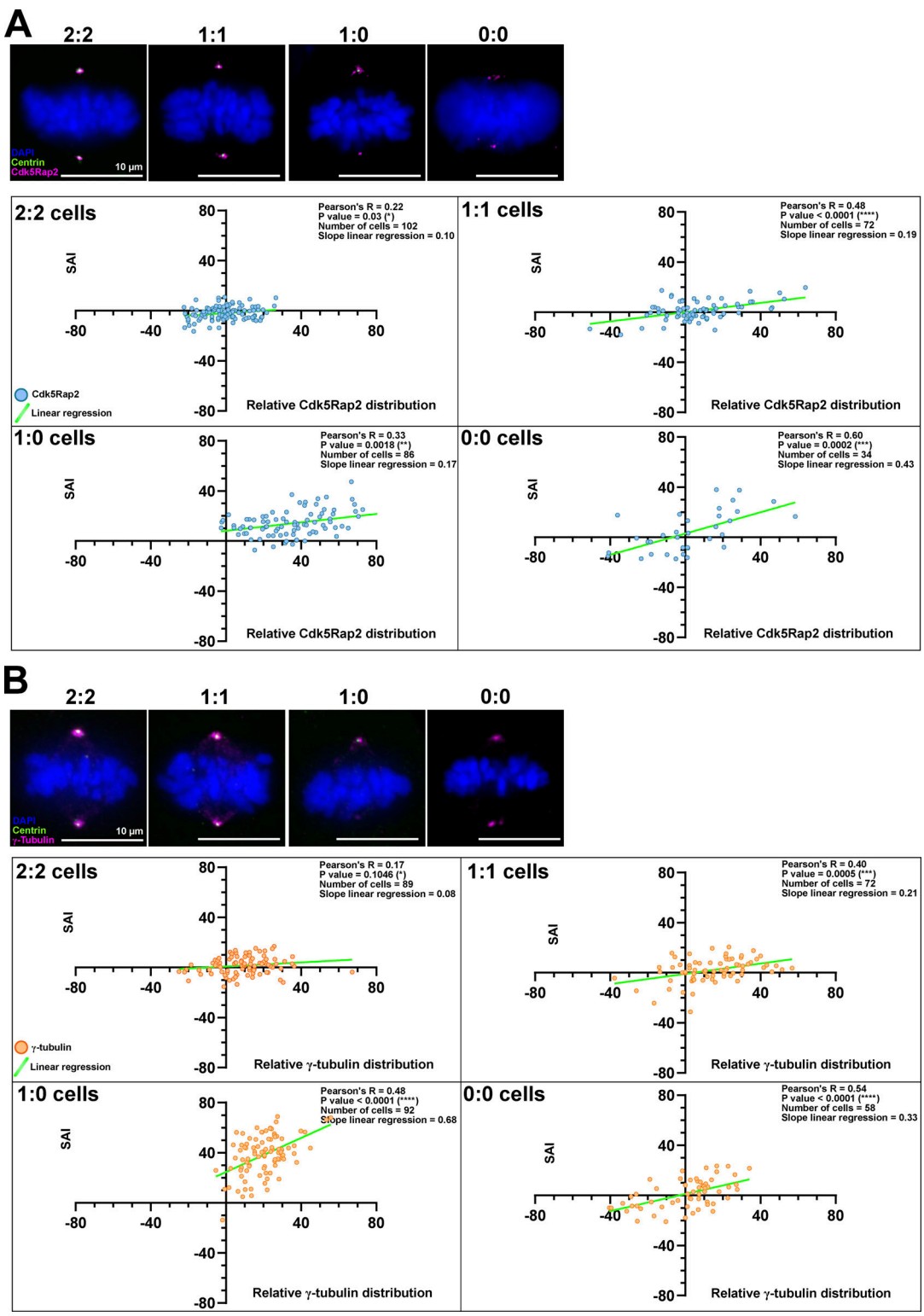

Figure S2. **Cdk5Rap2 and γ-tubulin abundance scales with spindle size. (A)** Immunofluorescence images of 2:2, 1:1, 1:0, and 0:0 RPE1 Centrin1-GFP metaphase cells stained with DAPI (blue), GFP-centrin1 (green), and Cdk5Rap2 antibody (magenta). Scale bars = 10 µm. Correlation plots between the relative Cdk5Rap2 distribution (x axis) and the SAI (y axis) for 2:2, 1:1, 1:0, and 0:0 cells. Dots represent single-cell values. The light green line indicates the slope of the linear regression. For each plot, the Pearson correlation coefficient, its associated P value, the number of cells analyzed, and the slope of the linear regression are indicated. **(B)** Immunofluorescence images of 2:2, 1:1, 1:0, and 0:0 RPE1 GFP-centrin1 metaphase cells stained with DAPI (blue), GFP-centrin1 (green), and γ-tubulin antibody (magenta). Note that these images are also shown in Fig. 1 A Scale bars = 10 µm. Correlation plots between the relative γ-tubulin distribution (x axis) and the SAI (y axis) for 2:2, 1:1, 1:0, and 0:0 cells. Dots represent single-cell values. The light green line indicates the slope of the linear regression. For each plot, the Pearson correlation coefficient, its associated P value, the number of cells analyzed, and the slope of the linear regression are indicated.

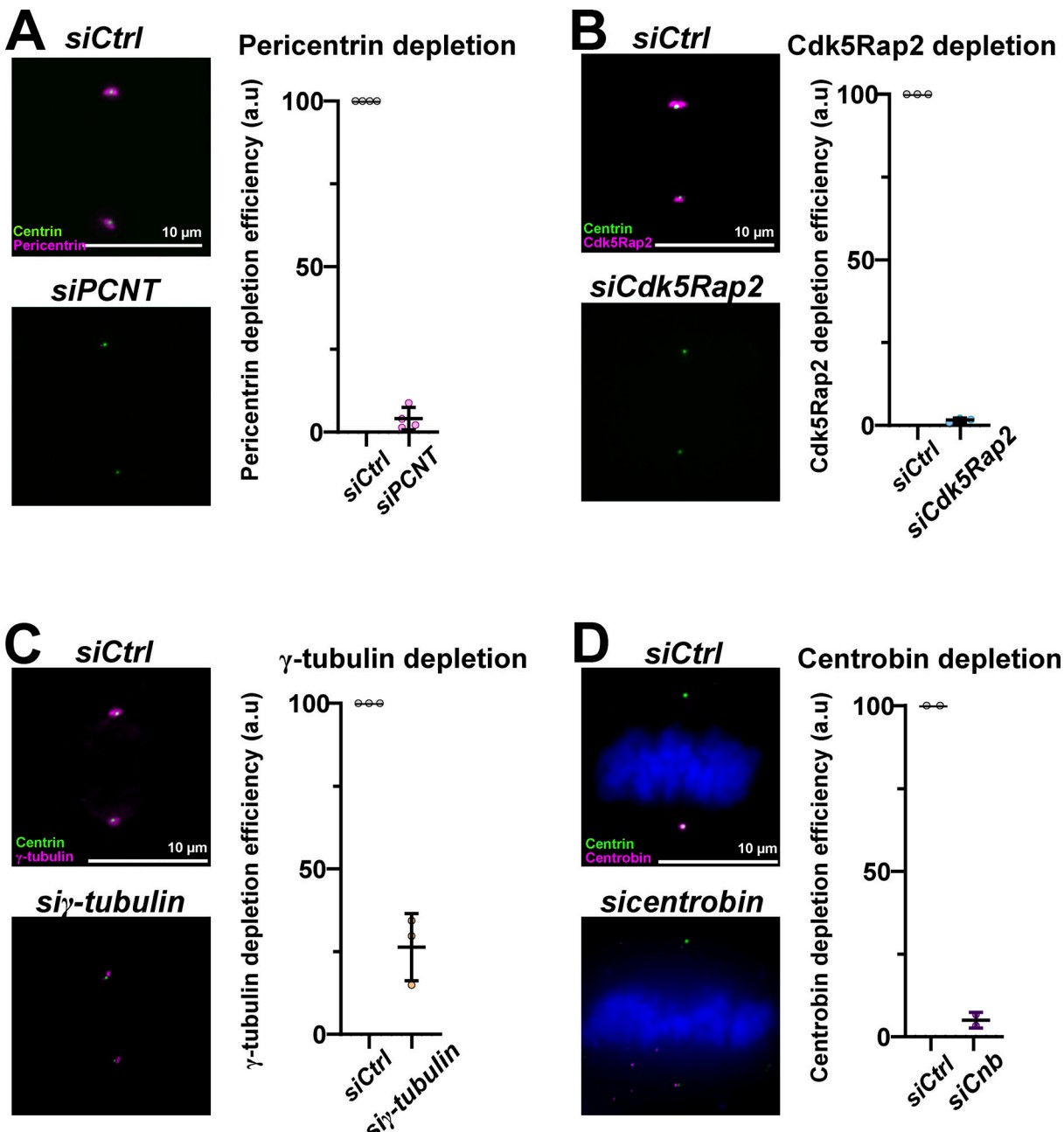

Figure S3. **Validation of protein depletion efficiency. (A)** Immunofluorescence and quantification of pericentrin levels at centrosomes after *siCtrl* and *siPCNT* treatments in 1:1 RPE1 GFP-centrin1 cells. Pericentrin fluorescence intensity means of 100%, *n* = 72 cells and 3.4 ± 2.73%, *n* = 54 cells in *siCtrl* and *siPCNT* cells. **(B)** Immunofluorescence and quantification of Cdk5Rap2 levels after *siCtrl* and *siCdk5Rap2* treatments in 1:1 RPE1 GFP-centrin1 cells. Cdk5Rap2 fluorescence intensity means of 100%, *n* = 67 cells and 1.2 ± 3%, *n* = 57 cells in *siCtrl* and *siCdk5Rap2* cells. **(C)** Immunofluorescence and quantification of γ-tubulin levels after *siCtrl* and *siγ-tubulin* treatment in 1:1 RPE1 GFP-centrin1 cells. γ-Tubulin fluorescence intensity means of 100%, *n* = 54 cells and 8.3 ± 15.1%, *n* = 59 cells in *siCtrl* and *siγ-tubulin* cells. **(D)** Immunofluorescence and quantification of centrobin levels after *siCtrl* and *sicentrobin* treatment in 1:1 RPE1 GFP-centrin1 cells. Centrobin fluorescence intensity means of 100%, *n* = 23 cells in *siCtrl* and 5.0 ± 2.4%, *n* = 33 cells in *sicentrobin* cells. Note that these images are also shown in Fig. 4 E. All scale bars = 10 µm.

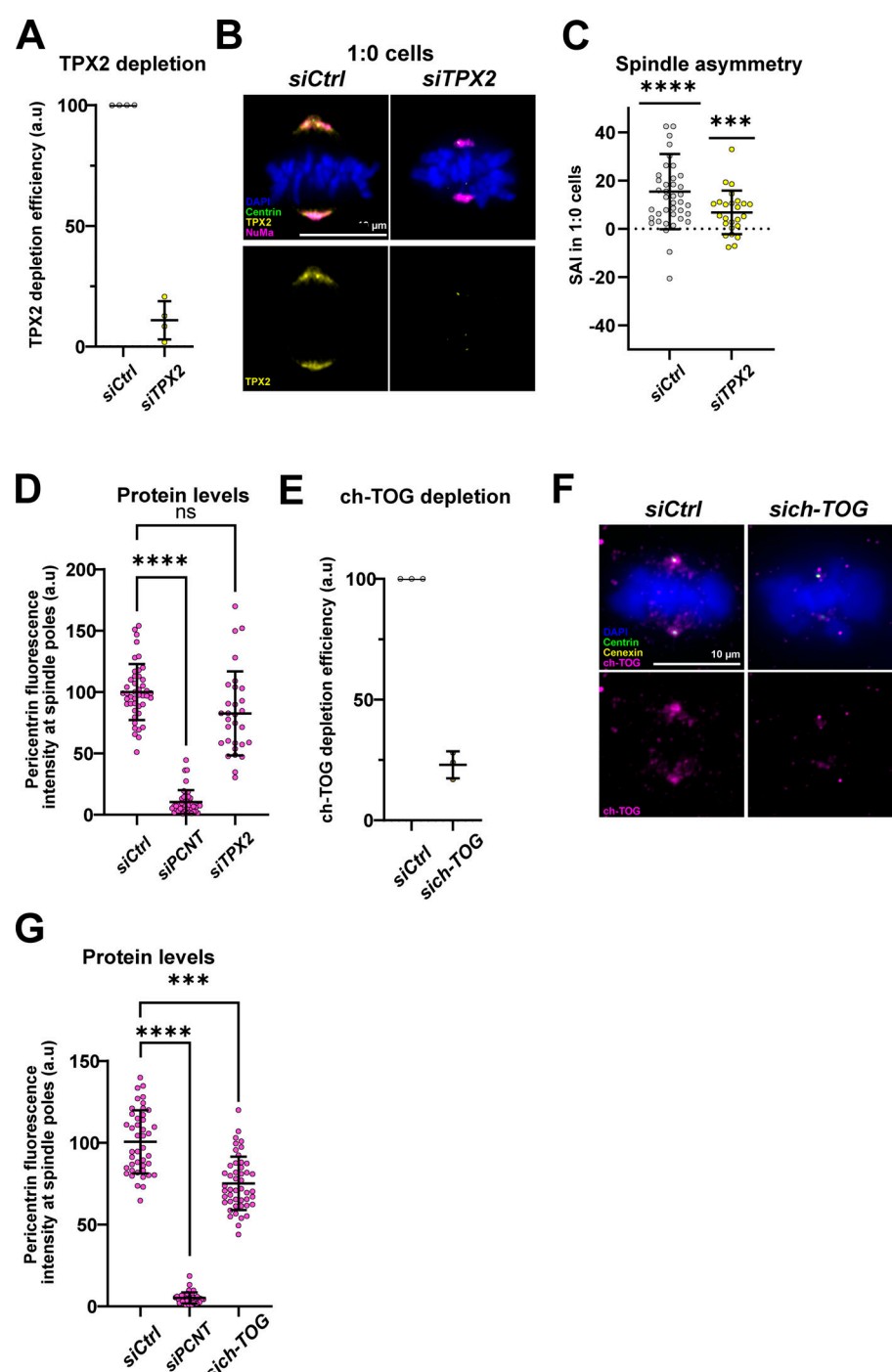

Figure S4. **Proteins regulating microtubule polymerization also regulate spindle size symmetry. (A)** Quantification of TPX2 depletion efficiency after *siCtrl* and *siTPX2* treatments in 1:1 RPE1 GFP-centrin1 cells. TPX2 fluorescence intensity means of 100%, *n* = 55 cells in *siCtrl* cells and 11 ± 7.9%, *n* = 80 cells in *siTPX2* cells. **(B)** Immunofluorescence images of *siCtrl* and *siTPX2*-treated 1:0 RPE1 Centrin1-GFP cells, stained with DAPI (blue), GFP-centrin1 (green), TPX2 (yellow), and NuMa (magenta) antibodies. Scale bar = 10 μm. **(C)** Quantification of the SAI of *siCtrl and siTPX2*-treated 1:0 RPE1 Centrin1-GFP cells. SAI means of 15.4 ± 15.6%, *n* = 41 cells, P < 0.0001, in *siCtrl* cells and 6.8 ± 9%, *n* = 25 cells, P = 0.0009, in *siTPX2* cells. One-sample *t* tests were used for statistical analyses. **(D)** Quantification of pericentrin fluorescence intensity at spindle poles of *siCtrl*, *siPCNT*, and *siTPX2*-treated 2:2 RPE1 GFP-centrin1 cells. Means of 100 ± 22.9%, *n* = 44 cells in *siCtrl* cells, 10.4 ± 9.6%, *n* = 45 cells, P < 0.0001 in *siPCNT* cells, and 82.7 ± 34.3%, *n* = 30 cells, P = 0.137 in *siTPX2*-treated cells. One-way Kruskal–Wallis with Dunnett's multiple comparisons tests. **(E)** Quantification of ch-TOG depletion efficiency after *siCtrl* and *sich-TOG* treatments in 1:1 RPE1 GFP-centrin1 mScarlet-cenexin cells. Ch-TOG fluorescence intensity means of 100%, *n* = 55 cells in *siCtrl* cells and 23 ± 5.6%, *n* = 63 cells in *sich-TOG* cells. **(F)** Immunofluorescence images of *siCtrl* and *sich-TOG*–treated 1:1 RPE1 GFP-centrin1 mScarlet-cenexin cells stained with DAPI (blue), GFP-centrin1 (green), cenexin (yellow), and ch-TOG (magenta) antibodies. Scale bar = 10 μm. **(G)** Quantification of pericentrin fluorescence intensity at spindle poles of *siCtrl*, *siPCNT*, and *sich-TOG*–treated 2:2 RPE1 GFP-centrin1 cells. Means of 100 ± 19.3%, *n* = 43 cells in *siCtrl* cells, 5.2 ± 3.3%, *n* = 41 cells, P < 0.0001 in *siPCNT* cells, and 75.3 ± 16.3%, *n* = 47 cells, P = 0.0003 in *sich-TOG*–treated cells. One-way Kruskal–Wallis with Dunnett's multiple comparisons tests. ns = not significant; P ≤ 0.001 = ***; P ≤ 0.0001 = ****.

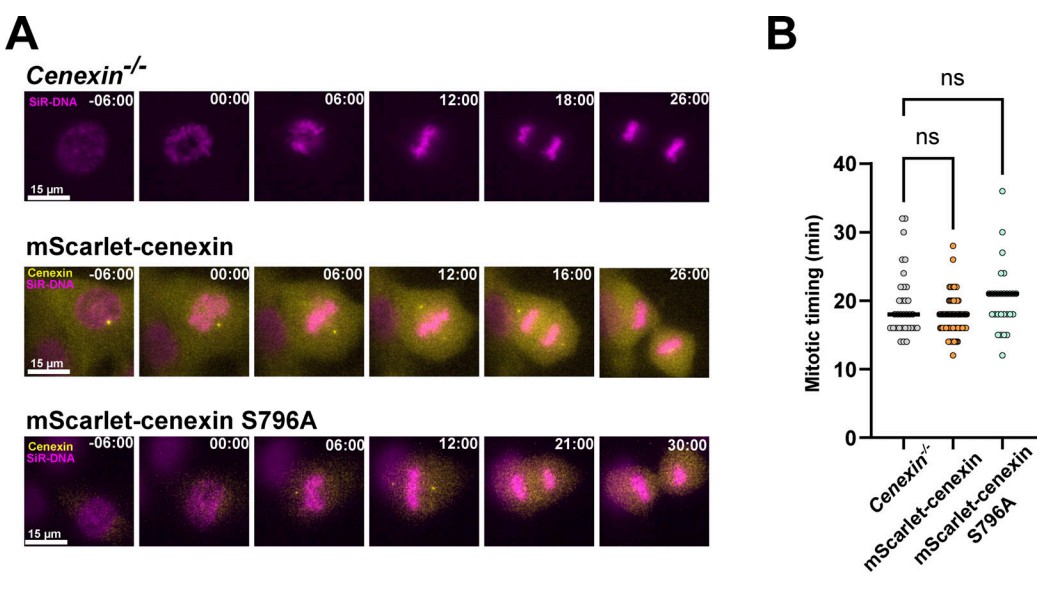

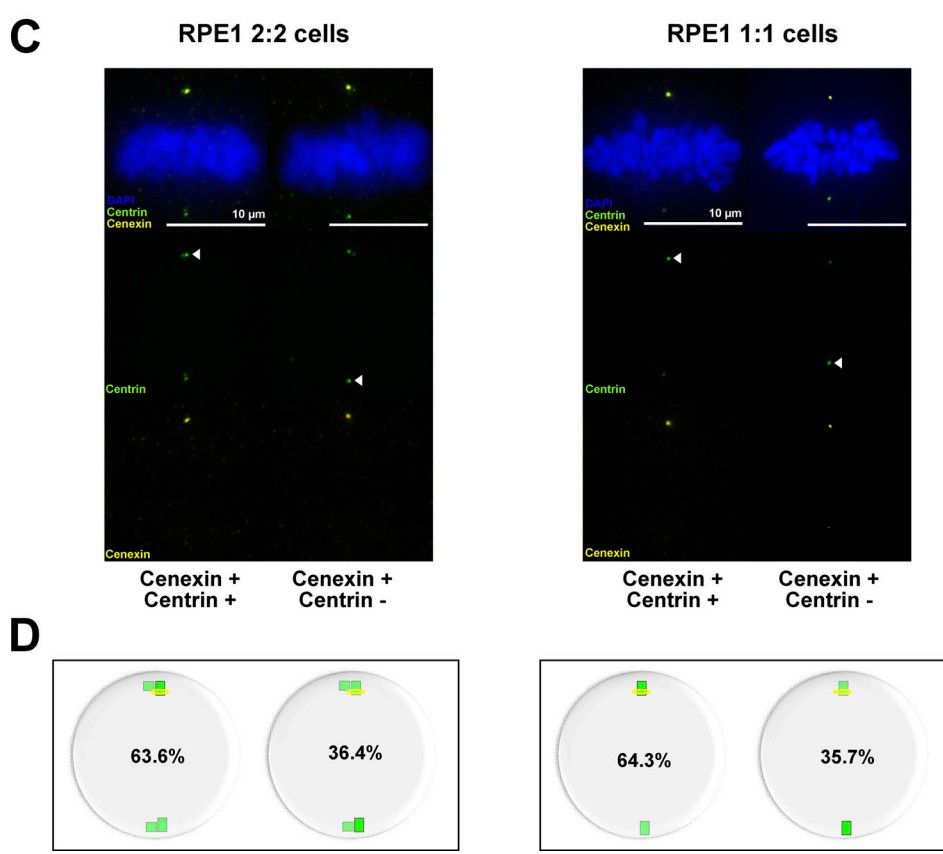

Figure S5.    **Expression of WT or S796A cenexin leaves mitotic timing unaffected. (A)** Image sequences of parental 2:2 RPE1 GFP-centrin1 *cenexin⁻/⁻* cells, 2:2 RPE1 GFP-centrin1 cells expressing WT mScarlet-cenexin, and 2:2 RPE1 GFP-centrin1 cells expressing S796A mScarlet-cenexin. **(B)** Plots of mitotic timing from nuclear envelope breakdown = 0 min till anaphase time in parental 2:2 RPE1 GFP-centrin1 *cenexin⁻/⁻*, 2:2 RPE1 GFP-centrin1 WT mScarlet-cenexin, and 2:2 RPE1 GFP-centrin1 S796A mScarlet-cenexin cells. Median of the mitotic timing of 19.5 ± 4.9 min, *n* = 33 cells, 17.1 ± 2.8 min, *n* = 76 cells, and 20.2 ± 4.9 min, *n* = 28 cells, respectively, for the parental cell line, cells expressing mScarlet-cenexin, and cells expressing mScarlet-cenexin S796A. P values = 0.30 (ns) and 0.45 (ns) for mScarlet-cenexin and mScarlet-cenexin S796A cells, respectively. A Kruskal–Wallis test with Dunn's multiple comparisons was used. ns = not significant. **(C)** Immunofluorescence images of 2:2 and 1:1 RPE1 GFP-centrin1 metaphase cells stained with DAPI (blue), GFP-centrin1 (green), and cenexin antibody (yellow). Scale bars = 10 μm. Examples of 2:2 and 1:1 cells displaying the brightest centrin signal (white arrowheads) located either at the cenexin-positive centriole or at the cenexin-negative centriole. **(D)** Quantification of the distribution of the brightest GFP-centrin1 signal with respect to cenexin localization. The brightest centrin signal is correlated with cenexin in only 63.6% (*n* = 132 cells) and 64.6% (*n* = 129 cells) in 2:2 and 1:1 cells, respectively.

Video 1.  **Live cell imaging movie of an RPE1 cell expressing mScarletCenexin (yellow) and GFP-Centrin1 (green) stained with SiR-DNA (blue) as it approaches anaphase.** The movie on the right shows the same cell but with a modified contrast in GFP-centrin1 channel; this reveals the cytoplasmic pool of GFP-centrin1, allowing us to visualize the cell cortex. Numbers indicate the time in minutes versus anaphase onset. Play rate = 2 frames per second.

