## [Peer Review File · The Journal of Cell Biology]

Centrosome age breaks spindle size symmetry even in cells thought to divide symmetrically

Alexandre Thomas and Patrick Meraldi

Corresponding Author(s): Patrick Meraldi, University of Geneva

Review Timeline:

Submission Date:	2023-11-27
Editorial Decision:	2023-12-12
Revision Received:	2024-03-14
Editorial Decision:	2024-04-11
Revision Received:	2024-05-01

Monitoring Editor: Monica Bettencourt-Dias

Scientific Editor: Tim Fessenden

Transaction Report:

DOI: <https://doi.org/10.1083/jcb.202311153>

Revision 0

Review #1

1. Evidence, reproducibility and clarity:

Evidence, reproducibility and clarity (Required)

In this paper, the authors demonstrated that there is asymmetry in mitotic spindles, which are usually considered symmetric. That is, they found that centrosome age causes asymmetry in the size of the half-spindle even when the number of centrioles forming the spindle poles is the usual pair or when there is only one mother centriole. It is also suggested that the difference in half-spindle size is due to the different microtubule-organizing activity of the centrosomes at each spindle pole. Furthermore, they observe that the difference in half-spindle size also results in asymmetries in the size of the daughter cells after cell division. In this study, they mainly analyze the mechanism by employing the condition of 1:1 number of centrioles, in which the difference in half-spindle size is more sharply pronounced. They showed that the subdistal appendage (SDA) of the centriole of the old centrosome is important for the molecular basis of this half-spindle size difference, as the SDA-dependent recruitment of the Plk1 pool mediates an asymmetric localization of pericentrin, Cdk5rap2, gamma-tubulin, TPX2, and other factors at the spindle poles. In addition, knockdown of these factors eliminated the half-spindle size asymmetry. They also confirm these findings using a different human cell line, BJ cells. In conclusion, they propose that, reflecting centrosome age, the old centrosome promotes asymmetric spindle formation by localizing a group of factors that promote microtubule organization, originating from the SDA-Plk1 pathway.

****Major points****

1. The discovery of differences in half-spindle size during symmetric division is intriguing. However, the methodology for quantification of the data remains unclear. Key questions, such as how the center of the metaphase plate is determined from the image data, the definition of exact pole position when centrioles are located at spindle poles, the objective determination of daughter cell diameter and width from the image data, and the referential position of the cortex, need more detailed explanation in the manuscript. Additionally, it's crucial to elucidate the specific index used to quantify differences from the image data, especially when dealing with data that only varies by a few percent. Providing clarity on these aspects and, in some cases, re-quantifying the data should be necessary.
2. The mechanism behind the difference in half-spindle size, related to the subdistal appendage (SDA), raises questions, especially considering that SDA is believed to disassemble during mitosis. Exploring whether differences in the localization of PCM components and half-spindle size result from disparities in Plk1 and PCM loading during G2/early mitosis, prior to SDA disassembly, necessitates experimental verification.
3. For investigating the mechanism of half-spindle size asymmetry, many perturbation experiments employ knock-down techniques. To directly address the cause of asymmetry, it

might be valuable to artificially localize Plk1 and PCM factors to one spindle pole using optogenetic tools or similar approaches and then quantify half-spindle and daughter cell sizes.

4. The asymmetry in Plk1 sub-population recruitment by SDA triggers the observed effects, but the evidence for this is relatively weak, given the small difference in spindle asymmetry.

Quantifying the amount of Plk1 in its activated form, particularly in the context of SDA dismantling during metaphase, could strengthen this aspect of the study.

5. While the focus on half-spindle size asymmetry during symmetric division is intriguing, it's important to address the broader physiological significance. The primary outcome of this asymmetry is differences in daughter cell size, which limits the broader significance of the study. Furthermore, the quantification method for daughter cell size warrants scrutiny and clarification.

****Minor points****

1. Table 1 lists factors with asymmetric localization not analyzed in detail in this paper. It would be beneficial to discuss whether these factors play a role in spindle asymmetry, and the authors should address the completeness of the data in Table 1 in terms of selecting factors for analysis.

2. In Figure 1H, the impact of centriolin knock-out on the distribution of unaligned polar chromosomes is different from the effect of cenexin S796A in Figure 6H. This difference should be explained to provide clarity on the observed discrepancies.

3. In Figure 2A, there is no correlation data presented between daughter cell asymmetry and the presence or absence of cenexin signal. This relationship should be elucidated for a more comprehensive understanding.

4. In Figure 4G and H, the mean value of spindle asymmetry increases with siRNA treatment of Cdk5Rap2 or PCNT compared to the control. The possible interpretation of this finding should be discussed.

5. Figure 4K shows that the asymmetry of PCNT distribution is not eliminated by centriolin knock-down. This observation requires clarification and discussion.

6. It appears that the difference in spindle asymmetry of the control group in Figure 5A is smaller than in other data. This discrepancy should be addressed. Additionally, the influence of TPX2 depletion on spindle formation, and any corresponding spindle staining data, should be included.

7. Claiming that the daughter centriole recruits PCM based on Figure 6A data alone may require additional supporting evidence. It is essential to investigate whether there is a clear PCM signal when the daughter centriole disengages in late mitosis and maintain consistency in the interpretation.

8. The lack of difference in TPX2 distribution in Figure 7E should be explained, along with a discussion of how this observation aligns with the spindle asymmetry data and any inconsistencies.

9. The differing N numbers between samples in all the figures may affect the validity of comparisons. The authors should discuss whether it is necessary to have consistent N numbers in each experiment for more robust conclusions.

2. Significance:

Significance (Required)

In summary, while the study is intriguing for its exploration of spindle asymmetry during symmetric division, the major points raised here highlight areas where further clarification and data interpretation are needed. A more rigorous quantification method and additional evidence to support the proposed SDA-Plk1 signal as the initiator of asymmetry would enhance the study's validity. Moreover, addressing concerns about daughter cell size quantification and the physiological relevance of spindle asymmetry is essential for a more comprehensive understanding of the findings. This research presents an interesting challenge for researchers in the centrosome and mitotic spindle field.

3. How much time do you estimate the authors will need to complete the suggested revisions:

Estimated time to Complete Revisions (Required)

(Decision Recommendation)

Between 3 and 6 months

Yes

Review #2

1. Evidence, reproducibility and clarity:

Evidence, reproducibility and clarity (Required)

****Summary:****

In this study, the authors show that in two types of human tissue culture cells the half spindles associated with mother centrosomes are slightly longer, on average, than the half spindles

associated with daughter centrioles. They show that this correlates with centrosome age, with mother centrosomes tending to be associated with the longer half spindle, and with a correlative asymmetry in the size of daughter cells. They show that spindle asymmetry relates to asymmetries in the amount of certain PCM components at centrosomes, including Pericentrin, CDK5RAP2, TPX2, and γ -tubulin, which preferentially accumulate at mother centrosomes. Pericentrin/CDK5RAP2/TPX2/ γ -tubulin are known to be involved, directly or indirectly, with microtubule nucleation, and the authors also show how that microtubule nucleation is more robust at mother centrosomes and that depletion of either Pericentrin, CDK5RAP2, TPX2, or γ -tubulin abolishes (or reduces) spindle asymmetry. The suggestion is that enhanced microtubule nucleation at the mother centrosome leads to longer half spindles and subsequent asymmetric positioning of the division plane and daughter cells of unequal size. Centrosome and spindle asymmetry is partially masked by the apparent equal accumulation of PCM at daughter centrioles, such that cells with centrosomes containing only mother centrioles show higher levels of asymmetry.

Mechanistically, the authors show that a Cenexin-bound pool of Plk1, a kinase required for PCM assembly, is important for centrosome and spindle asymmetry. Cenexin is an "upstream" sub-distal appendage protein only found at mother centrosomes (due to appendage structures only being present on the grandmother centriole). Nevertheless, depletion of a more downstream sub-distal appendage protein, Centriolin, also abrogated spindle asymmetry, suggesting that multiple proteins of the sub-distal appendages are necessary for asymmetry. Results from some experiments show that spindle asymmetry and a known asymmetry in the distribution of polar centrosomes are mechanistically separable, while other experiments show a link.

****Major comments:****

1. It is not completely clear how the authors determined whether a spindle was asymmetric or not. In the methods, they say that statistical tests are described in the legends. In Figure 1 legend they say: "Each condition was compared to a theoretical distribution centered at 0 (dashed line)". How did they generate this theoretical distribution?
2. The authors claim that TPX2 depletion results in loss of spindle asymmetry in 1:1 cells, but the difference is very small (1.7% in control vs 1.3% in TPX2 depletion, Fig 5B) and the data is more variable in TPX2 depletion, which makes it less likely that a statistically significant difference from 0 would be found. Firstly, perhaps the authors could check the standard error of the mean, which provides a measure of how accurate the mean is with regard to N and variation. If a dataset is more spread (such as in TPX2 depletion) a higher N is required to attain the same accuracy in the mean value. This is normally not so important when directly comparing two datasets, but in this case the authors are comparing each dataset to 0. So, are the authors measuring enough cells in the TPX2 depletion to be sure that a 1.3% value is not significantly different from 0? Secondly, I don't understand why the control cells have such a low asymmetry index (1.7%), when previous data in the paper shows an asymmetry index of 4.1% (Fig 1D) and 3.4% (Fig 4E) in control 1:1 cells. This suggests that something about the way this experiment was carried out dampens the asymmetry, which could therefore lead the authors to conclude that TPX2 is more important than it really is.

3. The authors claim that daughter centrioles are associated with some Pericentrin and suggest that this may be why 2:2 centrosomes have less of an asymmetry than 1:1 centrosomes (Fig 6A). It is unclear whether the authors consider these daughter centrioles as being prematurely disengaged (they make reference to the fact that they previously showed how disengaged daughters recruit γ -tubulin, but it's unclear if this is related to their current observations). In Figure 6A, the Centrin spots look too far apart for engaged centrioles (~750nm). I appreciate that this may be the only way to detect Pericentrin around the daughter at this resolution, but it may also force the authors to select cells where the centrioles have prematurely disengaged. For the asymmetry measurements, the authors presumably did not select cells where they could distinguish mother and daughter centrioles. One way to address this issue would be to compare PCM size at centrosomes in 2:2 cells with centrosomes in 1:1 cells. The expectation would be that centrosomes in 2:2 cells would have more PCM, due to the contribution of the daughter centrioles.
4. The authors show that Plk1 recruitment by Cenexin (via S796 phosphorylation), which happens only at mother centrosomes, is important for asymmetry. Nevertheless, they show that Plk1 is symmetrically distributed between mother and daughter centrosomes (Table 1). This does not really fit, unless daughter centrosomes recruit more cenexin-independent Plk1 than mother centrosomes or if the cenexin-bound pool of Plk1 is only a minor fraction of total Plk1. If so, do the authors think that the Cenexin-bound pool of Plk1 is more potent than the rest of centrosomal Plk1?
5. The circles drawn to measure cell size in Figures 2A,E and 7C do not look like a good representation of cell area (as the cells are not perfectly round). The authors use a formula for circle area with an approximation of the radius (based on mean length/width of an oval). It would be much better to use ImageJ to draw a freehand line around the perimeter of the cell and use the in-built tool to measure the area.

****Minor comments:****

1. Asymmetry in centrosome size that correlates with centrosome age in apparently symmetrically dividing "cells" has been observed previously in *Drosophila* syncytial embryos (Conduit et al., 2010a, Curr. Bio.). I think this should be mentioned somewhere given the topic of the study.
2. A full description of statistical tests and n numbers for each experiment should be provided in the methods, even if this duplicates information in the Figure legends.

OPTIONAL EXPERIMENTS:

3. Given that chTOG is very important for microtubule nucleation, it seems strange that this protein was not analysed for a potential asymmetry.
4. Cooling-warming experiments could be done using higher concentration of formaldehyde, as it's likely that microtubule nucleation is not immediately halted when using 4% formaldehyde.

2. Significance:

Significance (Required)

This a well-conducted study with results being presented clearly and concisely. The methodology is solid in the main. The study reveals something unexpected - that apparently symmetrically dividing human tissue culture cells divide asymmetrically. While the asymmetry is only slight, it could be important - although the authors do not address its relevance for the cell population. Having analysed only 2 cultured cell types, it remains unclear if this is a widespread phenomenon, and whether this occurs in a more natural setting. Nevertheless, the proposed model (Plk1 at SDA's => increased PCM at mother => increased nucleation => offset division plane), which is supported by the data, would suggest this could be a widespread phenomenon. This study will be of interest to anyone studying cell division, but it would require some degree of insight into the importance of the observations for it to appeal to a very broad audience.

I am a cell biologist with an interest in cell division and microtubule regulation.

3. How much time do you estimate the authors will need to complete the suggested revisions:

Estimated time to Complete Revisions (Required)

(Decision Recommendation)

Between 1 and 3 months

Yes

Review #3

1. Evidence, reproducibility and clarity:

Evidence, reproducibility and clarity (Required)

The manuscript entitled "centrosome age breaks spindle size symmetry even in "symmetrically" dividing cells" by Thomas and Meraldi reports that centrosome age impacts microtubule-nucleation capacity and is sufficient to tune spindle symmetry and cell size in human culture cell lines. The manuscript is overall clear, well written, illustrated and discussed. Nonetheless, some key experiments are missing as the authors report very subtle differences that need to be confirmed with complementary experiments, including time-lapse microscopy and alternative evaluations of cell sizes. The mechanism by which spindle symmetry breaking is established by centrosome age is not clear, even if the authors have identified some important actors at the spindle poles.

Major points:

1. The evaluation of spindle and cell size asymmetry related to centrosome age only relies on fixed sample preparation. Cells should be followed by time-lapse microscopy as the metaphase plate position relative to the spindle poles and/or the cell cortex may fluctuate over time and as the observed differences remain in a very subtle range. This is an important possibility to consider for 1:1, 1:0 or 0:0 spindle pole configurations where centrosome integrity is impaired.
2. Cell size asymmetry was evaluated based on cell area at the equator. Volumes will be a better indicator as daughter cell shapes can be different in telophase if they do not re-adhere at the same speed. This evaluation should also be confirmed with another readout, like the position of the cleavage furrow relative to the spindle poles in late anaphase, as again the observed differences are in a very subtle range.
3. The authors propose that differential microtubule nucleation at the spindle poles underlies spindle size symmetry breaking without providing direct evidence. If the observed spindle symmetry in the 1:1 configuration after pericentrin, CDK5RAP2 or γ -tubulin siRNA fuels this interpretation (Fig4C), the differential microtubule nucleation capacity at the spindle poles after microtubule-depolymerisation-repolymerisation assays was not evaluated in these conditions, as compared to the control situation.
4. If differential microtubule nucleation at the spindle poles is responsible for spindle asymmetry, overexpression of PCM proteins or γ -tubulin should be sufficient for re-establishment of symmetric protein distribution, spindle and cell size symmetry in 2:2 or 1:1 configuration. The authors should evaluate whether this is the case or not.
5. The authors describe that the cortex-centrosome distance is not changed according to centrosome age (Fig2C), but centrosome-metaphase plate distance is (Fig1D). These observations are difficult to reconcile if differential microtubule-nucleation capacity is at play. Again, time-lapse microscopy would enable to detect over time whether only metaphase plate position relative to spindle poles is changing or if spindle pole position relative to the cell cortex is also fluctuating.

Minor points:

6. Main PCM and MT nucleation protein "depletion" do not appear to impact spindle assembly, but only spindle symmetry in 1:1 and 1:0 configurations (Fig4A and 4F-H). Can it be explained

by the fact that their depletion is not always total (for pericentrin, Fig5F versus FigS2A or Fig7G)? Can they comment on this point?

7. If centrosome age dictates spindle and cell size asymmetry through differential MT-nucleation capacity at the spindle poles, how can this process be modulated? Indeed, centrosome age is common to all cell types, but cell size asymmetry is more or less pronounced. The authors should further discuss this point based on the literature.

2. Significance:

Significance (Required)

The question of whether centrosome age is translated into different capacity to nucleate microtubules and related consequences on spindle and cell size symmetry has already been addressed in different model systems. Nonetheless, cell lines were previously described as dividing symmetrically since their spindle is symmetric in size and since they give rise to daughter cells of equivalent sizes. The present manuscript reports a thorough re-evaluation of this question and provides evidence that subtle differences in PCM and spindle pole protein recruitment, microtubule-nucleation capacity and spindle symmetry can be observed as a function of centrosome age. They also identify some key actors whose differential recruitment at the spindle poles can underlie spindle symmetry breaking, even if their involvement seems to differ from one cell line to another one. This manuscript could be submitted after appropriate revisions as a report and will benefit to the basic research cell biology community.

3. How much time do you estimate the authors will need to complete the suggested revisions:

Estimated time to Complete Revisions (Required)

(Decision Recommendation)

Between 1 and 3 months

Yes

UNIVERSITÉ
DE GENÈVE

FACULTÉ DE MÉDECINE

DÉPARTEMENT DE PHYSIOLOGIE CELLULAIRE
ET METABOLISME

Prof. Patrick Meraldi
Physiology and Metabolism Department
Medical Faculty, University of Geneva
Rue Michel Servet 1, 1211 Geneva 4
Switzerland
Mail : Patrick.meraldi@unige.ch

Wednesday, 24th of November 2023

Dear Sara Monaco,

We would like to thank the reviewers for their time and their constructive feedback on our manuscript. Their insightful comments will help us to improve the quality of our study. Please find below a plan of the future experiments we will perform according to reviewers' suggestions. We also provide in some cases a pre-rebuttal, when we think that the proposed experiments are not feasible in a reasonable time or impossible.

The main reviewers' concern is the methods we used to measure half-spindle size and daughter cell size asymmetries. For the half-spindle size asymmetry measurements, we will provide clear explanations in the method section and a supplementary figure. Specifically, we used the GFP-centrin signal (or a spindle pole protein in centriole-free spindle poles) to determine the 3D localization of the spindle poles using the point detection function in IMARIS. For the center of the metaphase plate, the center of the DAPI signal was determined with the surface function in IMARIS (as used in Dudka et al., 2019). The half-spindle lengths were extracted with IMARIS in 3D as the distance between the spindle poles and the center of the metaphase plate.

For the daughter cell size asymmetry, we measured the length and width in late telophase of cells displaying a spherical shape. To overcome potential bias due to a difference in re-adherence and therefore different shapes, we propose to use an alternative method and to measure the real volume of the two daughter cells.

Among the main experiments, we also plan to quantify PCM protein and Plk1 at both old and young centrosomes in G2 before sub-distal appendage reorganization occurs. However, due to the absence of a good antibody recognizing Plk1-Thr210-phosphorylated at the cellular level, we will not be able to assess the distribution of the activated form of Plk1 between the two poles.

We think that 3-4 months will be necessary to address those points and to complete the revision. By providing these new results, we hope to fully convince the reviewers of the novelty and relevance of our findings.

Kind regards,

Patrick Meraldi, Geneva

1) List of the detailed experiments we plan to perform (including aforementioned experiments):

- Careful analysis of the daughter cell size by measuring the real volume.
- Quantifications of PCM (pericentrin and γ -tubulin) proteins and Plk1 with respect to centrosome age in G2 and metaphase (for Plk1) cells.
- Analysis of the amount of Plk1 of metaphase cells when cenexin protein is absent (siControl vs siCenexin), and measurements of Plk1 in WT-cenexin vs. cenexinS796A mutant to test if Cenexin controls a subpool of Plk1 at centrosomes.
- Careful analysis of Ctrl and TPX2 depletion experiment data in 1:1 cells. We plan to repeat the experiment to confirm or infirm on the contribution of TPX2 in spindle asymmetry.
- Measurement of the PCM volume/intensity in 2:2 and 1:1 metaphase cells, to highlight on the contribution of the daughter centrioles in recruiting PCM proteins.
- Live cell imaging of 2:2 cells and measurements of different parameters; cortex-to-centrosome and spindle pole to metaphase plate (half-spindle (a)symmetry) distances.
- Long-term live cell imaging of 2:2 cells to investigate whether the asymmetry in centrosome-age dependent daughter cell size also affects the duration of the ensuing cell cycle. While we have carried out such long-term movies in the past, we are aware that they can be challenging due to high cell mobility over longer time courses.
- Investigation of the microtubule nucleation capacity under different conditions of PCM protein depletion (depletion of Cdk5rap2 and/or pericentrin).
- Analysis of the effect of the over-expression of PCM protein (Cdk5Rap2) on the (a)symmetry of the mitotic spindle size

2) detailed answers (in green) to the reviewers' comments:

Reviewer #1: (Major points)

1. The discovery of differences in half-spindle size during symmetric division is intriguing. However, the methodology for quantification of the data remains unclear. Key questions, such as how the center of the metaphase plate is determined from the image data, the definition of exact pole position when centrioles are located at spindle poles, the objective determination of daughter cell diameter and width from the image data, and the referential position of the cortex, need more detailed explanation in the manuscript. Additionally, it's crucial to elucidate the specific index used to quantify differences from the image data, especially when dealing with data that only varies by a few percent. Providing clarity on these aspects and, in some cases, re-quantifying the data should be necessary.

We have already included clearer explanations in the method parts and results part about our methodology and will include a supplementary figure on how precisely we defined and measured the half-spindle sizes, as well as the index used for the asymmetry (using a methodology that we previously used in Dudka et al., Nature Comm., 2018). In addition, we will use a second method to measure the real daughter cell volume.

2. The mechanism behind the difference in half-spindle size, related to the subdistal appendage (SDA), raises questions, especially considering that SDA is believed to disassemble during mitosis. Exploring whether differences in the localization of PCM components and half-spindle size result from disparities in Plk1 and PCM loading during G2/early mitosis, prior to SDA disassembly, necessitates experimental verification.

As suggested by the reviewer we will quantify the amounts of PCM proteins on the old and young centrosome in G2 cells (and therefore prior SDA reorganization). This will also allow us to test whether the asymmetry depends on the SDA themselves, or the corresponding SDA proteins, which still accumulate specifically on the oldest centrosomes during mitosis

3. For investigating the mechanism of half-spindle size asymmetry, many perturbation experiments employ knock-down techniques. To directly address the cause of asymmetry, it might be valuable to artificially localize Plk1 and PCM factors to one spindle pole using optogenetic tools or similar approaches and then quantify half-spindle and daughter cell sizes.

We thank the reviewers for this suggestion, as it could indeed, be of great interest and provide a direct proof of principle. Unfortunately, based on our experience in establishing such a cell line we know that just the generation of such a light-manipulated stable cell line that contains markers for centrosomes and chromosomes or kinetochores takes 6-9 months, in the best-case scenario. This experiment is therefore not possible within a normal revision round (even if extended to 6 months).

4. The asymmetry in Plk1 sub-population recruitment by SDA triggers the observed effects, but the evidence for this is relatively weak, given the small difference in spindle asymmetry. Quantifying the amount of Plk1 in its activated form, particularly in the context of SDA dismantling during metaphase, could strengthen this aspect of the study.

While the commercial antibodies against the activated form of Plk1 (phospho-T210) work very well by immunoblotting, we have not been able to get it to work by immunofluorescence. We will nevertheless, test whether variation in the fixation methods can solve this issue. Alternatively, we will test to which extent depletion of Cenexin, or the presence of Cenexin WT vs the non-phosphorylatable Cenexin mutant affects the overall population of Plk1 on both spindle poles.

5. While the focus on half-spindle size asymmetry during symmetric division is intriguing, it's important to address the broader physiological significance. The primary outcome of this asymmetry is differences in daughter cell size, which limits the broader significance of the study. Furthermore, the quantification method for daughter cell size warrants scrutiny and clarification.

As mentioned above, we will use different method to measure and investigate daughter cell size (a)symmetry. Moreover, we will attempt with long-term live cell movies to test whether the variation in centrosome-age dependent daughter cell size also affects the duration of the ensuing cell cycle.

(Minor points)

1. Table 1 lists factors with asymmetric localization not analyzed in detail in this paper. It would be beneficial to discuss whether these factors play a role in spindle asymmetry, and the authors should address the completeness of the data in Table 1 in terms of selecting factors for analysis.

We agree with this comment that other factors may participate in the regulation of spindle asymmetry. However, we performed this screening to identify key drivers of spindle (a)symmetry based on an investigation of the Pearson's correlation coefficient and the value of slope.

In addition, some of these proteins are known to control spindle size in acting in a same pathway (TPX2/Kif2A/Katanin) and (Pericentrin/CDK5RAP2/Y-tubulin). We will incorporate these points and the reasons for our selection in the discussion

2. In Figure 1H, the impact of centriolin knock-out on the distribution of unaligned polar chromosomes is different from the effect of cenexin S796A in Figure 6H. This difference should be explained to provide clarity on the observed discrepancies.

We will better explain this difference.

3. In Figure 2A, there is no correlation data presented between daughter cell asymmetry and the presence or absence of cenexin signal. This relationship should be elucidated for a more comprehensive understanding.

We will clarify this point. Specifically, we plotted the daughter cell symmetry index for 2:2 and 1:1 cells with respect to centrosome age. All the daughter cells display the presence of a cenexin signal

at both grandmother and mother centrioles with a difference in fluorescence intensity that enables us to assign them to “old” vs “young centrosomes. We found a significant result indicating that there is a relationship between centrosome age and the formation of daughter cell with different sizes.

4. In Figure 4G and H, the mean value of spindle asymmetry increases with siRNA treatment of Cdk5Rap2 or PCNT compared to the control. The possible interpretation of this finding should be discussed.

This is an interesting observation that needs to be discussed in our revision.

5. Figure 4K shows that the asymmetry of PCNT distribution is not eliminated by centriolin knock-down. This observation requires clarification and discussion.

It has been shown that pericentrin is directly recruited by Plk1 at centriole (Soung et al., 2009). In addition, pericentrin has a PACT-domain that directly targets pericentrin to the centriole (Gillingham and Munro., 2000). Moreover, it has been demonstrated that the grandmother centriole is slightly longer than the mother one (Kong et al., 2020). Altogether, this suggests that the old and young centrosomes, based on this intrinsic property, may recruit different amount of pericentrin.

We will add this explanation in the discussion.

6. It appears that the difference in spindle asymmetry of the control group in Figure 5A is smaller than in other data. This discrepancy should be addressed. Additionally, the influence of TPX2 depletion on spindle formation, and any corresponding spindle staining data, should be included.

This point will be discussed in the revised version of the manuscript.

7. Claiming that the daughter centriole recruits PCM based on Figure 6A data alone may require additional supporting evidence. It is essential to investigate whether there is a clear PCM signal when the daughter centriole disengages in late mitosis and maintain consistency in the interpretation.

As suggested by the reviewer 2, we will measure PCM volume/intensity in both 2:2 and 1:1 cells to demonstrate that daughter centrioles directly recruit PCM proteins.

8. The lack of difference in TPX2 distribution in Figure 7E should be explained, along with a discussion of how this observation aligns with the spindle asymmetry data and any inconsistencies.

We will discuss this point in the revised manuscript.

9. The differing N numbers between samples in all the figures may affect the validity of comparisons. The authors should discuss whether it is necessary to have consistent N numbers in each experiment for more robust conclusions.

Indeed, this is an important point that must be discussed.

Reviewer #2:

Major comments:

1) It is not completely clear how the authors determined whether a spindle was asymmetric or not. In the methods, they say that statistical tests are described in the legends. In Figure 1 legend they say: "Each condition was compared to a theoretical distribution centered at 0 (dashed line)". How did they generate this theoretical distribution?

As explained under point 1 of reviewer 1, we will provide a more thorough explanation of our methodology and how we decide whether a spindle is symmetric or not. In brief, a perfectly symmetric spindle would yield an asymmetry index of 0, as there is no difference between the two half-spindle sizes.

2) The authors claim that TPX2 depletion results in loss of spindle asymmetry in 1:1 cells, but the difference is very small (1.7% in control vs 1.3% in TPX2 depletion, Fig 5B) and the data is more variable in TPX2 depletion, which makes it less likely that a statistically significant difference from 0 would be found. Firstly, perhaps the authors could check the standard error of the mean, which

provides a measure of how accurate the mean is with regard to N and variation. If a dataset is more spread (such as in TPX2 depletion) a higher N is required to attain the same accuracy in the mean value. This is normally not so important when directly comparing two datasets, but in this case the authors are comparing each dataset to 0. So, are the authors measuring enough cells in the TPX2 depletion to be sure that a 1.3% value is not significantly different from 0? Secondly, I don't understand why the control cells have such a low asymmetry index (1.7%), when previous data in the paper shows an asymmetry index of 4.1% (Fig 1D) and 3.4% (Fig 4E) in control 1:1 cells. This suggests that something about the way this experiment was carried out dampens the asymmetry, which could therefore lead the authors to conclude that TPX2 is more important than it really is.

We agree with this comment, the mean of the control condition is smaller compared to others controls. As mentioned above, we will carefully look at the data (SD vs SEM) and in case add a new replicate to confirm or infirm the involvement of TPX2 in the formation of asymmetric spindles.

3) The authors claim that daughter centrioles are associated with some Pericentrin and suggest that this may be why 2:2 centrosomes have less of an asymmetry than 1:1 centrosomes (Fig 6A). It is unclear whether the authors consider these daughter centrioles as being prematurely disengaged (they make reference to the fact that they previously showed how disengaged daughters recruit γ -tubulin, but it's unclear if this is related to their current observations). In Figure 6A, the Centrin spots look too far apart for engaged centrioles (~750nm). I appreciate that this may be the only way to detect Pericentrin around the daughter at this resolution, but it may also force the authors to select cells where the centrioles have prematurely disengaged. For the asymmetry measurements, the authors presumably did not select cells where they could distinguish mother and daughter centrioles. One way to address this issue would be to compare PCM size at centrosomes in 2:2 cells with centrosomes in 1:1 cells. The expectation would be that centrosomes in 2:2 cells would have more PCM, due to the contribution of the daughter centrioles.

We agree that on those high-resolution images the daughter centrioles seem to be far from the mother ones. The metaphase cells presented in this figure, are wild-type non-treated cells for which the daughter centrioles are engaged. Indeed, our own investigation of the centriole engagement status by expansion microscopy, indicates that over 98% of centriole pairs in metaphase RPE1 cells are engaged.

Nevertheless, as suggested by the reviewer and to validate that daughter centrioles participate in this process, we will compare PCM size in 2:2 and 1:1 metaphase cells.

4) The authors show that Plk1 recruitment by Cenexin (via S796 phosphorylation), which happens only at mother centrosomes, is important for asymmetry. Nevertheless, they show that Plk1 is symmetrically distributed between mother and daughter centrosomes (Table 1). This does not really fit, unless daughter centrosomes recruit more cenexin-independent Plk1 than mother centrosomes or if the cenexin-bound pool of Plk1 is only a minor fraction of total Plk1. If so, do the authors think that the Cenexin-bound pool of Plk1 is more potent than the rest of centrosomal Plk1?

As indicated in point 4 of reviewer 1 we will test which proportion of the Plk1 pool at spindle poles depends on the presence of Cenexin, as we suspect that this Plk1 population is only a subpopulation.

5) The circles drawn to measure cell size in Figures 2A,E and 7C do not look like a good representation of cell area (as the cells are not perfectly round). The authors use a formula for circle area with an approximation of the radius (based on mean length/width of an oval). It would be much better to use ImageJ to draw a freehand line around the perimeter of the cell and use the in-built tool to measure the area.

As mentioned in point 1 of reviewer 1 we will use another method to measure daughter cell size.

Minor comments:

1) Asymmetry in centrosome size that correlates with centrosome age in apparently symmetrically

dividing "cells" has been observed previously in *Drosophila* syncytial embryos (Conduit et al., 2010a, Curr. Bio.). I think this should be mentioned somewhere given the topic of the study.

We thank the reviewer for this information. This paper will be discussed in the revised version.

2) A full description of statistical tests and n numbers for each experiment should be provided in the methods, even if this duplicates information in the Figure legends.

We will add this information in the method.

OPTIONAL EXPERIMENTS:

3) Given that chTOG is very important for microtubule nucleation, it seems strange that this protein was not analysed for a potential asymmetry.

As suggested by the reviewers we will test for a potential chTOG asymmetry and its impact on spindle size asymmetry.

4) Cooling-warming experiments could be done using higher concentration of formaldehyde, as it's likely that microtubule nucleation is not immediately halted when using 4% formaldehyde.

The fixation solution was chilled at 4°C, which should halt any further depolymerization. We will specify this point in the Material and Methods section.

Reviewer #3:

Major points:

1) The evaluation of spindle and cell size asymmetry related to centrosome age only relies on fixed sample preparation. Cells should be followed by time-lapse microscopy as the metaphase plate position relative to the spindle poles and/or the cell cortex may fluctuate over time and as the observed differences remain in a very subtle range. This is an important possibility to consider for 1:1, 1:0 or 0:0 spindle pole configurations where centrosome integrity is impaired.

We agree with the reviewer that this is a drawback of our approach, but the experiments the reviewer suggests is not possible for 1:0 or 0:0 or only in an approximate manner. Indeed, we do not have a centriole-independent spindle pole marker that would allow us to mark precisely the position of the spindle pole. In the past we used Sir-tubulin, which gave us an approximate position of the spindle poles, and which allowed to us monitor the spindle asymmetry over time of 1:0 cells (see Dudka et al., 2019), a point that we will discuss. Nevertheless, as suggested by the reviewer we will attempt to monitor these asymmetries in 2:2 and/or 1:1 cells expressing GFP-Centrin1 and GFP-CENPA (kinetochore marker) in WT conditions. Indeed, we cannot expand this approach to all the conditions, as the calculation of the spindle asymmetry index is based on a very high number of cells, and the monitoring of spindle asymmetry can only be achieved by selecting mitotic cells one-by-one and then monitoring them over a short period of them (Tan et al., eLife, 2015), which makes such an approach extremely time-consuming.

2) Cell size asymmetry was evaluated based on cell area at the equator. Volumes will be a better indicator as daughter cell shapes can be different in telophase if they do not re-adhere at the same speed. This evaluation should also be confirmed with another readout, like the position of the cleavage furrow relative to the spindle poles in late anaphase, as again the observed differences are in a very subtle range.

As indicated in the similar points of reviewer 1 and 2, we will improve our methodology to take this comment in account

3) The authors propose that differential microtubule nucleation at the spindle poles underlies spindle size symmetry breaking without providing direct evidence. If the observed spindle symmetry in the 1:1 configuration after pericentrin, CDK5RAP2 or γ -tubulin siRNA fuels this interpretation (Fig4C), the differential microtubule nucleation capacity at the spindle poles after microtubule-

depolymerisation-repolymerisation assays was not evaluated in these conditions, as compared to the control situation.

As suggested by the reviewer we will analyze the microtubule nucleation capacity after the downregulation of PCM proteins.

4) If differential microtubule nucleation at the spindle poles is responsible for spindle asymmetry, overexpression of PCM proteins or γ -tubulin should be sufficient for re-establishment of symmetric protein distribution, spindle and cell size symmetry in 2:2 or 1:1 configuration. The authors should evaluate whether this is the case or not.

This is an interesting suggestion, which we will test, although overexpression of these proteins might also lead to other defects in the spindle, such as multipolar spindles.

5) The authors describe that the cortex-centrosome distance is not changed according to centrosome age (Fig2C), but centrosome-metaphase plate distance is (Fig1D). These observations are difficult to reconcile if differential microtubule-nucleation capacity is at play. Again, time-lapse microscopy would enable to detect over time whether only metaphase plate position relative to spindle poles is changing or if spindle pole position relative to the cell cortex is also fluctuating.

We plan to give a try to image WT 2:2 cells by time lapse microscopy and to measure several parameters such as half-spindle size, spindle (a)symmetry and the cortex to centrosome distance over time.

Minor points:

6) Main PCM and MT nucleation protein "depletion" do not appear to impact spindle assembly, but only spindle symmetry in 1:1 and 1:0 configurations (Fig4A and 4F-H). Can it be explained by the fact that their depletion is not always total (for pericentrin, Fig5F versus FigS2A or Fig7G)? Can they comment on this point?

Spindles displaying abnormal centriole number at spindle poles (1:1 and 1:0) can still assemble bipolar spindle in absence of the main PCM proteins (Chinen et al., JCB, 2021, and Watanabe et al., JCB, 2020).

In our study, the depletion of PCM protein is almost total (97% for pericentrin, 98% for Cdk5Rap2).

7) If centrosome age dictates spindle and cell size asymmetry through differential MT-nucleation capacity at the spindle poles, how can this process be modulated? Indeed, centrosome age is common to all cell types, but cell size asymmetry is more or less pronounced. The authors should further discuss this point based on the literature.

We will discuss this point in the discussion.

3. Description of the revisions that we have already carried out in the revised manuscript

1. The discovery of differences in half-spindle size during symmetric division is intriguing. However, the methodology for quantification of the data remains unclear. Key questions, such as how the center of the metaphase plate is determined from the image data, the definition of exact pole position when centrioles are located at spindle poles, the objective determination of daughter cell diameter and width from the image data, and the referential position of the cortex, need more detailed explanation in the manuscript. Additionally, it's crucial to elucidate the specific index used to quantify differences from the image data, especially when dealing with data that only varies by a few percent. Providing clarity on these aspects and, in some cases, re-quantifying the data should be necessary.

We have already included clearer explanations in the method parts and results part about our methodology and will include a supplementary figure on how precisely we defined and measured the half-spindle sizes, as well as the index used for the asymmetry (using a methodology that we

previously used in Dudka et al., Nature Comm., 2018). In addition, we will use a second method to measure the real daughter cell volume.

4. Description of the experiments that we prefer not to carry out:

Point 3 of reviewer 1 : For investigating the mechanism of half-spindle size asymmetry, many perturbation experiments employ knock-down techniques. To directly address the cause of asymmetry, it might be valuable to artificially localize Plk1 and PCM factors to one spindle pole using optogenetic tools or similar approaches and then quantify half-spindle and daughter cell sizes.

We thank the reviewers for this suggestion, as it could indeed, be of great interest and provide a direct proof of principle. Unfortunately, based on our experience in establishing such a cell line we know that just the generation of such a light-manipulated stable cell line that contains markers for centrosomes and chromosomes or kinetochores takes 6-9 months, in the best-case scenario. This experiment is therefore not possible within a normal revision round (even if extended to 6 months).

December 12, 2023

Re: JCB manuscript #202311153T

Prof. Patrick Meraldi
University of Geneva
Cell physiology and metabolism department
Centre Medical Universitaire Rue Michel Servet 1
Geneva 1211
Switzerland

Dear Prof. Meraldi,

Thank you for submitting your manuscript entitled "Centrosome age breaks spindle size symmetry even in "symmetrically" dividing cells". The manuscript and reviews were assessed by at least two editors. By finding subtle but consistent effects of centriole age, this work refines the established concept that centrosome presence/absence alters spindle symmetry. The proposed mechanism offers details into PCM generation and function consistent with known pathways for spindle regulation, but with an intriguing new claim on the ramifications of centriole age. We invite you to submit a revised manuscript, and we agree with the additions proposed in the plan for revision you provided.

GENERAL GUIDELINES:

Text limits: Character count for an Transfer is < 40,000, not including spaces. Count includes title page, abstract, introduction, results, discussion, and acknowledgments. Count does not include materials and methods, figure legends, references, tables, or supplemental legends.

Figures: Transfers may have up to 10 main text figures. Figures must be prepared according to the policies outlined in our Instructions to Authors, under Data Presentation, <https://jcb.rupress.org/site/misc/ifora.xhtml>. All figures in accepted manuscripts will be screened prior to publication.

Supplemental information: There are strict limits on the allowable amount of supplemental data. Transfers may have up to 5 supplemental figures. Up to 10 supplemental videos or flash animations are allowed. A summary of all supplemental material should appear at the end of the Materials and methods section.

Please note that JCB now requires authors to submit Source Data used to generate figures containing gels and Western blots with all revised manuscripts. This Source Data consists of fully uncropped and unprocessed images for each gel/blot displayed in the main and supplemental figures. Since your paper includes cropped gel and/or blot images, please be sure to provide one Source Data file for each figure that contains gels and/or blots along with your revised manuscript files. File names for Source Data figures should be alphanumeric without any spaces or special characters (i.e., SourceDataF#, where F# refers to the associated main figure number or SourceDataFS# for those associated with Supplementary figures). The lanes of the gels/blots should be labeled as they are in the associated figure, the place where cropping was applied should be marked (with a box), and molecular weight/size standards should be labeled wherever possible.

The typical timeframe for revisions is three to four months. While most universities and institutes have reopened labs and allowed researchers to begin working at nearly pre-pandemic levels, we at JCB realize that the lingering effects of the COVID-19 pandemic may still be impacting some aspects of your work, including the acquisition of equipment and reagents. Therefore, if you anticipate any difficulties in meeting this aforementioned revision time limit, please contact us and we can work with you to

find an appropriate time frame for resubmission. Please note that papers are generally considered through only one revision cycle, so any revised manuscript will likely be either accepted or rejected.

Thank you for this interesting contribution to Journal of Cell Biology. You can contact us at the journal office with any questions at cellbio@rockefeller.edu.

Sincerely,

Monica Bettencourt-Dias
Monitoring Editor
Journal of Cell Biology

Tim Fessenden
Scientific Editor
Journal of Cell Biology

1) Synopsis of the main experiments we carried out

- We remeasured the daughter cell area using previously published approaches (Kiyomitsu and Cheeseman, 2013) and confirmed that daughter cells inheriting the old centrosomes have a larger size (**Figure 2D**, same is observed for BJ 2:2 cells, **Figure 9B**). This new analysis also confirms that *centriolin*^{-/-} cells divide symmetrically (**Figure 2H**).
- We quantified the levels of pericentrin and γ -tubulin in G2 and prophase on old and young centrosomes and find that pericentrin is enriched on the old centrosome already in G2 and that asymmetry persists throughout mitosis (**Figure 6A and B**). In contrast, γ -tubulin becomes asymmetric only upon mitotic entry (**Figure 6C and D**).
- We measured Plk1 abundance on old and young centrosomes at mitotic entry and in metaphase and found that Plk1 is enriched on old centrosomes (**Figure 6E and F**). Moreover, we show that this asymmetric distribution depends on the S796 phosphorylation site on cenexin (**Figure 7D and E**). Finally, we show that Plk1 activity is necessary for spindle size asymmetry (**Figure 6G-K**).
- Using a higher sample size, we confirm that spindle size asymmetry depends on TPX2 (**Figure 5A and B**).
- We also fully characterized the contribution of ch-TOG to this process, showing that it is enriched on the old centrosomes, that its presence is required for an asymmetric spindle size, and that it acts downstream of pericentrin (**Figure 5F-H**).
- We quantified PCM proteins (pericentrin and γ -tubulin) and PCM size in 2:2 vs. 1:1 cells and showed that daughter centrioles contribute to their recruitment to centrosomes (**Figure 8A-C**) and that PCM size is wider in 2:2 compared to 1:1 cells (**Figure 8D**).
- Using live cell imaging experiments, we validated our previous observations made in fixed cells in terms of spindle size asymmetry and cortex to centrosome distances (**Figure 2A and 2F**).
- We analyzed microtubule nucleation in the absence of Cdk5Rap and found that microtubule nucleation from centrosomes is very slow under such conditions and that microtubules originate from secondary MTOCs, such as chromatin and/or kinetochores (*siCdk5Rap2*, **Figure 4L**).

2) Point-by-point responses to the reviewers' comments:

We thank all three reviewers for the constructive feedback, which we addressed in the following manner (in blue).

Reviewer #1:

(Major points)

1. The discovery of differences in half-spindle size during symmetric division is intriguing. However, the methodology for quantification of the data remains unclear. Key questions, such as how the center of the metaphase plate is determined from the image data, the definition of exact pole position when centrioles are located at spindle poles, the objective determination of daughter cell diameter and width from the image data, and the referential position of the cortex, need more detailed explanation in the manuscript. Additionally, it's crucial to elucidate the specific index used to quantify differences from the image data, especially when dealing with data that only varies by a few percent. Providing clarity on these aspects and, in some cases, re-quantifying the data should be necessary.

We added clarifying explanations in the result and material & method sections explaining how we used the 3D image analysis software IMARIS to precisely determine the center of the spindle poles and the center of gravity of the DNA mass, which defines the position of the metaphase plate. We also specify in the results section and in **Supplementary Figure S1** how we calculate the spindle asymmetry index, a methodology that we have already used in the past (Dudka et al., Current Biol. 2019). Finally, we improved the method to calculate the daughter cell diameter and explain better our approach.

2. The mechanism behind the difference in half-spindle size, related to the subdistal appendage (SDA), raises questions, especially considering that SDA is believed to disassemble during mitosis. Exploring whether differences in the localization of PCM components and half-spindle size result from disparities in Plk1 and PCM loading during G2/early mitosis, prior to SDA disassembly, necessitates experimental verification.

We thank the reviewer for this excellent suggestion. We analyzed the distribution of pericentrin, γ -tubulin and Plk1 also prior to metaphase and found a very interesting, differentiated pattern. As shown in the novel **Figure 6**, pericentrin is already asymmetric in G2, γ -tubulin is initially symmetric, but becomes asymmetric at mitotic entry, while endogenous Plk1 is already asymmetric at the earliest timepoint (prophase) we can reliably quantify it on centrosomes by immunofluorescence. Overall, this indicates that centrosome age already imposes an asymmetry prior to mitotic entry, and that this asymmetry is expanded to γ -tubulin as cells enter mitosis. This propagation of symmetry most likely depends on Plk1, as we find that inhibition of this kinase abolishes the asymmetric distribution of the PCM proteins.

3. For investigating the mechanism of half-spindle size asymmetry, many perturbation experiments employ knock-down techniques. To directly address the cause of asymmetry, it might be valuable to artificially localize Plk1 and PCM factors to one spindle pole using optogenetic tools or similar approaches and then quantify half-spindle and daughter cell sizes.

We agree with the reviewers that such an approach is in principle of great interest to provide a more direct proof of principle. Unfortunately, based on our experience in establishing such a cell line, we know that already the generation of such a light-manipulated stable cell line that contains markers for centrosomes and chromosomes or kinetochores takes 6-9 months, in the best-case scenario. This experiment was therefore not possible within a normal revision round, even if extended to 6 months.

4. The asymmetry in Plk1 sub-population recruitment by SDA triggers the observed effects, but the evidence for this is relatively weak, given the small difference in spindle asymmetry. Quantifying the amount of Plk1 in its activated form, particularly in the context of SDA dismantling during metaphase, could strengthen this aspect of the study.

While the commercial antibodies against the activated form of Plk1 (phospho-T210) work very well by immunoblotting, they do not work in our hands by immunofluorescence, despite having tried multiple fixation methods. Instead, we quantified endogenous Plk1 levels on centrosomes in prophase and metaphase. First, we show that at both time points, Plk1 is asymmetrically enriched on the old centrosome, and that Plk1 activity drives the spindle size asymmetry (**Figure 6**). Moreover, by comparing cell lines expressing either WT-cenexin or the S796A cenexin, we show that the asymmetric distribution of Plk1 depends on S796 phosphorylation site, uncovering a direct mechanistic link (**Figure 7**).

5. While the focus on half-spindle size asymmetry during symmetric division is intriguing, it's important to address the broader physiological significance. The primary outcome of this asymmetry is differences in daughter cell size, which limits the broader significance of the study. Furthermore, the quantification method for daughter cell size warrants scrutiny and clarification.

With regard to the latter point, we have now improved our quantification of cell daughter size, using a method that was first established in Cheeseman laboratory (Kiyomitsu and Cheeseman., Cell, 2013). We now also provide both in the results section, in the **supplementary Figure 1** and the material and methods a clearer explanation of how we proceeded.

With regard to the broader significance of the study, we first would like to emphasize that already earlier studies have shown, how already small differences in daughter cell size can influence the length in the cell cycle and the survival rate (Kiyomitsu and Cheeseman., Cell, 2013). We aimed to reproduce these findings by performing long-term live cell imaging movies, but we were unfortunately not able to track enough cells, given the high mobility of RPE1 cells over time. We nevertheless believe that these types of questions will become in the future more easily addressable for the entire community using the direct comparison of the cell lines expressing mScarlet-Cenexin (asymmetric spindle) vs mScarlet-Cenexin S796A (symmetric cells).

(Minor points)

1. Table 1 lists factors with asymmetric localization not analyzed in detail in this paper. It would be beneficial to discuss whether these factors play a role in spindle asymmetry, and the authors should address the completeness of the data in Table 1 in terms of selecting factors for analysis.

First, we have expanded our analysis of potential factors, by investigating the contribution of ch-TOG (**Figure 5**), which we find to be also required for asymmetric spindle size formation. Second, also briefly discuss the potential role of EB1 in the discussion, the other significant “hit” that we had left out from Table 1. While with every screen it is often necessary to make a choice of the hits one follows up, we believe that we have now at minimum discussed all our potential hits in this study.

2. In Figure 1H, the impact of centriolin knock-out on the distribution of unaligned polar chromosomes is different from the effect of cenexin S796A in Figure 6H. This difference should be explained to provide clarity on the observed discrepancies.

As we state in our discussion (below), this difference is not a discrepancy, it is a result. While the cenexin-bound pool of cenexin drives both the asymmetry in spindle size and polar chromosome resolution, only the asymmetry in spindle size depends on centriolin and potential downstream factors, indicating that these two types of asymmetries work, at least partially, via different molecular mechanisms. While this study investigated the downstream molecular players involved in spindle size symmetry, future studies will be required to understand how Cenexin-Plk1 controls polar chromosome resolution in a centriolin-independent manner.

“Regarding previous studies in somatic tissue culture cells, our data show that the half-spindle size asymmetry is related, yet different to the previously identified polar chromosome asymmetry (Gasic et al., 2015; Colicino et al., 2019). Both depend on the cenexin-bound pool of Plk1, but only the spindle size asymmetry depends on centriolin at sub-distal appendages. This indicates that both types of asymmetries are independent of each other, as they can be uncoupled, and point to, at least in part, differing molecular mechanisms imposing these asymmetries.”

3. In Figure 2A, there is no correlation data presented between daughter cell asymmetry and the presence or absence of cenexin signal. This relationship should be elucidated for a more comprehensive understanding.

As we now explain more explicitly in the results section, in telophase both centrosomes contain cenexin, but the old centrosome has higher cenexin levels (Kong et al., JCB, 2014). This allowed us to correlate daughter cell size to the age of the centrosome.

4. In Figure 4G and H, the mean value of spindle asymmetry increases with siRNA treatment of Cdk5Rap2 or PCNT compared to the control. The possible interpretation of this finding should be discussed.

We now point to this data in the results section, and briefly discuss them in the context of our findings that different mechanisms might contribute to spindle length, whether centrosomes are present or not.

5. Figure 4K shows that the asymmetry of PCNT distribution is not eliminated by centriolin knock-down. This observation requires clarification and discussion.

We now explore this point specifically in the discussion mentioning that pericentrin might bind to the longer old centriole via its PACT domain (Gillingham and Munro., 2000; Kong et al., 2020).

6. It appears that the difference in spindle asymmetry of the control group in Figure 5A is smaller than in other data. This discrepancy should be addressed. Additionally, the influence of TPX2 depletion on spindle formation, and any corresponding spindle staining data, should be included.

We have added one additional replicate to confirm or infirm this result, and we found that the percentage of asymmetry in the control condition is higher than before and that now those data show a clear contribution of TPX2 to the formation of asymmetric spindle in size (**Figure 5B**).

7. Claiming that the daughter centriole recruits PCM based on Figure 6A data alone may require additional supporting evidence. It is essential to investigate whether there is a clear PCM signal when the daughter centriole disengages in late mitosis and maintain consistency in the interpretation.

As a more quantitative approach and as suggested by the reviewer 2, we measured the PCM size and the relative levels of the corresponding proteins in both 2:2 and 1:1 cells (**new Figure 8A-D**). These quantifications demonstrate that the daughter centrioles contribute to PCM recruitment in mitosis.

8. The lack of difference in TPX2 distribution in Figure 7E should be explained, along with a discussion of how this observation aligns with the spindle asymmetry data and any inconsistencies.

We explore this point in the discussion, emphasizing that the principle of an asymmetric spindle size appears to be conserved, even when the molecular mechanisms might vary from cell type to cell type. This is also consistent with our findings that this asymmetry depends on a large number of factors that might be differentially modulated in different cell types.

“Our data also suggest that the exact molecular mechanisms that creates an asymmetric microtubule nucleation might vary depending on the context, as we find that TPX2 does not correlate with spindle asymmetry in BJ cells. Given that we find that spindle size asymmetry responds to abundance of various centrosomal proteins, this might give cells the ability to modulate this asymmetry, particularly in the context of asymmetric stem cell divisions.”

9. The differing N numbers between samples in all the figures may affect the validity of comparisons. The authors should discuss whether it is necessary to have consistent N numbers in each experiment for more robust conclusions.

We agree with this important consideration. We aimed to have approximately the same number of cells when comparing similar experiments, and in particular to add more cells when claiming the absence of an asymmetry (see new TPX2 data in **Figure 5A and B**). However, for instance for the screening, we could not analyze the same number of cells in the 2:2 vs 0:0 conditions, because of the difficulty in obtaining certain configurations.

Reviewer #2:

Major comments:

1) It is not completely clear how the authors determined whether a spindle was asymmetric or not. In the methods, they say that statistical tests are described in the legends. In Figure 1 legend they say: "Each condition was compared to a theoretical distribution centered at 0 (dashed line)". How did they generate this theoretical distribution?

We have now made our calculations and statistical tests more explicit in the results section:

“By subtracting the half-spindle length associated to the young centrosome from the half-spindle length of the old centrosome, and by dividing this difference by the sum of both values, we obtained for each cell a Spindle (A)symmetry Index (SAI hereafter; Supplementary Fig. 1C; note that in 0:0 cells the two half-spindles were randomly assigned). A SAI of +5%, for example, indicates that this cell possessed an “old” half-spindle that was 5% longer than the average, or 10% longer than the “young” half-spindle. The distribution of the SAIs in each condition (2:2, 1:1, 1:0 and 0:0 cells) was statistically compared to the null hypothesis, a symmetric spindle with a SAI of 0.”

We hope that this more explicit explanation of our methodology should help the reader.

2) The authors claim that TPX2 depletion results in loss of spindle asymmetry in 1:1 cells, but the difference is very small (1.7% in control vs 1.3% in TPX2 depletion, Fig 5B) and the data is more variable in TPX2 depletion, which makes it less likely that a statistically significant difference from 0 would be found. Firstly, perhaps the authors could check the standard error of the mean, which provides a measure of how accurate the mean is with regard to N and variation. If a dataset is more spread (such as in TPX2 depletion) a higher N is required to attain the same accuracy in the mean value. This is normally not so important when directly comparing two datasets, but in this case the authors are

comparing each dataset to 0. So, are the authors measuring enough cells in the TPX2 depletion to be sure that a 1.3% value is not significantly different from 0? Secondly, I don't understand why the control cells have such a low asymmetry index (1.7%), when previous data in the paper shows an asymmetry index of 4.1% (Fig 1D) and 3.4% (Fig 4E) in control 1:1 cells. This suggests that something about the way this experiment was carried out dampens the asymmetry, which could therefore lead the authors to conclude that TPX2 is more important than it really is.

We agree with the reviewers that this result was not solid enough and have now repeated it, by measuring both more control-depleted and TPX2-depleted 1:1 cells. Our extended analysis confirmed our initial finding that control spindles tend to be asymmetric (SAI of 2.5%) and that TPX2-depleted cells have symmetric spindles (SAI of 1.1%, not significant with over 100 cells counted). We therefore feel that our interpretation rests on more solid ground.

3) The authors claim that daughter centrioles are associated with some Pericentrin and suggest that this may be why 2:2 centrosomes have less of an asymmetry than 1:1 centrosomes (Fig 6A). It is unclear whether the authors consider these daughter centrioles as being prematurely disengaged (they make reference to the fact that they previously showed how disengaged daughters recruit γ -tubulin, but it's unclear if this is related to their current observations). In Figure 6A, the Centrin spots look too far apart for engaged centrioles (~750nm). I appreciate that this may be the only way to detect Pericentrin around the daughter at this resolution, but it may also force the authors to select cells where the centrioles have prematurely disengaged. For the asymmetry measurements, the authors presumably did not select cells where they could distinguish mother and daughter centrioles. One way to address this issue would be to compare PCM size at centrosomes in 2:2 cells with centrosomes in 1:1 cells. The expectation would be that centrosomes in 2:2 cells would have more PCM, due to the contribution of the daughter centrioles.

We thank the reviewer for this suggestion. We have now quantified both the relative PCM size and the levels of pericentrin and γ -tubulin in 1:1 versus 2:2 cells and found consistent with the presence of these proteins at the daughter centrioles that the absence of daughter centrioles reduces the abundance of these proteins and the size of the PCM by roughly one third (Figure 8A-D).

4) The authors show that Plk1 recruitment by Cenexin (via S796 phosphorylation), which happens only at mother centrosomes, is important for asymmetry. Nevertheless, they show that Plk1 is symmetrically distributed between mother and daughter centrosomes (Table 1). This does not really fit, unless daughter centrosomes recruit more cenexin-independent Plk1 than mother centrosomes or if the cenexin-bound pool of Plk1 is only a minor fraction of total Plk1. If so, do the authors think that the Cenexin-bound pool of Plk1 is more potent than the rest of centrosomal Plk1?

Table 1 data showed that there is no strong correlation between Plk1 distribution at poles and spindle (a)symmetry but this analysis did not directly test whether Plk1 presence at centrosomes is centrosome age-dependent. Our indirect measurements could have masked this connection, as the connection of two partial correlations might hide the signal. We now have directly tested whether Plk1 is asymmetrically distributed with regard to centrosome age and found a significant enrichment on the old centrosome that depends on the S796 phosphorylation site of cenexin. We also show that Plk1 activity is required for spindle size asymmetry.

5) The circles drawn to measure cell size in Figures 2A,E and 7C do not look like a good representation of cell area (as the cells are not perfectly round). The authors use a formula for circle area with an approximation of the radius (based on mean length/width of an oval). It would be much better to use ImageJ to draw a freehand line around the perimeter of the cell and use the in-built tool to measure the area.

As suggested by the reviewer, rather than extracting the area based on the previous formula, we have drawn a freehand line using ImageJ to measure the real area of the cell, using the methodology established by Kiyomitsu and Cheeseman in their 2013 publication.

Minor comments:

1) Asymmetry in centrosome size that correlates with centrosome age in apparently symmetrically

dividing "cells" has been observed previously in *Drosophila* syncytial embryos (Conduit et al., 2010a, *Curr. Bio.*). I think this should be mentioned somewhere given the topic of the study.

We thank the reviewer for this information. This paper is now included in our discussion.

2) A full description of statistical tests and n numbers for each experiment should be provided in the methods, even if this duplicates information in the Figure legends.

We now have added a full description of the statistical tests performed for each experiment in the methods, as suggested by the reviewer.

OPTIONAL EXPERIMENTS:

3) Given that chTOG is very important for microtubule nucleation, it seems strange that this protein was not analysed for a potential asymmetry.

As suggested by the reviewer we examined the contribution of ch-TOG, a potent microtubule dynamics regulator that came out in our initial screen (Table 1) Similar to TPX2, we show that ch-TOG is asymmetrically distributed on the old centrosome, that its presence is necessary for the spindle size asymmetry, and that it acts downstream of pericentrin (**Figure 5F-H**).

4) Cooling-warming experiments could be done using higher concentration of formaldehyde, as it's likely that microtubule nucleation is not immediately halted when using 4% formaldehyde.

We agree that the formaldehyde solution might not immediately stop the microtubule nucleation reaction, but this will be the case for both the old and the young centrosome, and our aim here was to detect a difference between the two within the same cell. We therefore do not believe that this is a concern.

Reviewer #3:

Major points:

1) The evaluation of spindle and cell size asymmetry related to centrosome age only relies on fixed sample preparation. Cells should be followed by time-lapse microscopy as the metaphase plate position relative to the spindle poles and/or the cell cortex may fluctuate over time and as the observed differences remain in a very subtle range. This is an important possibility to consider for 1:1, 1:0 or 0:0 spindle pole configurations where centrosome integrity is impaired.

We thank the reviewer for suggesting these important complementary experiments. Since we cannot study 1:0 or 0:0 cells by live cell imaging (lack of suitable live marker for centrosome-free spindle poles), we focused our analysis on 2:2 cells expressing GFP-centrin1 and labelled with the live dye SiR-DNA. As shown in Figure 2, live cell imaging experiments confirmed: a) that the spindle is asymmetric in terms of size; b) that the symmetry persists until the onset of anaphase; and c) that our measurements of the distance between the spindle and the cell cortex are consistent between live and fixed cells.

2) Cell size asymmetry was evaluated based on cell area at the equator. Volumes will be a better indicator as daughter cell shapes can be different in telophase if they do not re-adhere at the same speed. This evaluation should also be confirmed with another readout, like the position of the cleavage furrow relative to the spindle poles in late anaphase, as again the observed differences are in a very subtle range.

As indicated in response to point 5 of reviewer 1, we have adapted our quantification method using the standard established by Kiyomitsu and Cheeseman in their 2013 publication to quantify daughter cell size. At this stage of telophase, we did not observe consequent differences in re-adherence, as the z-position of the maximal area of each daughter cell was generally the same.

3) The authors propose that differential microtubule nucleation at the spindle poles underlies spindle size symmetry breaking without providing direct evidence. If the observed spindle symmetry in the 1:1 configuration after pericentrin, CDK5RAP2 or γ -tubulin siRNA fuels this interpretation (Fig4C), the differential microtubule nucleation capacity at the spindle poles after microtubule-depolymerisation-repolymerisation assays was not evaluated in these conditions, as compared to the control situation.

We thank the reviewer for this suggestion. We have performed this experiment, which we show in **Figure 4L**. As can be seen in our images, in the absence of CDK5RAP2 there is very little microtubule nucleation from centrosomes; instead, we see microtubules emerging from chromatin region, most likely from kinetochores. As we now state in the results section:

“This suggested that centrosome-age dependent spindle asymmetry depends on the rapid centrosome-driven microtubule nucleation, and that spindles are symmetric, when alternative nucleation sources dominate, for example kinetochores”

4) If differential microtubule nucleation at the spindle poles is responsible for spindle asymmetry, overexpression of PCM proteins or γ -tubulin should be sufficient for re-establishment of symmetric protein distribution, spindle and cell size symmetry in 2:2 or 1:1 configuration. The authors should evaluate whether this is the case or not.

We have attempted to overexpress a myc-tagged Cdk5Rap2 by transient transfection, using a previously described plasmid (Graser et al., 2007, Journal of Cell Science). Unfortunately, despite testing several transfection methods and transfection times we could not observe any mitotic cell overexpressing Cdk5RAP2-Myc, which precluded further analysis. We believe that this is due to a combination of low transfection rate in RPE1-cells, the fact that the plasmid is rather large, which in our experience often further reduces the transient transfection efficiency.

5) The authors describe that the cortex-centrosome distance is not changed according to centrosome age (Fig2C), but centrosome-metaphase plate distance is (Fig1D). These observations are difficult to reconcile if differential microtubule-nucleation capacity is at play. Again, time-lapse microscopy would enable to detect over time whether only metaphase plate position relative to spindle poles is changing or if spindle pole position relative to the cell cortex is also fluctuating.

We have now measured these two parameters in live cell imaging and found similar results to the ones measured in fixed samples (**Figure 2**). Neither of our fixed or live cell imaging measurements revealed a significant difference in the centrosome-cortex distance between the old and young centrosome, at least within our sample size, whereas equivalent sample sizes revealed a difference in half-spindle sizes.

Minor points:

6) Main PCM and MT nucleation protein "depletion" do not appear to impact spindle assembly, but only spindle symmetry in 1:1 and 1:0 configurations (Fig4A and 4F-H). Can it be explained by the fact that their depletion is not always total (for pericentrin, Fig5F versus FigS2A or Fig7G)? Can they comment on this point?

The depletion efficiency of our siRNAs against pericentrin (-97%) and Cdk5Rap2 (-98%) is high, but it is true that Watanabe et al., JBC, 2020 have shown that a knock-out of these proteins impairs bipolar spindle formation in 1:0 RPE1 cells (but not U2OS or DLD1). The remaining protein might support bipolar spindle formation, but does not allow the formation of an asymmetric spindle, as other microtubule nucleation sites will dominate.

7) If centrosome age dictates spindle and cell size asymmetry through differential MT-nucleation capacity at the spindle poles, how can this process be modulated? Indeed, centrosome age is common to all cell types, but cell size asymmetry is more or less pronounced. The authors should further discuss this point based on the literature.

We thank the reviewer for this suggestion, we have now explored this point in the discussion, speculating how a modulation of the different factors contributing to spindle size asymmetry might allow cells to modulate the asymmetry depending on the biological context, for example asymmetric stem cell divisions.

April 11, 2024

RE: JCB Manuscript #202311153R

Prof. Patrick Meraldi
University of Geneva
Cell physiology and metabolism department
Centre Medical Universitaire Rue Michel Servet 1
Geneva 1211
Switzerland

Dear Prof. Meraldi:

Thank you for submitting your revised manuscript entitled "Centrosome age breaks spindle size symmetry even in "symmetrically" dividing cells". As you will see, two reviewers were satisfied with the changes in place while one expressed ongoing concerns about the level of support for the main conclusions reached. While we appreciate that some experimental data sought by this reviewer was not provided, in light of strong support from the other reviewers we are ready to proceed towards publishing your paper in JCB, pending changes to the text.

Please note that reviewer order has changed from Review Commons:

Reviewer 1 below was formerly Reviewer 3

Reviewer 2 below was formerly Reviewer 1

Reviewer 3 below was formerly Reviewer 2

A revised manuscript, which we will evaluate without further reviewer input, must include supplementary videos which were missing in this revision. In addition, the concerns of Reviewer 1 must be noted in text (in particular on variability between cell lines, and the nature and significance of the spindle pole-cortex distance fluctuations). Finally, Reviewer 3 requests text changes to the introduction and discussion.

A. MANUSCRIPT ORGANIZATION AND FORMATTING:

Full guidelines are available on our Instructions for Authors page, <http://jcb.rupress.org/submission-guidelines#revised>.

Submission of a paper that does not conform to JCB guidelines will delay the acceptance of your manuscript.

- 1) Text limits: Character count for Articles is < 40,000, not including spaces. Count includes abstract, introduction, results, discussion, and acknowledgments. Count does not include title page, figure legends, materials and methods, references, tables, or supplemental legends.
- 2) Figures limits: Articles may have up to 10 main figures and 5 supplemental figures/tables.
- 3) Figure formatting: Scale bars must be present on all microscopy images, including inset magnifications. Molecular weight or nucleic acid size markers must be included on all gel electrophoresis. Please avoid pairing red and green for images and graphs to ensure legibility for color-blind readers. If red and green are paired for images, please ensure that the particular red and green hues used in micrographs are distinctive with any of the colorblind types. If not, please modify colors accordingly or provide separate images of the individual channels.
- 4) Statistical analysis: Error bars on graphic representations of numerical data must be clearly described in the figure legend. The number of independent data points (n) represented in a graph must be indicated in the legend. Statistical methods should be explained in full in the materials and methods. For figures presenting pooled data the statistical measure should be defined in the figure legends. Please also be sure to indicate the statistical tests used in each of your experiments (either in the figure legend itself or in a separate methods section) as well as the parameters of the test (for example, if you ran a t-test, please indicate if it was one- or two-sided, etc.). Also, if you used parametric tests, please indicate if the data distribution was tested for normality (and if so, how). If not, you must state something to the effect that "Data distribution was assumed to be normal but this was not formally tested."
- 5) Abstract and title: The abstract should be no longer than 160 words and should communicate the significance of the paper for a general audience. The title should be less than 100 characters including spaces. Make the title concise but accessible to a general readership.

** Article titles in JCB cannot include quotation marks. The wording in your title nicely conveys the meaning of this work without

them.

6) Materials and methods: Should be comprehensive and not simply reference a previous publication for details on how an experiment was performed. Please provide full descriptions in the text for readers who may not have access to referenced manuscripts. We also provide a report from SciScore and an associate score, which we encourage you to use as a means of evaluating and improving the methods section.

7) Please be sure to provide the sequences for all of your primers/oligos and RNAi constructs in the materials and methods. You must also indicate in the methods the source, species, and catalog numbers (where appropriate) for all of your antibodies. Please also indicate the acquisition and quantification methods for immunoblotting/western blots.

8) Microscope image acquisition: The following information must be provided about the acquisition and processing of images:

- a. Make and model of microscope
- b. Type, magnification, and numerical aperture of the objective lenses
- c. Temperature
- d. Imaging medium
- e. Fluorochromes
- f. Camera make and model
- g. Acquisition software
- h. Any software used for image processing subsequent to data acquisition. Please include details and types of operations involved (e.g., type of deconvolution, 3D reconstitutions, surface or volume rendering, gamma adjustments, etc.).

10) Supplemental materials: There are strict limits on the allowable amount of supplemental data. Articles may have up to 5 supplemental figures. Please also note that tables, like figures, should be provided as individual, editable files. A summary of all supplemental material should appear at the end of the Materials and methods section.

13) ORCID IDs: ORCID IDs are unique identifiers allowing researchers to create a record of their various scholarly contributions in a single place. At resubmission of your final files, please provide an ORCID ID for all authors.

15) A data availability statement is required for all research article submissions. The statement should address all data underlying the research presented in the manuscript. Please visit the JCB instructions for authors for guidelines and examples of statements at (<https://rupress.org/jcb/pages/editorial-policies#data-availability-statement>).

Please note that JCB requires authors to submit Source Data used to generate figures containing gels and Western blots with all revised manuscripts. This Source Data consists of fully uncropped and unprocessed images for each gel/blot displayed in the main and supplemental figures. Since your paper includes cropped gel and/or blot images, please be sure to provide one Source Data file for each figure that contains gels and/or blots along with your revised manuscript files. File names for Source Data figures should be alphanumeric without any spaces or special characters (i.e., SourceDataF#, where F# refers to the associated main figure number or SourceDataFS# for those associated with Supplementary figures). The lanes of the gels/blots should be labeled as they are in the associated figure, the place where cropping was applied should be marked (with a box), and molecular weight/size standards should be labeled wherever possible. Source Data files will be directly linked to specific figures in the published article.

WHEN APPROPRIATE: The source code for all custom computational methods published in JCB must be made freely available as supplemental material hosted at www.jcb.org. Please contact the JCB Editorial Office to find out how to submit your custom

macros, code for custom algorithms, etc. Generally, these are provided as raw code in a .txt file or as other file types in a .zip file. Please also include a one-sentence summary of each file in the Online Supplemental Material paragraph of your manuscript.

B. FINAL FILES:

****It is JCB policy that if requested, original data images must be made available to the editors. Failure to provide original images upon request will result in unavoidable delays in publication. Please ensure that you have access to all original data images prior to final submission.****

****The license to publish form must be signed before your manuscript can be sent to production. A link to the electronic license to publish form will be sent to the corresponding author only. Please take a moment to check your funder requirements before choosing the appropriate license.****

Thank you for your attention to these final processing requirements. Please revise and format the manuscript and upload materials within 7 days. If you need an extension for whatever reason, please let us know and we can work with you to determine a suitable revision period.

Thank you for this interesting contribution, we look forward to publishing your paper in Journal of Cell Biology.

Sincerely,

Monica Bettencourt-Dias
Monitoring Editor
Journal of Cell Biology

Tim Fessenden
Scientific Editor
Journal of Cell Biology

Reviewer #1 (Comments to the Authors (Required)):

The main novelty of the manuscript by Thomas and Meraldi was, to my sense, to dissect the mechanisms that translate differences in centrosome age into spindle and daughter cell size differences. They propose that differences in MT nucleation/polymerization activity at the two spindle poles are required. I am still not convinced that these differences exist in the cells studied by the authors to an extent where they are sufficient to control metaphase plate position and consequent daughter cell sizes.

First, when the authors break spindle asymmetry after siRNA of PCM proteins (Figure 4), alternative MT nucleation pathways are activated to enable bipolar spindle formation (new Figure 4L, after Cdk5Rap2 siRNA). This is an important information, as in this case, spindle asymmetry breaking cannot be related anymore to a lack of differences in MT nucleation capacity between the

two poles. The same is true for the other factors (PCM proteins in Figure 4; TPX2 and Ch TOG in Figure 5) where spindle asymmetry breaking could not be directly related to a loss of differences in MT nucleation capacity between the two poles along with equal distribution of the factor of interest. The authors mention that "spindles are symmetric, when alternative nucleation sources dominate", so how to explain that the asymmetry is worse after siRNA of *cdk5rap 2* or pericentrin (where alternative MT nucleation pathways are activated) in the 1:0 configuration? (Figures 4G, 4H and 4L). It should instead favour spindle symmetry. Second, along this line, the authors did not provide (as advised) any experimental set-up to re-establish equal protein distribution at the two poles and evaluate the consequences on spindle asymmetry breaking. If the experiments cannot be performed after *cdk5rap2* overexpression, one of the other factors of interest can be expressed or forced to be preferentially localized at the centrosome. As only very small differences in protein accumulation at the two centrosomes are observed, only mild expression of a tagged protein of interest should be sufficient.

Third, the authors now provide better evidence after time-lapse microscopy that only half-spindle sizes but not centrosome to cortex distances are different (at least at the last time point before anaphase onset). If differences in MT nucleation capacity at the two spindle poles exist, both centrosome to cortex distances (that also rely on MTs nucleated from the poles) and metaphase plate position should be impacted. This observation is difficult to explain. One possible explanation is that the metaphase plate position is not only regulated by MTs nucleated from the spindle poles but also from spindle MTs (nucleated on MTs or from kinetochores).

Fourth, the authors chose to compare human cell lines with different number of centrioles at the spindle poles during mitosis (2:2, 1:1; 1:0 and 0:0). This strategy enabled to identify different PCM components, appendage proteins and microtubule polymerization regulators that contribute to this process. Nonetheless, their relative contributions vary from one cell type to another one (RPE1 versus BJ cells) and according to the spindle pole configurations considered (as evidenced for example in Table 1). Spindle asymmetry appears therefore as a default pathway related to centrosome age, indeed, in all normal cells, but it is the extent of asymmetry that is readily under regulation. The title is misleading regarding the main message of the paper, which needs to be reconsidered.

Reviewer #2 (Comments to the Authors (Required)):

In this revision, the authors answered most of my concerns. In particular, I found the experiment demonstrating that the asymmetry at spindle poles already started before mitotic entry to be very convincing. This result strongly supports that SDA factors are involved in this phenomenon. The physiological significance of this phenomenon has not been experimentally tested in this manuscript, but should at least be better discussed.

Reviewer #3 (Comments to the Authors (Required)):

The authors have done a great job addressing the Reviewer's comments. I only have a few more minor comments that I would like them to address before publication

1) In the intro the authors say: "In most animal cells, microtubules are first nucleated from the centrosome, the main microtubule organizing center (Sanchez and Feldman, 2017; Meraldi, 2016)." The authors should add "... during mitosis.", as centrosomes are often not the main microtubule organising centre in differentiated cells.

2) In the intro: "Old and young centrosomes are segregated in a stereotypical manner in many stem cell divisions, such as the unicellular budding yeast *Saccharomyces cerevisiae*, murine neuroprogenitors, *Drosophila melanogaster* neuroblasts, or *Drosophila melanogaster* male germline stem cell divisions (Chen and Yamashita, 2021; Yamashita et al., 2007; Januschke et al., 2013; Wang et al., 2009; Pereira et al., 2001)." Here the references Conduit and Raff 2010 and Januschke et al. 2011 are missing - they were first to show daughter centriole inheritance in fly neuroblasts. The Januschke et al. 2013 paper described the role of Centrobin in this process.

3) In the Discussion, I appreciate that the authors have now mentioned the previous finding that centrosome size asymmetry was observed in fly syncytial embryos back in 2010, but the authors seem to group this data in with findings from asymmetrically dividing stem cells. They say: "It is known since a long time that old and young centrosomes display different behaviors in stem cell divisions (Chen and Yamashita, 2021; Yamashita et al., 2007; Januschke et al., 2013; Wang et al., 2009; Pereira et al., 2001). For instance, in *Drosophila melanogaster*, PCM components are enriched on the old centrosomes in syncytial embryos (Conduit et al., 2010a)...." These embryos are not asymmetrically dividing stem cells. The authors should separate the discussion of these findings with those in stem cells and make the point that an observed aged-based centrosome asymmetry could be a common feature in symmetrically dividing cells.

Point-by-point Rebuttal

Reviewer #1 (Comments to the Authors (Required)):

The main novelty of the manuscript by Thomas and Meraldi was, to my sense, to dissect the mechanisms that translate differences in centrosome age into spindle and daughter cell size differences. They propose that differences in MT nucleation/polymerization activity at the two spindle poles are required. I am still not convinced that these differences exist in the cells studied by the authors to an extent where they are sufficient to control metaphase plate position and consequent daughter cell sizes.

First, when the authors break spindle asymmetry after siRNA of PCM proteins (Figure 4), alternative MT nucleation pathways are activated to enable bipolar spindle formation (new Figure 4L, after Cdk5Rap2 siRNA). This is an important information, as in this case, spindle asymmetry breaking cannot be related anymore to a lack of differences in MT nucleation capacity between the two poles. The same is true for the other factors (PCM proteins in Figure 4; TPX2 and Ch TOG in Figure 5) where spindle asymmetry breaking could not be directly related to a loss of differences in MT nucleation capacity between the two poles along with equal distribution of the factor of interest. The authors mention that "spindles are symmetric, when alternative nucleation sources dominate", so how to explain that the asymmetry is worse after siRNA of cdk5rap 2 or pericentrin (where alternative MT nucleation pathways are activated) in the 1:0 configuration? (Figures 4G, 4H and 4L). It should instead favour spindle symmetry.

We do not believe that the results we obtained in 1:0 cells disprove our model of spindle asymmetry in 1:1 and 2:2 cells. The asymmetry of 1:0 cell is much more consequent and thus unlikely to purely rely on the asymmetry in microtubule nucleation. In fact we have previously shown that the microtubule-associated protein HURP is specifically required for spindle asymmetry in 1:0 cells, acting via an asymmetric stabilization of kinetochore-microtubules in 1:0 spindles (Dudka et al., 2019). Moreover, the Kitagawa laboratory has shown that centriole-associated CEP192 plays a important role in spindle assembly independently of the PCM, which will create an additional layer of asymmetry in 1:0 cells. Therefore, the fact that 1:0 cells are still asymmetric in the near-absence of PCM does not contradict our model. To better reflect this line of thought, we have extended the discussion in the following way:

“Our results also point to the existence of other, centrosome-independent mechanisms that may control spindle size symmetry: while TPX2 depletion reduces spindle asymmetry in 1:0 cells, depletion of pericentrin or Cdk5Rap2 increases it; moreover, we find that the abundance of TACC3 only correlates with spindle size asymmetry in spindles with centrosome-free spindle poles. Consistently, the HURP MAP is specifically required for spindle asymmetry in 1:0 cells by controlling kinetochore-microtubule stability (Dudka et al., 2019). Such mechanisms might be particularly important in centrosome-free systems, such as meiotic oocytes. In line this hypothesis, TACC3 and TPX2, whose abundance also strongly correlated with spindle size asymmetry in 1:0 cells, but also HURP are particularly important for spindle assembly in murine and human oocytes (Brunet et al., 2008; Wu et al., 2022; Breuer et al., 2010).”

Second, along this line, the authors did not provide (as advised) any experimental set-up to re-establish equal protein distribution at the two poles and evaluate the consequences on spindle asymmetry breaking. If the experiments cannot be performed after cdk5rap2 overexpression, one of the other factors of interest can be expressed or forced to be preferentially localized at the centrosome. As only

very small differences in protein accumulation at the two centrosomes are observed, only mild expression of a tagged protein of interest should be sufficient.

We agree that a such experiment would have been interesting to test. Unfortunately, and as mentioned in our previous rebuttal letter, we have not been successful in overexpressing Cdk5Rap2, despite numerous attempts. We focused on this cell as we had previously tried for an unrelated project to create a with lentiviruses stable cell line expressing exogenous pericentrin, which failed. Moreover, while an abrogation of spindle asymmetry would have strongly supported our model, a negative result would not have told us much, as the asymmetry in protein distribution might not depend on the overall abundance of the protein, but more on the rate-limiting binding sites on the old and young centrosome.

Third, the authors now provide better evidence after time-lapse microscopy that only half-spindle sizes but not centrosome to cortex distances are different (at least at the last time point before anaphase onset). If differences in MT nucleation capacity at the two spindle poles exist, both centrosome to cortex distances (that also rely on MTs nucleated from the poles) and metaphase plate position should be impacted. This observation is difficult to explain. One possible explanation is that the metaphase plate position is not only regulated by MTs nucleated from the spindle poles but also from spindle MTs (nucleated on MTs or from kinetochores).

As has been elegantly shown by Kiyomitsu and Cheeseman in their 2012 study, the position of the centrosomes with respect to cell cortex is controlled mainly by a dynamic feedback loop, in which pole-bound Plk1 displaces dynein from the cell cortex, leading to a rocking of the entire spindle that on average ensures a central position of the spindle. There is no indication that this feedback loop is impacted by microtubule nucleation. Moreover, if an unequal microtubule nucleation were to create a force imbalance in terms of cortical forces, with more microtubules pulling on the old centrosome, this would most likely be compensated the unequal Plk1 distribution, which would more rapidly displace dynein from the pole facing the old centrosomes. We therefore believe that our findings are consistent with the current stand of knowledge on spindle positioning, but we thank the reviewers for raising this point, which we now state explicitly in the discussion:

“Microtubule nucleation-dependent spindle size asymmetry also biases the placement of the cell division plane, resulting in asymmetric daughter cell sizes. It did, however, not bias the placement of the whole spindle, which relies on a dynamic negative feedback loop, in which spindle pole-bound Plk1 displaces cortical dynein in a proximity-dependent manner (Kiyomitsu and Cheeseman, 2012). Since old centrosome contain more Plk1 this most likely counterbalances any force imbalance due to additional microtubules at old poles reaching the cell cortex.”

Fourth, the authors chose to compare human cell lines with different number of centrioles at the spindle poles during mitosis (2:2, 1:1; 1:0 and 0:0). This strategy enabled to identify different PCM components, appendage proteins and microtubule polymerization regulators that contribute to this process. Nonetheless, their relative contributions vary from one cell type to another one (RPE1 versus BJ cells) and according to the spindle pole configurations considered (as evidenced for example in Table 1). Spindle asymmetry appears therefore as a default pathway related to centrosome age, indeed, in all normal cells, but it is the extent of asymmetry that is readily under regulation. The title is misleading regarding the main message of the paper, which needs to be reconsidered.

We agree with the reviewer that that what is conserved in all normal cell lines is a centrosome-age dependent spindle size asymmetry that depends on proteins implicated in microtubule nucleation. We also agree that our data suggest that the extent of this asymmetry and the specific regulators of this

asymmetry might vary from cell line to cell, as suggested by the partial differences between RPE1 and BJ cells. We have now made this line of thought more explicit in the discussion:

“Our comparison of two different cell lines suggests that the principle of an asymmetric spindle size is conserved in human cells, but that the molecular regulation of this process can vary depending on the cellular background, as we find that TPX2 does not correlate with spindle asymmetry in BJ cells. Since spindle size asymmetry responds to the abundance of multiple centrosomal proteins, this might give cells the ability to modulate this asymmetry, particularly in the context of asymmetric stem cell divisions”

We also modified the title to better highlight this central message and avoid possible misunderstandings.

Reviewer #2:

In this revision, the authors answered most of my concerns. In particular, I found the experiment demonstrating that the asymmetry at spindle poles already started before mitotic entry to be very convincing. This result strongly supports that SDA factors are involved in this phenomenon. The physiological significance of this phenomenon has not been experimentally tested in this manuscript, but should at least be better discussed.

We thank the reviewer for this comment, we have now extended this part of the discussion, highlighting how unequal daughter cell size can not only affect cell survival (Kiyomitsu and Cheeseman, 2013), but also the distribution of intracellular organelles, as has been seen in male germline stem cell divisions (Chen et al. 2016)

“Even though the difference in daughter size is subtle, previous studies showed that such differences can impact the length of the ensuing G1 phase and the probability of cell death (Kiyomitsu and Cheeseman, 2013). Altering the symmetry of the cell division could also bias the inheritance of organelles between the two daughter cells, as has been seen in D. melanogaster male germline stem cell divisions (Chen et al., 2016).”

Reviewer #3:

The authors have done a great job addressing the Reviewer's comments. I only have a few more minor comments that I would like them to address before publication

1) In the intro the authors say: "In most animal cells, microtubules are first nucleated from the centrosome, the main microtubule organizing center (Sanchez and Feldman, 2017; Meraldi, 2016)." The authors should add "... during mitosis.", as centrosomes are often not the main microtubule organising centre in differentiated cells.

We agree with the reviewer that this sentence required more precision, we have added this information.

2) In the intro: "Old and young centrosomes are segregated in a stereotypical manner in many stem cell divisions, such as the unicellular budding yeast *Saccharomyces cerevisiae*, murine neuroprogenitors, *Drosophila melanogaster* neuroblasts, or *Drosophila melanogaster* male germline stem cell divisions (Chen and Yamashita, 2021; Yamashita et al., 2007; Januschke et al., 2013; Wang et al., 2009; Pereira et al., 2001)." Here the references Conduit and Raff 2010 and Januschke et al. 2011 are missing - they

were first to show daughter centriole inheritance in fly neuroblasts. The Januschke et al. 2013 paper described the role of Centrobin in this process.

We thank the reviewer for pointing out this imprecision, we have now corrected this point.

3) In the Discussion, I appreciate that the authors have now mentioned the previous finding that centrosome size asymmetry was observed in fly syncytial embryos back in 2010, but the authors seem to group this data in with findings from asymmetrically dividing stem cells. They say: "It is known since a long time that old and young centrosomes display different behaviors in stem cell divisions (Chen and Yamashita, 2021; Yamashita et al., 2007; Januschke et al., 2013; Wang et al., 2009; Pereira et al., 2001). For instance, in *Drosophila melanogaster*, PCM components are enriched on the old centrosomes in syncytial embryos (Conduit et al., 2010a)...." These embryos are not asymmetrically dividing stem cells. The authors should separate the discussion of these findings with those in stem cells and make the point that an observed aged-based centrosome asymmetry could be a common feature in symmetrically dividing cells.

We thank the reviewer for pointing this out. We now specify in separate sentences how unequal PCM distribution has been observed in stem cells and syncytial embryos.